# Benchmark on Drug Target Interaction Modeling from a Drug Structure Perspective

## Abstract

The prediction modeling of drug-target interactions is crucial to drug discovery and design, which has seen rapid advancements owing to deep learning technologies. Recently developed methods, such as those based on graph neural networks (GNNs) and Transformers, demonstrate exceptional performance across various datasets by effectively extracting structural information. However, the benchmarking of these novel methods often varies significantly in terms of hyperparameter settings and datasets, which limits algorithmic progress. In view of these, we conducted a comprehensive survey and benchmark for drug-target interaction modeling from a structure perspective, via integrating tens of explicit (i.e., GNN-based) and implicit (i.e., Transformer-based) structure learning algorithms. To this end, we first unify the hyperparameter setting within each class of structure learning methods. Moreover, we conducted a macroscopical comparison between these two classes of encoding strategies as well as the different featurization techniques that inform molecules' chemical and physical properties. We then carry out the microscopical comparison between all the integrated models across the six datasets, via comprehensively benchmarking their effectiveness and efficiency. Remarkably, the summarized insights from the benchmark studies lead to the design of model combos. We demonstrate that our combos can achieve new state-of-the-art performance on various datasets associated with cost-effective memory and computation.

## 1 Introduction

The prediction modeling of drug-target interactions (DTI) has emerged as an irreplaceable task for efficacious therapeutic interventions. The binding affinity between a drug molecule and its target protein plays a significant role in the design and repurpose of drugs, where a high affinity typically indicates the desired therapeutics, target specificity, long residence, and drug resistance delay (Hughes et al., 2011; Copeland et al., 2006; Swinney, 2004). The precise modeling of DTI can expedite the drug discovery process and circumvent the associated cost (Ashburn & Thor, 2004; Strittmatter, 2014). Deep learning-based frameworks have recently revolutionized this field, enabling more accurate predictions and accelerating the discovery of new compounds by guiding laboratory experiments more efficiently (Wen et al., 2017; Abbasi et al., 2021; Huang et al., 2020a).

Within deep learning frameworks (Öztürk et al., 2018; 2019), drugs are commonly represented using the Simplified Molecular Input Line Entry System (SMILES) (Weininger, 1988a), and proteins are represented as sequences of amino acids. These representations are typically processed using various neural network architectures, such as convolutional neural networks (CNNs) (Krizhevsky et al., 2017; He et al., 2016), recurrent neural networks (RNNs), transformers and so on, before being integrated and processed by a multi-layer perceptron (MLP) for DTI prediction. It is notorious that the reliance on sequence-based representations can result in the loss of structural information, which can potentially compromise the DTI predictive capability. From the drug perspective, molecular structure modeling helps identify the specific binding sites (Ma et al., 2011), contribute to predicting pharmacokinetic properties (Ekins et al., 2007), and allow conformational flexibility (Karplus & Kuriyan, 2005).

To address this problem, a number of drug algorithms have been proposed to promote DTI prediction, which can be categorized into explicit and implicit structure learning. First, graph neural networks (GNNs) (Kipf & Welling, 2016; Nguyen et al., 2020) have been widely adopted to learn the molecular structures, owing to their ability to directly operate on graph-based representations of molecules. By explicitly propagating information through the graph, GNNs can learn node and edge features and thereby capture the structural and functional relationships between atoms and bonds. Second, Transformers, originally focused on natural language processing (Vaswani et al., 2017a), have also shown promise in biomedical applications (Huang et al., 2020b; Chen et al., 2020). They rely on self-attention mechanisms to implicitly weight the correlations between different parts of the input SMILES, allowing them to capture long-range dependencies and contextual information.

While these techniques contribute to the learning of drug structures, there is still a key knob under-explored: we lack a systematic study to benchmark their effectiveness and efficiency. Without such a standardized benchmark, it is unachievable to offer fair comparisons and subsequently summarize designing philosophy necessary to inform DTI. There have been several surveys and benchmarks on computational methods for DTI prediction (Öztürk et al., 2018; Huang et al., 2020a; 2021; Xu et al., 2022), which leave out the recent developments of structure learning algorithms and unavoidably fail to focus on drug structure benchmarking. Moreover, although massive efforts (Bal et al., 2024; Zhu et al., 2023; Nguyen et al., 2020) have been made to explore the effectiveness of modeling structural information, they predominantly use their proprietary training hyperparameters, datasets, and evaluation metrics. Due to the various settings, one cannot reach convincing answers whether a configuration of structure encoders and/or featurization methods generally performs well. The complex of DTI classification and regression tasks and datasets complicates the benchmark comparison.

In this study, we introduce GTB-DTI, a comprehensive **b**enchmark customized for **G**NN and **T**ransformer-based methodologies for **DTI** prediction. I) We thoroughly examine the implementation details for each category of drug structure learning methods and integrate three widely-used datasets for classification and regression tasks, respectively. Then, we harmonize the sensitive hyperparameters across different methods using a greedy search to identify an optimal *sweet spot* configuration. The unified setting lays the foundation for a fair and reproducible benchmark. II) To gain macroscopical insights into the structure encoders and featurization methods, we fix the drug encoder to be either GNN or Transformer-based approaches and benchmark these two strategies in the various settings. We also integrate tens of drug features given their importance to inform molecules' chemistry and physical properties and evaluate them on the representative datasets. III) To gain macroscopical insights into nuance between 31 concerned models, we conduct the benchmark studies of their effectiveness on the six datasets with the unified setting, Moreover, we assess the efficiency of each method by measuring peak GPU memory usage, running time, and convergences. IV) The comprehensive study finally provides a number of surprising observations: *i*) The CNN-encoder accompanied with integer features has the close protein embedding performance compared to the Transformer or larger language models, but they are more efficient. *ii*) The explicit and implicit structure encoders for drugs exhibit unequal performances across the different datasets, which suggests their hybrid usage for generalization purpose. *iii*) Inspired from these insights, we conclude with a model combos that leads us to attaining state-of-the-art (SOTA) regression results and performing similarly to SOTA in the DTI classifications. Our combos further deliver cost-effective memory usage and running time as well as faster convergence, which can serve as new baseline for the following explorations.

## 2 Formulations for Drug-target Interaction Modeling

In this research, we focus on the formulations of recently-emerging structure modeling approaches for drug molecules, which could be categorized into explicit methods based on graph neural networks and implicit methods based on transformer. The target proteins are learned by the sophisticated tools of convolutional/recurrent neural networks (CNNs/RNNs) or transformers, after which both the molecules' and proteins' embeddings are integrated to facilitate interaction prediction. We will also summarize and benchmark the various widely-adopted molecule features.

## 2.1 Graph Neural Networks based Methods

A drug molecule is typically represented as a graph $G = (\mathcal{V}, \mathcal{E})$, where $\mathcal{V}$ and $\mathcal{E}$ denotes the sets of atoms and chemical bonds, respectively. The classical GNN frameworks involve key processes of aggregating and updating node features, collectively referred to as message passing, which can be mathematically represented as (Scarselli et al., 2008; Duan et al., 2022):

$$\mathbf{h}_i^{(l+1)} = \text{COMBINE}_{\text{node}}^{(l)} \left( \mathbf{h}_i^{(l)}, \text{AGGREGATE}_{\text{node}}^{(l)} \left( f_\alpha \left( \left\{ \mathbf{h}_j^{(l)}, \mathbf{e}_{ij}^{(l)} : j \in \mathcal{N}_i \right\} \right) \right) \right), \tag{1}$$

$$\mathbf{e}_{ij}^{(l+1)} = \text{COMBINE}_{\text{edge}}^{(l)} \left( \mathbf{e}_{ij}^{(l)}, \text{AGGREGATE}_{\text{edge}}^{(l)} \left( g_\beta \left( \left\{ \mathbf{h}_i^{(l)}, h_j^{(l)} : j \in \mathcal{N}_i \right\} \right) \right) \right), \tag{2}$$

where $\mathbf{h}_i^{(l)}$ is the feature representation of node $v_i$ at layer $l$, $\mathbf{e}_{ij}^{(l)}$ is the feature representation of edge between nodes $v_i$ and $v_j$, $\mathcal{N}_i$ refers to the set of neighboring nodes next to node $v_i$. Functions $\text{AGGREGATE}^{(l)}$ and $\text{COMBINE}^{(l)}$ aim to aggregate the neighborhood representations and integrate them together with the nodes features, respectively. Additionally, $f_\alpha$ and $g_\beta$ are feature mapping functions, parameterized by $\alpha$ and $\beta$, respectively. The molecule's representation can be derived using READOUT function, which processes on the set of vertex features $\mathbf{H}^{(L)}$ at the last layer.

**Graph Convolutional Networks (GCN).** Given a molecule with $N$ atoms, the adjacency matrix $\mathbf{A} \in R^{N \times N}$ indicates its connectivity, with $A_{ij} = 1$ if atom $v_i$ is adjacent to atom $v_j$, and 0 otherwise. Considering the self-connection of atoms, we have $\tilde{\mathbf{A}} = \mathbf{A} + \mathbf{I}$. Let $\mathbf{X} \in R^{N \times C}$ denote the initial atom feature matrix. GCN (Kipf & Welling, 2017) models the message passing as follows:

$$\mathbf{H}^{(l+1)} = \sigma(\tilde{\mathbf{D}}^{-\frac{1}{2}} \tilde{\mathbf{A}} \tilde{\mathbf{D}}^{-\frac{1}{2}} \mathbf{H}^{(l)} \mathbf{W}^{(l)}), \tag{3}$$

where $\mathbf{H}^{(l)}$ is the node feature matrix at layer $l$, starting with $\mathbf{H}^{(0)} = \mathbf{X}$. Matrix $\mathbf{W}^{(l)}$ represents the learnable weights for layer $l$, $\sigma$ denotes a non-linear activation function, e.g., ReLU, and $\tilde{\mathbf{D}}$ is a diagonal degree matrix of $\tilde{\mathbf{A}}$. A couple of pioneering works have leveraged GCN to facilitate drug-protein interaction prediction (Mukherjee et al., 2022; Tran et al., 2022; Tsubaki et al., 2018; Pan et al., 2023b). For example, DeepGLSTM (Mukherjee et al., 2022) uses mixture-of-depths GCNs to capture drug representations from different scales. CPI (Tsubaki et al., 2018) considers cross-atom distance and introduces the concept of r-radius subgraphs (Costa & Grave, 2010), using r-radius vertices and edges to redefine the structure of graphs.

**Graph Isomorphism Networks (GIN).** GIN excels in learning distinct graph features by approximating the Weisfeiler-Lehman test, enabling it to distinguish a wide range of graph structures (Xu et al., 2018). The message passing process at the $(l+1)$-th layer is of the following form:

$$\mathbf{h}_i^{(l+1)} = \text{MLP}^{(l)}((1 + \epsilon^{(l)})\mathbf{h}_i^{(l)} + \sum_{j \in \mathcal{N}_i} \mathbf{h}_j^{(l)}), \tag{4}$$

where $\text{MLP}^{(l)}$ is a multi-layer perceptron that parameterizes the update function, and $\epsilon^{(l)}$ is a learnable parameter. We benchmark several GIN-based drug-target interaction modeling methods. GraphCPI (Quan et al., 2019) and GraphDTA (Nguyen et al., 2020) adopt GIN-based models with batch normalization to obtain the drug representation. SubMDTA (Pan et al., 2023a) uses subgraph's generation task and contrastive learning to pretrain a molecular graph encoder with multiple GIN layers for further prediction.

**Graph Attention Networks (GAT).** Unlike fixed-weight aggregation, GAT (Velikovi et al., 2018) employs an attention mechanism to determine neighborhood importance and learn the node embeddings as:

$$\mathbf{h}_i^{(l+1)} = \sigma( \sum_{j \in i \cup \mathcal{N}_i} \text{softmax}(\text{LeakyReLU}(\mathbf{W}_a^{\text{T}}[\mathbf{W}^{(l)}\mathbf{h}_i^{(l)}||\mathbf{W}^{(l)}\mathbf{h}_j^{(l)}]))\mathbf{W}^{(l)}\mathbf{h}_j^{(l)}). \tag{5}$$

$\mathbf{W}_a^{\text{T}}$ denotes attention weights, and $||$ is concatenating operation. GraphDTA (Nguyen et al., 2020) and AMMVF (Wang et al., 2023) leverage the multi-head GAT layers to optimize the atom messaging. They integrate GAT with other architectural modules, such as GCN, facilitating a more comprehensive representation of drugs.

**Graph Transformers.** Graph Transformers (Rong et al., 2020; ukasz Maziarka et al., 2020) have emerged as powerful alternatives to traditional graph neural networks (GNNs) for molecular representation learning. Unlike conventional GNNs, which rely on message-passing mechanisms to propagate local node information, Graph Transformers leverage self-attention mechanisms to capture both local and global dependencies more effectively. By integrating Message Passing Networks into Transformer-style architectures, these models enhance expressiveness, enabling more comprehensive encoding of molecular structures. This hybrid approach allows Graph Transformers to preserve structural information while benefiting from the flexibility of attention-based learning.

## 2.2 Transformer-based Methods

Besides the graph representation, drugs could also be decorated as SMILES strings (Weininger, 1988b) and encoded similarly to natural language processing. Specifically, after tokenizing SMILES strings, Transformer model utilizes multi-head attention to model the interactions between different segments of the input and obtain the molecular representations. Positional encodings are also integrated to preserve the sequence order, enhancing the model's ability to process sequential information effectively. We review and benchmark two typical types of attention mechanisms used for molecular representations.

**Self-Attention**(Huang et al., 2020b; Qian et al., 2023; Yin et al., 2024)**.** Self-attention computes a weighted sum of all input values based on their relevance to each other. Considering an embedding of SMILES sequence $\mathbf{H}^{(l)} \in \mathbb{R}^{d \times N}$ at a specific transformer layer, where $N$ and $d$ are token length and dimension, respectively, the attention is calculated by $\text{Attention}(\mathbf{Q}, \mathbf{K}, \mathbf{V}) = \text{softmax}(\mathbf{Q}\mathbf{K}^{\text{T}}/\sqrt{d_k})\mathbf{V}$. $\mathbf{Q}, \mathbf{K} \in \mathbb{R}^{d_k \times N}$ and $\mathbf{V} \in \mathbb{R}^{d_v \times N}$ are projections of the input matrix $\mathbf{H}^{(l)}$. Multi-head attention combines these projections across different subspaces for a more detailed analysis. Following by normalization and feed-forward neural networks, the SMILES embedding is updated as $\mathbf{H}^{(l+1)}$ and the output from the last layer is treated as molecular representations. Transformer encoders like MolTrans (Huang et al., 2020b) and FOTFCPI (Yin et al., 2024) are adopted to enhance sub-structure embeddings in proteins and drugs.

**Cross Attention**(Kurata & Tsukiyama, 2022; Qian et al., 2023)**.** Cross-attention is designed to capture the interaction between the drug and protein sequences, with the query matrix $\mathbf{Q}$ derived from one sequence and the key and value matrices $\mathbf{K}, \mathbf{V}$ from another. This mechanism is particularly useful in integrating hybrid representations such as drug graphs and SMILES (Wang et al., 2023), as well as drugs and proteins (Pan et al., 2023b; Kurata & Tsukiyama, 2022).

## 2.3 Feature Processing Methods

Beyond the drugs' structure or sequence learning with GNNs or Transformers, the extra molecular properties, such as molecular weight, solubility, and lipophilicity, are crucial for building accurate and quantitative drug-target relationship models. We summarize two typical featurization methods.

**Sequence Processing Methods.** Both drugs and proteins are input as strings of ASCII characters, whose features can be extracted using statistical solutions. Integer encoding (Nguyen et al., 2020) simply converts the string to a sequence of integers, which assigns an integer to each character. The N-gram (Dong et al., 2005) captures the statistical dependencies between characters in an input string. Specifically, a 3-gram model breaks down a sequence $S = \{s_1, s_2, ..., s_m\}$ into $\{[s_1, s_2, s_3], [s_2, s_3, s_4], ..., [s_{m-2}, s_{m-1}, s_m]\}$, analyzing the relationship between adjacent characters.

**Drug-unique Featurization Methods.** The additional chemical properties and structural details of SMILES strings are often considered to gain a more comprehensive understanding. Extended-Connectivity Fingerprints (ECFP) (Morgan, 1965; Rogers & Hahn, 2010), involves generating unique identifiers for atoms based on their local chemical environment and iteratively updating these through a hash function to capture a broader molecular context, ultimately producing a set of fingerprints that represent the molecules overall structure. Another approach, RDKit, is used to convert SMILES into molecular graphs (Landrum et al., 2006; Nguyen et al., 2020), where nodes represent the physical and chemical properties of molecules, and bonds are represented by an adjacency matrix. For example, atomic properties such as atom type, degree,

and hydrogen information (like the number of explicit hydrogens) are all crucial for constructing a graph. More detailed properties can be found in Appendix G.

**Embedding Featurization Methods.** Embedding methods are used to translate these discrete sequences into continuous embedding spaces. Notably, Smi2Vec (Quan et al., 2018) and Prot2Vec (Asgari & Mofrad, 2015) convert discrete tokens of drug SMILES and protein sequences into vectors that encapsulate semantic and syntactic similarities, effectively grouping similar tokens together in vector space. Additionally, pre-trained language models (Bal et al., 2024; Lin et al., 2022) are increasingly utilized to leverage large-scale learned patterns, fine-tuned to analyze complex protein data representations effectively.

## 3 A Fair Benchmark Platform Setup

**Benchmark Model and Dataset Selection.** From the perspective of reproducibility, we restrict our analysis to models for which the source code has been publicly released. To enhance the comprehensiveness, credibility, and sophistication of our benchmark, we conduct experiments on more than 30 models, including both GNN-based and Transformer-based methods. These models are derived from papers spanning the years 2018 to 2024. We run these models on 6 frequently evaluated datasets including both binary interaction classification and continuous affinity regression. For the classification aspect, we utilize datasets including Human (Liu et al., 2015), *Caenorhabditis elegans* (*C. elegans*) (Tsubaki et al., 2018), and DrugBank (Wishart et al., 2008). For regression, we employ the Davis (Davis et al., 2011), KIBA (Tang et al., 2014), and BindingDB (Liu et al., 2007) with dissociation constant (Kd) measures datasets, as processed in Huang et al. (2021). The statistical details of these models and datasets are presented in Appendix B and Table 3, respectively. In all experiments, we employ the five-fold cross-validation method with a random split to evaluate all different methods and report the averaged results.

**Unifying Hyperparameter Configuration.** Given the critical role of hyperparameters in achieving optimal performance, we perform a detailed review of the hyperparameters associated with the selected models in Appendix F. There is significant variability in the hyperparameters across different models, making it unfair to conduct comparisons directly. To achieve equitable comparisons between varied models, we select two representative approaches from both the GNN-based and Transformer-based categories, i.e., GraphDTA (Nguyen et al., 2020), GraphCPI (Quan et al., 2019), MRBDTA (Zhang et al., 2022), and TransformerCPI (Chen et al., 2020), to perform a greedy hyperparameter search to find their *sweet spot* for classification and regression tasks, respectively. For the search space of hyperparameters, we mainly focus on the influence of batch size (BS), learning rate (LR), and dropout rate (DR), as these are the common hyperparameters utilized by all models. We use a weight decay of 0 and the Adam optimizer for all models according to Table 4. Considering the original training epochs, we use 1000 epochs for GNNs and 300 epochs for transformers. We illustrate the selected results for the metrics MSE and CI for the regression task, along with AUC-ROC and accuracy for the classification task, in Fig. 1 and the results of all metrics in Table 5. We observe that different models exhibit distinct preferences for hyperparameters. Taking into account various models and metrics, we recommend the configuration $\{512, 0.0005, 0.1\}$ as the *sweet point* hyperparameter configuration for the GNN-based model. Similarly, for the Transformer-based model, we suggest $\{128, 0.0005, 0.1\}$.

**Optimized Hyperparameter Configuration and Convergence.** To ensure a fair comparison, we also consider comparing each model using its optimal hyperparameters, as reported in the corresponding perspective papers. We use 1000 epochs for GNNs and 500 epochs for transformers. To avoid overfitting, we consider an early stop mechanism in training. Considering the complexity and size of the datasets, we use 10 patients for Human and Celegans. For other datasets, we use 50 patience. We use MSE as early stopping evaluation metric for regression and F1 for classification.

## 4 A Macroscopical Benchmark on Encoder and Featurization Strategies

⋆**Encoder Exploration for Drugs and Proteins.** To investigate the influence of different encoding strategies for extracting the structural information of drugs, we employ GIN (Xu et al., 2019) and vanilla Transformer (Vaswani et al., 2017b) as the encoders for drugs. Meanwhile, integer encoding with CNN, n-gram encoding with CNN, and the vanilla Transformer are considered to capture protein's representations,

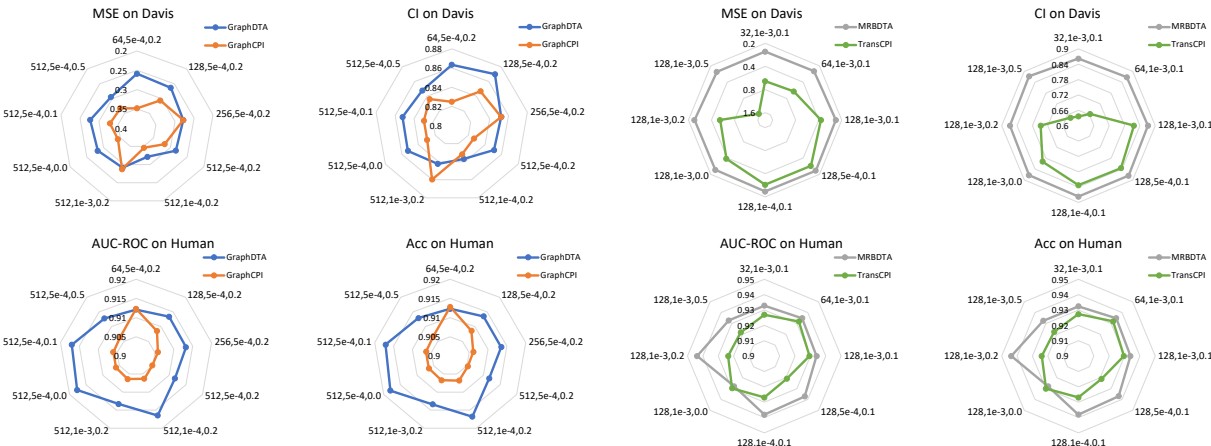

Figure 1: The greedy hyperparameter searching results on Davis and Human datasets. BS traverses from $\{64, 128, 256, 512\}$ for GNN-based approaches, and $\{32, 64, 128\}$ for Transformer-based approaches. LR and DR are selected from $\{0.0001, 0.0005, 0.001\}$ and $\{0, 0.1, 0.2, 0.5\}$, respectively. The outer values represent hyperparameter settings, while the plotted values indicate model performance on corresponding metrics.

which are frequently adopted. To leverage the advantages of the pretrained protein information, we include a language model, i.e., Evolutionary Scale Modeling (ESM2) (Lin et al., 2022). The results of various combinations of drug and protein encoders are shown in Fig. 2. All results are averaged by five-fold cross-validation with an early stop mechanism.

**Obs. 1. GNN and Transformer-based drug encoders exhibit unequal performance depending on DTI tasks.** When the encoder for the protein sequence is fixed, drug features extracted by the GNN structures GIN generally perform better than those by Transformers in regression tasks, but the opposite is true in classification tasks. This disparity may be due to the smaller size of the Human dataset compared to the Davis dataset, which allows for faster convergence in classification tasks than in regression tasks.

**Obs. 2. Transformer models are better but sensitive in extracting features from protein.** Although we only consider the simplest pretrained protein language model of ESM2, it still significantly outperforms other encoders in both tasks. This improvement can likely be attributed to the robust and generalizable representations learned from extensive data by the pretrained model. In addition, the Transformer encoder for the protein achieves the best performance on the classification task but shows unstable performance in the regression task. This is likely due to the smaller size and simpler classification dataset compared to the regression dataset, making the training stops for a fixed early stop threshold.

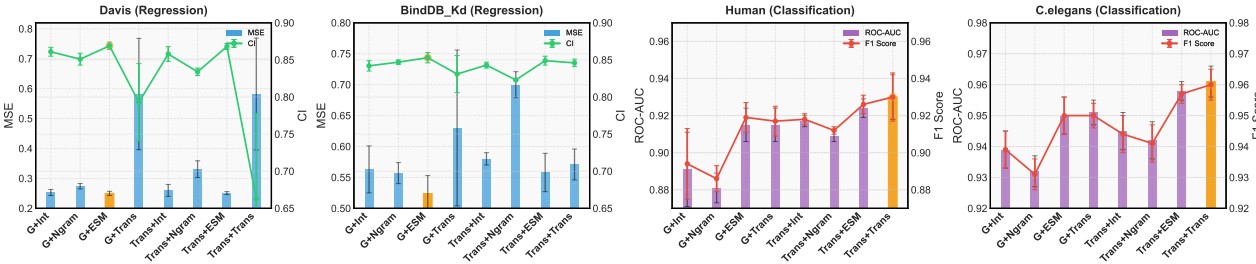

Figure 2: Comparison of different encoding strategies with early stop mechanism for drugs and proteins when the total epoch is 1000, LR is 0.0005, BS is 512, and DR is 0.2. Trans is a Transformer-based model, which is composed of two parts: embedding with the position encoding and the encoder in the Transformer.

**Obs 3. Integer encoding appears to be more effective when paired with a CNN as the protein encoder and a fixed drug encoder.** Compared to this specific model configuration, the local context

provided by 3-gram encoding does not significantly enhance the model's predictive performance. This implies that the simple relationships in amino acids' immediate neighbors, as modeled by Word2Vec, do not capture much useful information compared with simple integer encoding.

⋆**Featurization Exploration.** Despite the efficacy of GNNs in learning drug structures, the featurization of nodes plays a critical role in capturing both the intrinsic properties of atoms and their contextual relevance. We conduct a detailed analysis of various methods (summarized in Section G of Appendix) for constructing graph features within the DTI context. The node feature is constructed via various characteristics, such as chemical and physical properties. We categorize each feature into five main classes, e.g., atomic properties (AP), hydrogen information (HI), electron properties (EP), stereochemistry (Ste), and structural information (Str). To better determine which types of features are more effective in capturing the structural information, we conduct an ablation study on the different featurization strategies. We choose GprahDTA (Nguyen et al., 2020) and GraphCPI (Quan et al., 2019) with GIN as our backbone models. The results of feature combinations are reported in Fig. 3.

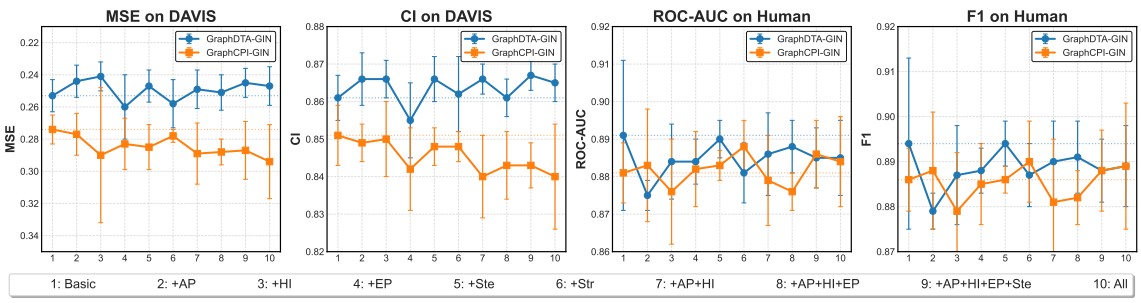

Figure 3: Various performance of GraphDTA-GIN and GraphCPI-GIN versus different features on DAVIS and Human datasets. $+x$ means that $x$ is added to the basic featurization. All means using all features.

**Obs. 4. More complex featurization does not necessarily bring a positive effect, and its effectiveness is highly task-dependent.** As shown in Fig. 3, adding features like atomic properties (AP), hydrogen information (HI), and stereochemistry information (Ste) improves performance in the regression task by reducing the MSE loss, suggesting that these features provide valuable information. However, features like electron properties (EP) and structural information (Str) may introduce noise rather than useful information, especially when combined with other features, leading to inconsistent results. Furthermore, in classification tasks, the trend differs, with additional features sometimes negatively impacting performance (blue line) rather than providing benefits. This highlights the importance of careful feature selection, as indiscriminate inclusion of complex features may lead to increased noise, affecting model generalization and robustness.

**Obs. 5. Protein representation plays a crucial role in the effectiveness of different drug featurizations.** As shown in Fig. 3, even when using the same drug featurization strategies, the trends vary depending on the protein representation. This suggests that the way proteins are encoded directly influences how drug features interact with the model. For instance, in GraphDTA, features like atomic properties (AP) and hydrogen information (HI) help improve performance, but in GraphCPI, those same features dont always provide the same benefits. In some cases, adding more drug features introduces noise rather than useful information. This highlights the fact that drug featurization and protein representation are deeply interconnected, and optimizing one without considering the other may not yield the best results. To build more effective drug-protein interaction models, both components should be considered holistically rather than in isolation.

## 5 A Microscopic Benchmark on DTI Models

⋆**Benchmark over Effectiveness.** As shown in Table. 1 and Table. 2, we conduct experiments on models with their optimal hyperparameters across two tasks and three datasets, respectively (see more comprehensive

model comparisons in Appendix I). For the unified hyperparameters, we summarized the result in Appendix H. All results are averaged by five-fold cross-validation with an early stop mechanism.

**Obs. 6. Molecular graphs are better than fingerprints for capturing the graph features of a drug.** In reference to Table. 6, it is evident that GNN-based approaches utilizing the molecular graph generally yield superior performance compared with fingerprints (CPI (Tsubaki et al., 2018), BACPI (Li et al., 2022), GANDTI (Wang et al., 2021)). This reinforces the idea that the rich structural and atom property information inherent to molecular graphs is pivotal for representation extraction, leading to enhanced model performance.

**Obs. 7. Graph structure is a crucial part of extracting a drug's features.** Different GNNs have distinct performances in both tasks when the protein representation is fixed. Specifically, GIN, with its unique ability to distinguish non-isomorphic graphs, consistently outperforms other models across different protein encoders in regression tasks. Although transformer-based methods such as MRBDTA are proficient in handling sequential information from SMILES and proteins, the depth of information they capture appears to be marginally less comprehensive than that provided by molecular graph-based approaches. This is substantiated by the superior performance of GNN-based methods, including MGraphDTA, ColdDTA, and SubMDTA, which suggest that GNN captures intricate structural details more effectively.

Table 1: Regression task benchmark on DAVIS, KIBA, and BindingDB_Kd datasets, respectively. For the GraphDTA and GraphCPI, we only show the one with a specific GNN encoder that has the best performance. The best result is highlighted in bold, and the runner-up is underlined. The Avg. Reduce of MSE are computed by the average (across 3 datasets) of the differences between our model's MSE and each model's MSE, divided by the average (across 3 datasets) of each model's MSE, respectively.

| Category | Models | DAVIS | | | KIBA | | | BindingDB_Kd | | | Avg. Reduce of MSE (%) |
|---|---|---|---|---|---|---|---|---|---|---|---|
| | | MSE | R2 | CI | MSE | R2 | CI | MSE | R2 | CI | |
| GNN | GraphDTA-GIN | $0.253 \pm 0.010$ | $0.623 \pm 0.015$ | $0.861 \pm 0.006$ | $0.255 \pm 0.007$ | $-1.840 \pm 0.779$ | $0.553 \pm 0.019$ | $0.563 \pm 0.038$ | $0.693 \pm 0.021$ | $0.842 \pm 0.007$ | 34.827% |
| | GraphCPI-GIN | $0.274 \pm 0.009$ | $0.593 \pm 0.013$ | $0.851 \pm 0.008$ | $1.681 \pm 0.946$ | $-17.724 \pm 10.533$ | $0.553 \pm 0.094$ | $0.557 \pm 0.017$ | $0.696 \pm 0.009$ | $0.847 \pm 0.003$ | 72.213% |
| | MGraphDTA | $0.232 \pm 0.012$ | $0.655 \pm 0.018$ | $0.869 \pm 0.007$ | $0.032 \pm 0.012$ | $0.642 \pm 0.133$ | $0.832 \pm 0.040$ | $0.529 \pm 0.011$ | $0.712 \pm 0.006$ | $0.852 \pm 0.005$ | 11.980% |
| | SAGDTA | $0.324 \pm 0.064$ | $0.518 \pm 0.096$ | $0.833 \pm 0.027$ | $0.065 \pm 0.008$ | $0.279 \pm 0.085$ | $0.713 \pm 0.032$ | $0.529 \pm 0.011$ | $0.712 \pm 0.006$ | $0.852 \pm 0.005$ | 23.965% |
| | EmbedDTI | $0.280 \pm 0.024$ | $0.583 \pm 0.018$ | $0.851 \pm 0.009$ | $0.289 \pm 0.142$ | $-2.217 \pm 1.579$ | $0.558 \pm 0.038$ | $0.542 \pm 0.019$ | $0.705 \pm 0.010$ | $0.850 \pm 0.004$ | 37.174% |
| | DeepGLSTM | $0.316 \pm 0.023$ | $0.529 \pm 0.035$ | $0.841 \pm 0.007$ | $8.539 \pm 7.479$ | $-94.109 \pm 83.400$ | $0.514 \pm 0.036$ | $0.594 \pm 0.061$ | $0.677 \pm 0.033$ | $0.840 \pm 0.013$ | 92.613% |
| | CPI | $0.402 \pm 0.082$ | $0.401 \pm 0.122$ | $0.811 \pm 0.033$ | $0.052 \pm 0.003$ | $0.416 \pm 0.036$ | $0.734 \pm 0.037$ | $0.762 \pm 0.165$ | $0.585 \pm 0.090$ | $0.815 \pm 0.028$ | 42.599% |
| | BACPI | $0.334 \pm 0.015$ | $0.502 \pm 0.023$ | $0.827 \pm 0.006$ | $0.031 \pm 0.004$ | $0.658 \pm 0.043$ | $0.831 \pm 0.020$ | $0.550 \pm 0.010$ | $0.700 \pm 0.006$ | $0.845 \pm 0.002$ | 23.716% |
| | DeepNC-HGC | $0.309 \pm 0.025$ | $0.541 \pm 0.037$ | $0.841 \pm 0.005$ | $0.080 \pm 0.003$ | $0.110 \pm 0.036$ | $0.667 \pm 0.022$ | $0.572 \pm 0.011$ | $0.689 \pm 0.006$ | $0.844 \pm 0.003$ | 27.367% |
| | DeepNC-GEN | $0.270 \pm 0.012$ | $0.597 \pm 0.017$ | $0.852 \pm 0.009$ | $0.135 \pm 0.045$ | $-0.509 \pm 0.505$ | $0.608 \pm 0.037$ | $0.578 \pm 0.020$ | $0.685 \pm 0.011$ | $0.840 \pm 0.003$ | 28.993% |
| | DrugBAN | $0.242 \pm 0.007$ | $0.640 \pm 0.010$ | $0.869 \pm 0.003$ | $0.029 \pm 0.003$ | $0.676 \pm 0.032$ | $0.832 \pm 0.013$ | $\underline{0.465 \pm 0.018}$ | $\underline{0.747 \pm 0.010}$ | $0.862 \pm 0.003$ | 5.163% |
| | GANDTI | $0.318 \pm 0.018$ | $0.527 \pm 0.027$ | $0.844 \pm 0.006$ | $0.030 \pm 0.002$ | $0.662 \pm 0.026$ | $0.831 \pm 0.007$ | $0.621 \pm 0.012$ | $0.662 \pm 0.006$ | $0.836 \pm 0.002$ | 27.967% |
| | BridgeDPI | $1.241 \pm 1.432$ | $-0.848 \pm 2.133$ | $0.827 \pm 0.078$ | $0.325 \pm 0.109$ | $0.638 \pm 0.121$ | $0.857 \pm 0.001$ | $0.514 \pm 0.011$ | $0.720 \pm 0.006$ | $0.861 \pm 0.002$ | 66.442% |
| | ColdDTA | $\underline{0.220 \pm 0.009}$ | $\underline{0.672 \pm 0.014}$ | $\underline{0.880 \pm 0.004}$ | $0.110 \pm 0.029$ | $-0.224 \pm 0.329$ | $0.673 \pm 0.079$ | $0.463 \pm 0.008$ | $0.748 \pm 0.004$ | $\underline{0.866 \pm 0.001}$ | 11.980% |
| | SubMDTA | $0.289 \pm 0.012$ | $0.570 \pm 0.018$ | $0.841 \pm 0.007$ | $\underline{0.029 \pm 0.002}$ | $0.678 \pm 0.025$ | $\underline{0.836 \pm 0.011}$ | $0.532 \pm 0.032$ | $0.710 \pm 0.017$ | $0.852 \pm 0.006$ | 17.882% |
| | IMAEN | $0.230 \pm 0.009$ | $0.657 \pm 0.014$ | $0.874 \pm 0.004$ | $0.046 \pm 0.018$ | $0.484 \pm 0.196$ | $0.781 \pm 0.056$ | $0.479 \pm 0.012$ | $0.739 \pm 0.006$ | $0.863 \pm 0.002$ | 7.550% |
| Transformer | CSDTI | $0.331 \pm 0.012$ | $0.508 \pm 0.017$ | $0.832 \pm 0.005$ | $0.088 \pm 0.004$ | $0.014 \pm 0.041$ | $0.628 \pm 0.047$ | $0.768 \pm 0.021$ | $0.582 \pm 0.012$ | $0.805 \pm 0.004$ | 41.196% |
| | TDGraphDTA | $0.222 \pm 0.005$ | $0.669 \pm 0.008$ | $0.653 \pm 0.011$ | $0.091 \pm 0.019$ | $-0.009 \pm 0.209$ | $0.327 \pm 0.125$ | $0.497 \pm 0.016$ | $0.729 \pm 0.009$ | $0.777 \pm 0.005$ | 13.827% |
| | AMMVF | $0.377 \pm 0.030$ | $0.439 \pm 0.044$ | $0.815 \pm 0.005$ | — | — | — | — | — | — | — |
| | IIFDTI | $0.313 \pm 0.018$ | $0.534 \pm 0.027$ | $0.836 \pm 0.008$ | | | | $0.634 \pm 0.024$ | $0.655 \pm 0.013$ | $0.832 \pm 0.006$ | — |
| | ICAN | $0.371 \pm 0.013$ | $0.448 \pm 0.020$ | $0.818 \pm 0.006$ | $0.089 \pm 0.000$ | $-2.052 \pm 0.000$ | $0.500 \pm 0.000$ | $0.747 \pm 0.031$ | $0.593 \pm 0.017$ | $0.813 \pm 0.004$ | 42.171% |
| | MolTrans | $0.410 \pm 0.136$ | $0.390 \pm 0.202$ | $0.812 \pm 0.039$ | $4.314 \pm 2.290$ | $-47.055 \pm 25.515$ | $0.540 \pm 0.021$ | $0.695 \pm 0.183$ | $0.621 \pm 0.100$ | $0.822 \pm 0.009$ | 87.119% |
| | TransformerCPI | $0.393 \pm 0.022$ | $0.415 \pm 0.032$ | $0.802 \pm 0.008$ | $0.070 \pm 0.003$ | $0.217 \pm 0.033$ | $0.800 \pm 0.002$ | $0.659 \pm 0.040$ | $0.641 \pm 0.022$ | $0.829 \pm 0.013$ | 37.790% |
| | MRBDTA | $0.241 \pm 0.005$ | $0.640 \pm 0.008$ | $0.870 \pm 0.007$ | $0.050 \pm 0.005$ | $0.360 \pm 0.058$ | $0.735 \pm 0.015$ | $0.507 \pm 0.011$ | $0.724 \pm 0.003$ | $0.862 \pm 0.002$ | 12.531% |
| | FOTFCPI | $0.305 \pm 0.012$ | $0.546 \pm 0.018$ | $0.839 \pm 0.009$ | $0.229 \pm 0.180$ | $-1.555 \pm 2.003$ | $0.587 \pm 0.086$ | $0.567 \pm 0.008$ | $0.695 \pm 0.004$ | $0.848 \pm 0.006$ | 36.603% |
| | Our combos | $\mathbf{0.211 \pm 0.007}$ | $\mathbf{0.685 \pm 0.011}$ | $\mathbf{0.886 \pm 0.004}$ | $\mathbf{0.026 \pm 0.004}$ | $\mathbf{0.710 \pm 0.051}$ | $\mathbf{0.849 \pm 0.023}$ | $\mathbf{0.461 \pm 0.006}$ | $\mathbf{0.749 \pm 0.003}$ | $\mathbf{0.869 \pm 0.002}$ | 0.000% |

⋆**Benchmark over Efficiency** To analyze the training speed and memory usage, we empirically evaluate the peak memory and running time for various methods during the training procedure on one regression dataset and one classification task, respectively. To fairly compare various methods, we set the batch size as 32, as such maximum batch size is adopted by some methods. All results are measured on an RTX 3090 GPU. The memory and running time comparisons are illustrated in Fig. 4.

**Obs. 8. In general, the memory usage of GNN-based methods is smaller than that of Transformer-based methods, which is positively proportional to run time.** This difference is primarily due to the self-attention mechanism employed in Transformers, which requires significant memory resources. In contrast, model parameters, such as those in DeepGLSTM, do not exhibit a direct relationship with either runtime or performance.

⋆**Benchmark over Convergence.** We select the top two methods from the GNN-based and Transformer-based frameworks, respectively, and evaluate them across six datasets on two tasks. The training losses are

Table 2: Classification task benchmark on Human, *C.elegans*, and Drugbank datasets, respectively. Here, '−' means that the method cannot be reproduced on these datasets. For the GraphDTA and GraphCPI, we only show one which has the best performance. The best result is highlighted in bold, and the runner-up is underlined. The Avg. Improvement of Accuracy is computed by the average (across 3 datasets) of the differences between our model's accuracy and each model's accuracy, divided by the average (across 3 datasets) of each model's accuracy, respectively.

| Category | Models | Human | | | C.elegans | | | Drugbank | | | Avg. Improve |
|---|---|---|---|---|---|---|---|---|---|---|---|
| | | ROC-AUC | Accuracy | F1 | ROC-AUC | Accuracy | F1 | ROC-AUC | Accuracy | F1 | of Accuracy (%) |
| GNN | GraphDTA-GIN | 0.896 ± 0.001 | 0.897 ± 0.001 | 0.900 ± 0.001 | 0.947 ± 0.006 | 0.947 ± 0.006 | 0.947 ± 0.006 | 0.781 ± 0.004 | 0.781 ± 0.004 | 0.785 ± 0.003 | 3.124% |
| | GraphCPI-GIN | 0.898 ± 0.008 | 0.898 ± 0.008 | 0.901 ± 0.009 | 0.938 ± 0.008 | 0.938 ± 0.008 | 0.938 ± 0.008 | 0.783 ± 0.009 | 0.782 ± 0.009 | 0.784 ± 0.008 | 3.400% |
| | MGraphDTA | 0.939 ± 0.010 | 0.939 ± 0.010 | 0.941 ± 0.009 | 0.960 ± 0.004 | 0.959 ± 0.004 | 0.959 ± 0.004 | 0.802 ± 0.006 | 0.802 ± 0.006 | 0.807 ± 0.005 | 0.259% |
| | SAGDTA | 0.905 ± 0.005 | 0.905 ± 0.005 | 0.907 ± 0.005 | 0.936 ± 0.008 | 0.935 ± 0.008 | 0.935 ± 0.008 | 0.758 ± 0.008 | 0.758 ± 0.008 | 0.762 ± 0.005 | 4.196% |
| | EmbedDTI | - | - | - | - | - | - | 0.755 ± 0.004 | 0.755 ± 0.004 | 0.765 ± 0.003 | 4.551% |
| | DeepGLSTM | 0.918 ± 0.006 | 0.918 ± 0.006 | 0.921 ± 0.005 | 0.938 ± 0.006 | 0.938 ± 0.006 | 0.938 ± 0.006 | 0.746 ± 0.006 | 0.746 ± 0.006 | 0.751 ± 0.005 | 4.035% |
| | CPI | 0.911 ± 0.007 | 0.911 ± 0.007 | 0.914 ± 0.007 | 0.928 ± 0.007 | 0.928 ± 0.007 | 0.927 ± 0.007 | 0.678 ± 0.072 | 0.678 ± 0.072 | 0.687 ± 0.074 | 7.549% |
| | BACPI | 0.928 ± 0.007 | 0.928 ± 0.007 | 0.931 ± 0.007 | 0.952 ± 0.003 | 0.952 ± 0.004 | 0.951 ± 0.003 | 0.776 ± 0.009 | 0.776 ± 0.009 | 0.782 ± 0.008 | 1.920% |
| | DeepNC-HGC | 0.874 ± 0.005 | 0.874 ± 0.005 | 0.879 ± 0.005 | 0.934 ± 0.004 | 0.934 ± 0.004 | 0.934 ± 0.004 | 0.758 ± 0.008 | 0.758 ± 0.008 | 0.765 ± 0.007 | 5.495% |
| | DeepNC-GEN | 0.919 ± 0.004 | 0.919 ± 0.004 | 0.921 ± 0.005 | 0.941 ± 0.006 | 0.941 ± 0.006 | 0.941 ± 0.006 | 0.742 ± 0.008 | 0.742 ± 0.008 | 0.754 ± 0.006 | 4.035% |
| | DrugBAN | 0.935 ± 0.004 | 0.935 ± 0.004 | 0.938 ± 0.004 | 0.961 ± 0.003 | 0.961 ± 0.003 | 0.960 ± 0.003 | _0.804 ± 0.005_ | _0.804 ± 0.006_ | 0.806 ± 0.004 | 0.259% |
| | GANDTI | 0.933 ± 0.006 | 0.932 ± 0.006 | 0.934 ± 0.006 | 0.944 ± 0.005 | 0.943 ± 0.006 | 0.943 ± 0.005 | 0.752 ± 0.008 | 0.752 ± 0.008 | 0.763 ± 0.004 | 3.045% |
| | BridgeDPI | _0.948 ± 0.003_ | _0.948 ± 0.004_ | _0.949 ± 0.004_ | **0.969 ± 0.001** | **0.969 ± 0.001** | **0.968 ± 0.001** | 0.785 ± 0.007 | 0.785 ± 0.007 | 0.790 ± 0.003 | 0.185% |
| | ColdDTA | 0.937 ± 0.006 | 0.937 ± 0.006 | 0.938 ± 0.006 | 0.959 ± 0.004 | 0.959 ± 0.004 | 0.959 ± 0.004 | **0.815 ± 0.006** | **0.815 ± 0.006** | **0.818 ± 0.005** | −0.148% |
| | SubMDTA | 0.916 ± 0.014 | 0.916 ± 0.015 | 0.917 ± 0.015 | 0.921 ± 0.026 | 0.920 ± 0.026 | 0.921 ± 0.023 | 0.787 ± 0.007 | 0.787 ± 0.007 | 0.790 ± 0.005 | 3.202% |
| | IMAEN | 0.888 ± 0.009 | 0.888 ± 0.009 | 0.892 ± 0.011 | 0.904 ± 0.025 | 0.904 ± 0.025 | 0.902 ± 0.027 | 0.761 ± 0.026 | 0.761 ± 0.027 | 0.772 ± 0.013 | 6.032% |
| Transformer | CSDTI | 0.848 ± 0.009 | 0.848 ± 0.008 | 0.854 ± 0.007 | 0.873 ± 0.006 | 0.873 ± 0.006 | 0.874 ± 0.007 | 0.727 ± 0.006 | 0.727 ± 0.006 | 0.733 ± 0.004 | 10.580% |
| | TDGraphDTA | 0.940 ± 0.009 | 0.941 ± 0.008 | 0.943 ± 0.008 | 0.959 ± 0.004 | 0.959 ± 0.004 | 0.959 ± 0.004 | **0.815 ± 0.007** | **0.815 ± 0.006** | _0.815 ± 0.006_ | −0.295% |
| | AMMVF | 0.928 ± 0.005 | 0.929 ± 0.005 | 0.931 ± 0.006 | 0.962 ± 0.003 | 0.962 ± 0.003 | 0.961 ± 0.003 | 0.709 ± 0.023 | 0.709 ± 0.023 | 0.717 ± 0.021 | 4.115% |
| | IIFDTI | 0.938 ± 0.008 | 0.938 ± 0.007 | 0.940 ± 0.006 | 0.965 ± 0.004 | 0.965 ± 0.004 | _0.965 ± 0.004_ | 0.791 ± 0.010 | 0.791 ± 0.010 | 0.794 ± 0.012 | 0.483% |
| | ICAN | 0.938 ± 0.005 | 0.937 ± 0.005 | 0.939 ± 0.006 | 0.956 ± 0.005 | 0.956 ± 0.004 | 0.955 ± 0.005 | 0.764 ± 0.005 | 0.764 ± 0.005 | 0.768 ± 0.004 | 1.882% |
| | MolTrans | 0.937 ± 0.006 | 0.937 ± 0.006 | 0.939 ± 0.006 | 0.958 ± 0.004 | 0.958 ± 0.004 | 0.957 ± 0.004 | 0.796 ± 0.001 | 0.796 ± 0.001 | 0.799 ± 0.002 | 0.595% |
| | TransformerCPI | 0.920 ± 0.004 | 0.920 ± 0.004 | 0.923 ± 0.004 | 0.960 ± 0.005 | 0.960 ± 0.005 | 0.959 ± 0.004 | 0.797 ± 0.005 | 0.797 ± 0.005 | 0.802 ± 0.003 | 1.121% |
| | MRBDTA | 0.914 ± 0.009 | 0.914 ± 0.010 | 0.915 ± 0.011 | 0.847 ± 0.197 | 0.849 ± 0.194 | 0.745 ± 0.418 | 0.791 ± 0.009 | 0.791 ± 0.009 | 0.796 ± 0.006 | 5.991% |
| | FOTFCPI | 0.941 ± 0.004 | 0.941 ± 0.003 | 0.943 ± 0.003 | _0.966 ± 0.004_ | 0.965 ± 0.004 | _0.965 ± 0.004_ | 0.798 ± 0.008 | 0.798 ± 0.008 | 0.800 ± 0.007 | 0.111% |
| | Our combos | **0.950 ± 0.007** | **0.950 ± 0.007** | **0.959 ± 0.007** | _0.966 ± 0.001_ | _0.966 ± 0.001_ | _0.965 ± 0.001_ | 0.791 ± 0.006 | 0.791 ± 0.006 | 0.799 ± 0.004 | 0.000% |

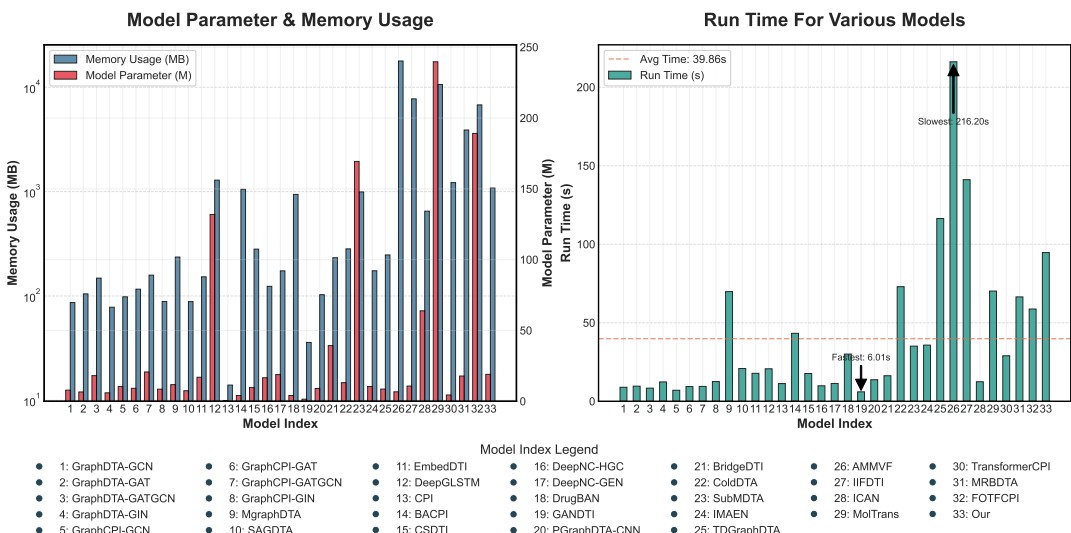

Figure 4: Model parameter, memory usage and running time comparisons on Davis dataset.

depicted in Fig. 5. In order to compare different methods, we only show the epochs before 300. Based on the empirical data, we summarize our primary observations as follows:

**Obs. 9. GNN-based methods demonstrate quicker and more stable convergence compared to Transformer-based methods.** This phenomenon arises from the fact that GNN-based methods have fewer memory usage and model parameters, leading to a larger batch size usage or faster convergence compared with Transformer-based methods.

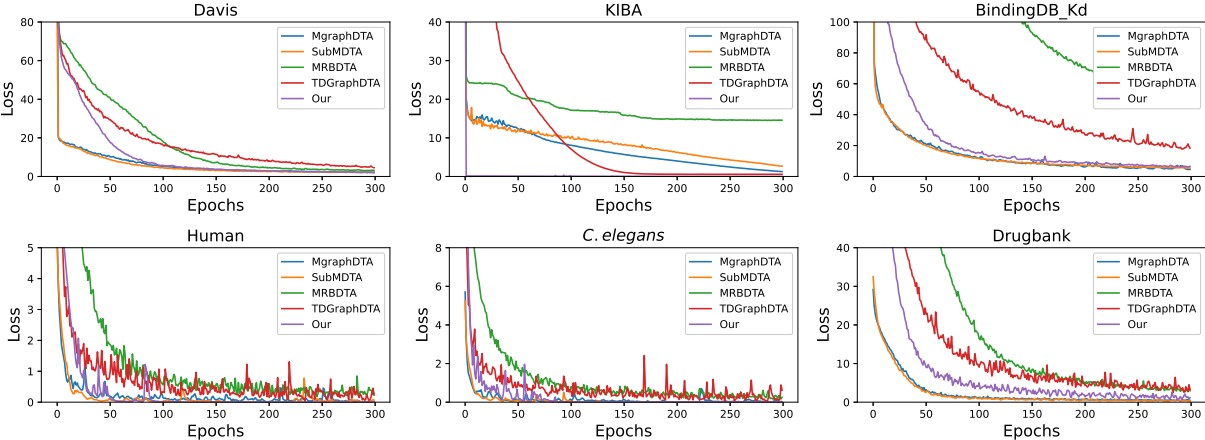

Figure 5: The empirical results of convergence for five selected methods.

## 5.1 Our Best Combo of Drug and Protein Encoders

Based on our benchmark results, we summarize the insights of protein and drug encoder usages and propose a light yet effective architecture, which could be treated as new strong baseline for the following explorations. ***Regarding the proteins***, we observe that multi-scale CNNs associated with a mixture of model depths can generally learn the effective protein representations (Yang et al., 2022; Zhu et al., 2023; Fang et al., 2023), which approximates the language model's accuracy while having lower memory and computation costs. ***Regarding the drug molecules***, both GNN and Transformer-based methods, such as MRBDTA (Zhang et al., 2022), MolTrans (Huang et al., 2020b) and MGraphDTA (Yang et al., 2022) prove promising in DTI tasks. This encourages us to leverage information from hybrid perspectives, i.e., implicit structure (via attention in Transformers) and explicit structure learning (via message passing along edges in GNNs).

We are thus motivated to integrate these powerful modules and shed novel insight into the design philosophy of drug-target interaction modeling. Our model combos are illustrated in Fig. 6, where the multi-scale CNNs and hybrid networks of molecular Transformer and GNNs are adopted to learn the representations of proteins and drugs, respectively. As shown by the graph encoder part in Fig. 6, the hybrid networks augment the differential attention matrix in molecular Transformer with inter-atomic distances and graph adjacency matrix (Maziarka et al., 2020), which provides the 3D and 2D molecule conformations to further facilitate atom interaction learning. Specifically, given the projections of molecular input at an attention head, i.e., $\mathbf{Q}, \mathbf{K}, \mathbf{V} \in \mathbb{R}^{N \times d}$, the adjacent matrix $\mathbf{A} \in \{0,1\}^{N \times N}$, and the inter-atomic distances matrix $\mathbf{D} \in \mathbb{R}^{N \times N}$ obtained using RDkit, the augmented attention is calculated as:

$$\text{Multi-Attn} = (\lambda_a \cdot \text{softmax}(\mathbf{Q}\mathbf{K}^{\text{T}}/\sqrt{d}) + \lambda_d g(\mathbf{D}) + \lambda_g \mathbf{A})\mathbf{V}, \tag{6}$$

where $g(\cdot)$ is a row-wise softmax function, and $\lambda_a, \lambda_d$ and $\lambda_g$ denote scalars weighting the self-attention, distance, and adjacency matrices, respectively. Besides the implicit and explicit structure learning, we integrate the features from drug SMILES. It is notable that simply utilizing the SMILES representation extracted from a transformer for downstream tasks does not perform as well as GNN. To align with the protein embedding paradigm, we adopt a simple CNN to unearth potential SMILES information, as suggested in Zhao et al. (2021). Subsequently, due to the fact that cross-attention is more complex and hard to optimize, we implement a straightforward attention mechanism to integrate the representations of the drug graph and SMILES, denoted as $\boldsymbol{f}_G$ and $\boldsymbol{f}_S$, respectively, using a weighting parameter $\lambda$, as follows:

$$\boldsymbol{f}_D = \lambda \cdot \boldsymbol{f}_G + (1 - \lambda) \cdot \boldsymbol{f}_S, \ \lambda = \text{MLP}\left(\text{MLP}(\boldsymbol{f}_G) + \text{MLP}(\boldsymbol{f}_S)\right). \tag{7}$$

Finally, the prediction is obtained by processing the concatenated protein and drug representations through a task-relevant head, as shown in Fig. 6

**Novelty of Combos.** While many different encoders and featurization methods has been proposed in the area of drug-target interaction predictionm, it is still unclear about their disentangled contribution

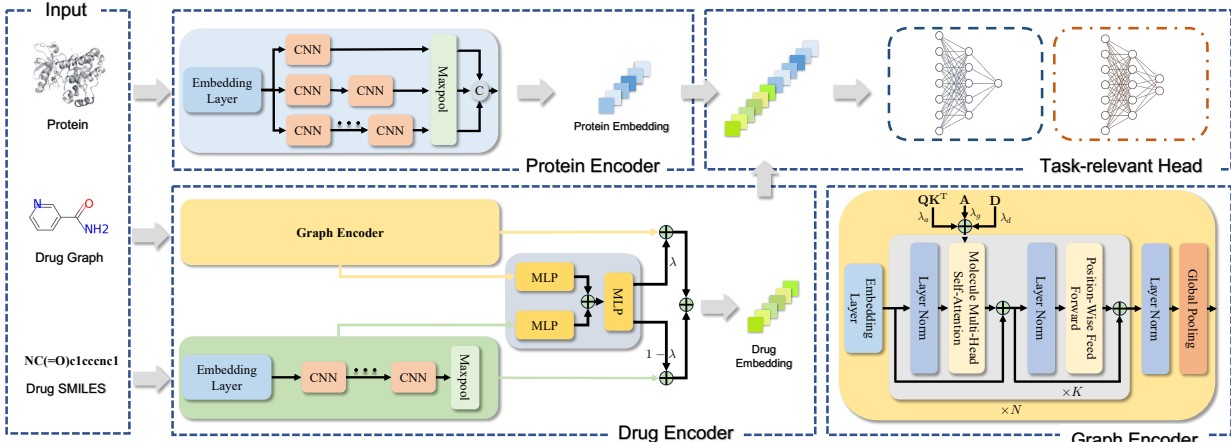

Figure 6: Overview of our proposed model combos.

due to the lack of fair benchmark platform. Without isolating the effects of diverse techniques, one might never reach convincing answers whether the drug-target interaction modeling with ceteris paribus should perform better. Additionally, our microscopic exploration reveals that more features do not always lead to better performance in GNN-based drug modeling. For example, atomic properties and hydrogen information positively contribute to predictive accuracy, while features such as electron properties surprisingly degrade DTI model performance. These findings provide valuable insights for future research, guiding more informed decisions on feature selection and improving molecular representation strategies in drug-target interaction modeling. Based on the benchmark analysis of both effectiveness and efficiency, we are thus motivated to summarize the powerful trick combos and shed novel insight into the design philosophy of the interaction prediction model. As shown in the paper, these combos achieve a better accuracy-efficiency trade-off in the commonly used datasets. It is expected that this design philosophy facilitates the model development for data scientists based on their downstream applications on hand.

**Benchmark Comparison to State-of-the-Art Frameworks.** We compare the proposed combos with the SOTA frameworks in Tables 1 and 2, and Figures 4 and 5. It is observed that our model consistently achieves the best performance in the regression tasks across three datasets and nearly outperforms most methods in classification tasks. By leveraging the physical conformation information from the molecular graph, our combos converge faster than the other two Transformer-based methods, MRBDTA (Zhang et al., 2022) and TDGraphDTA (Zhu et al., 2023), particularly on the KIBA dataset. Moreover, our model uses three times less peak memory and fewer parameters than other Transformer-based methods, enabling faster computation and reduced storage requirements

## 6 Conclusion

In this work, we establish a benchmark with fair and consistent experimental configurations, aiming to push DTI research, particularly emphasizing the utilization of structural information. Our meticulous approach has entailed thorough exploration of diverse encoder strategies and featurization techniques for both drug molecules and proteins. Moreover, dozens of existing approaches across six representative datasets for both regression and classification tasks are investigated on various metrics, including DTI classification and regression accuracy, peak memory usage, and model convergence. Provided with the comprehensive benchmark results, we propose a novel approach that integrates the strengths of GNN and transformer-based methods. Our studies on benchmarking and rethinking help lay a solid, practical, and systematic foundation for the DTI community and provide researchers with broader and deeper insights into the intricate dynamics of drug-target interactions.

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

# A   Related Works

**GNN-based Methods** GNNs play a crucial role in mining the intricate features of drug molecules for drug-target prediction. Numerous models, including Graph Convolutional Network (GCN), Graph Isomorphism Network (GIN), and Graph Attention Network (GAT) have been utilized (Nguyen et al., 2020; Quan et al., 2019; Wang et al., 2023; Lin et al., 2020;?; Jin et al., 2021) to process and enhance drug features. Additionally, MGraphDTA (Yang et al., 2022) employs a multi-scale GNN architecture, while DeepGLSTM (Mukherjee et al., 2022) leverages parallel GNN structures for drug representation. DeepNC integrates advanced techniques from generalized aggregation networks (Li et al., 2020) and hypergraph convolution (Bai et al., 2021) to improve feature extraction. BACPI (Li et al., 2022) develops a bi-directional attention network to integrate the representations of drug molecules and proteins, enhancing their mutual interaction. Besides, BridgeDPI (Wu et al., 2022) innovates by incorporating bridging nodes between proteins and drugs, utilizing a three-layer GNN for graph embeddings.

**Transformer-based Methods** Transformers, known for their efficacy in handling sequence data, are extensively applied in drug and protein feature processing. For instance, models like MolTrans (Huang et al., 2020b) and FOTFCPI (Yin et al., 2024) employ self-attention mechanisms to refine embeddings by focusing on drug and protein substructures. MRBDTA (Zhang et al., 2022) uses multi-head attention and skip connection to enhance drug and protein representation. Additionally, a cross-attention mechanism (Pan et al., 2023b; Kurata & Tsukiyama, 2022) is employed to facilitate the integration of drug and protein features, enabling effective mutual querying. TDGraphDTA (Zhu et al., 2023) captures contextual relationships between molecular substructures by using a multi-head cross-attention mechanism and graph optimization. Lastly, DrugormerDTI (Hu et al., 2023) incorporates degree centrality with positional information to highlight the positional relevance of amino acids in proteins.

**Input and Featurization** Structural information is crucial at the input stage for models such as BridgeDPI (Wu et al., 2022). Various libraries, such as DGLGraph (Wang et al., 2019), DGL-lifeSci (Li et al., 2021), and RDKit (Landrum et al., 2006), are employed to process input SMILES of drugs, with RDKit (Landrum et al., 2006) being pivotal for converting SMILE strings into molecular graphs and extracting diverse chemical properties, including chemical bonds, hydrogen presence, electron properties, and so on. Additionally, some approaches (Wang et al., 2023; Lin et al., 2020; Li et al., 2022; Wang et al., 2021) incorporate molecular fingerprints (Rogers & Hahn, 2010) to capture local chemical information. For protein sequences, typical pre-processing involves converting amino acid sequences into N-grams (Pan et al., 2023a; Dong et al., 2005) or integers (Nguyen et al., 2020) sequences. To enhance the expressiveness of embeddings, some models leverage pre-trained Word2Vec (Mikolov et al., 2013; Quan et al., 2019; Wang et al., 2023; Li et al., 2022; Tsubaki et al., 2018; Lin et al., 2020; Cheng et al., 2022) or pretrained protein language models (Bal et al., 2024).

# B    Model Descriptions

This section provides a comprehensive overview of 31 DTI methods, which are classified into GNN-based and Transformer-based approaches. The DTI framework can be simplified as using two encoders to process drugs and proteins separately, followed by an MLP to handle the integrated representations.

## B.1    GNN-based Methods

### B.1.1    GCN

⋆ *GraphDTA-GCN* (Nguyen et al., 2020): GraphDTA-GCN uses GCN to process the molecular graph, which is derived from SMILES using the RDkit tool, and a simple CNN with integer encoding to handle protein sequences.

⋆ *GraphCPI-GCN* (Quan et al., 2019): Similar to GraphDTA, GraphCPI-GCN employs 3-gram encoding with pretrained Word2Vec to process protein sequences, followed by a CNN to handle the protein embeddings.

⋆ *MGraphDTA* (Yang et al., 2022): MGraphDTA utilizes a multiscale GCN, inspired by dense connections, and a multiscale CNN to process drug graphs and protein sequences, respectively.

⋆ *SAGDTA* (Zhang et al., 2021): Similar to GraphDTA, SAGDTA introduces global or hierarchical pooling after GCN to aggregate node representations weightedly.

⋆ *EmbedDTI* (Jin et al., 2021): For protein sequences, EmbedDTI leverages GloVe for pretraining amino acid feature embeddings, which are then fed into a CNN. For drugs, it constructs both an atom graph and a substructure graph to capture structural information at different levels, processed by GCN.

⋆ *DeepGLSTM* (Mukherjee et al., 2022): DeepGLSTM processes molecular graphs using a parallel GCN module composed of three GCNs with different layers. For protein sequences, it adopts a bi-LSTM.

⋆ *CPI* (Tsubaki et al., 2018): CPI processes drug graphs using GCN. The protein sequence is handled via n-gram with integer encoding, followed by a CNN.

⋆ *DeepNC* (Tran et al., 2022): DeepNC adopts advanced techniques from generalized aggregation networks and hypergraph convolution, two variants of GCN, to capture the representations of drug. For protein sequences, it uses a simple CNN.

⋆ *DrugBAN* (Zhang et al., 2022): DrugBAN employs GCN and CNN blocks to encode molecular graph and proteins, respectively. Then they use a bilinear attention network module to learn local interactions between the representations of drugs and proteins.

⋆ *BridgeDPI* (Wu et al., 2022): BridgeDPI innovates by constructing a learnable drugprotein association network, which is processed using a three-layer GNN for graph embeddings. The learned representations for drug and protein pairs are then concatenated for further processing.

⋆ *ColdDTA* (Fang et al., 2023): ColdDTA removes the subgraphs of drugs. For the model, they adopt the dense GCN and multiscale CNN from MGraphDTA as the encoders for drugs and proteins, respectively. Additionally, an attention-based method is developed to integrate representations for improved prediction.

⋆ *IMAEN* (Zhang et al., 2024): IMAEN employs a molecular augmentation mechanism to enhance molecular structures by fully aggregating molecular node neighborhood information. It then uses multiscale GCN and CNN for drug and protein processing, respectively.

⋆ *GanDTI* (Wang et al., 2021): Inspired by residual networks, GanDTI add the input drug fingerprints to the output of three GCN layers as graph node features and use summation to get the final drug representation.

### B.1.2    GAT

⋆ *GraphDTA-GAT* (Nguyen et al., 2020): GraphDTA-GAT adopts a GAT as the encoder for drugs, while other components remain the same as in GraphDTA-GCN.

⋆ *GraphDTA-GATGCN* (Nguyen et al., 2020): GraphDTA-GATGCN adopts a combination of GAT and GCN as the encoder for drugs, while other components remain the same as in GraphDTA-GCN.

⋆ *GraphCPI-GAT* (Quan et al., 2019): GraphDTA-CPI adopts a GAT as the encoder for drugs, while other components remain the same as in GraphCPI-GCN.

⋆ *GraphCPI-GATGCN* (Quan et al., 2019): GraphCPI-GATGCN adopts a combination of GAT and GCN as the encoder for drugs, while other components remain the same as in GraphCPI-GCN.

⋆ *BACPI* (Li et al., 2022): BACPI adopts a GAT and a CNN for the features of the fingerprints and protein sequence, respectively. These features are then fed into a bi-directional attention neural network to obtain integrated representations.

⋆ *PGraphDTA-CNN* (Bal et al., 2024): PGraphDTA-CNN is a straightforward method that utilizes GAT for drug feature extraction and CNN for protein sequences.

## B.2 GIN

⋆ *GraphDTA-GIN* (Nguyen et al., 2020): GraphDTA-GAT adopts a GAT as the encoder for drugs, while other components remain the same as in GraphDTA-GCN.

⋆ *GraphCPI-GIN* (Quan et al., 2019): GraphDTA-GAT adopts a GAT as the encoder for drugs, while other components remain the same as in GraphDTA-GCN.

⋆ *SubMDTA* (Pan et al., 2023a): SubMDTA utilizes a pretrained GIN encoder obtained through contrastive learning for the molecular graph. For protein sequences, it employs N-gram embedding with different N to extract features at various scales, which are then processed by a BiLSTM.

## B.3 Transformer-based Methods

### B.3.1 Self-attention

⋆ *AMMVF* (Wang et al., 2023): AWMVF introduces the multi-head mechanism to GAT to learn features in different spaces, and the update function is obtained through the concatenation of different heads' outputs.

⋆ *IIFDTI* (Cheng et al., 2022): IIFDTI model attains the drug matrix and protein matrix and inputs them to the bi-directional encoder-decoder block, which considers both the drug and target directions. The decoder is mainly composed of multi-head attention.

⋆ *MolTrans* (Huang et al., 2020b): MolTrans uses transformer encoder layers to augment the embedding of sub-structure sequences of proteins and drugs.

⋆ *FOTFCPI* (Yin et al., 2024): Similar to MolTrans, FOTFCPI uses transformer encoder layers to extract the features of protein and drug fragments after the embedding layers.

⋆ *TransformerCPI* (Chen et al., 2020): TransformerCPI uses the decoder module of Transformer, which takes in the atom sequence embedding processed by GCN and the protein sequence embedding processed by word2vec and 1D CNN.

⋆ *MRBDTA* (Zhang et al., 2022): In MRBDTA, after the embedding layer, drug sequences are directly fed into a block consisting of three Transformer encoders. The first encoder has a linear layer before it and the following two encoders are parallel. The protein sequence is also processed by a block with similar structure.

### B.3.2 Cross attention

⋆ *CSDTI*(Pan et al., 2023b): CSDTI use cross attention to fuse the deep representations of drugs and proteins. Specifically, the different projections of protein feature are used as key and value respectively while the projection of drug feature is used as query.

⋆ *TDGraphDTA*(Zhu et al., 2023): TDGraphDTA use a multi-head cross-attention mechanism with two attention heads. Both drug and protein features are linearly transformed into query, key and value matrices.

One cross attention layer uses a drug query matrix, a protein key matrix, and a protein value matrix, while its parallel counterparts use the rest of the matrices. The outputs of these two layers are concatenated and fed into MLP to get the final output.

## C   Datasets Descriptions

In this subsection, we provide a detailed description of the datasets for both the regression task and classification task. The statistical characteristics of the datasets are summarized in Table 3.

Table 3: Statistics of the benchmark dataset for two tasks.

|  | Regression | | | Classification | | |
| --- | --- | --- | --- | --- | --- | --- |
|  | Davis | KIBA | BindingDB_Kd | Human | *C. elegans* | DrugBank |
| Number of drugs | 68 | 2068 | 10661 | 2726 | 1767 | 6645 |
| Number of target proteins | 379 | 229 | 1413 | 2001 | 1876 | 4256 |
| Number of total samples | 25772 | 117657 | 52274 | 6728 | 7786 | 35021 |

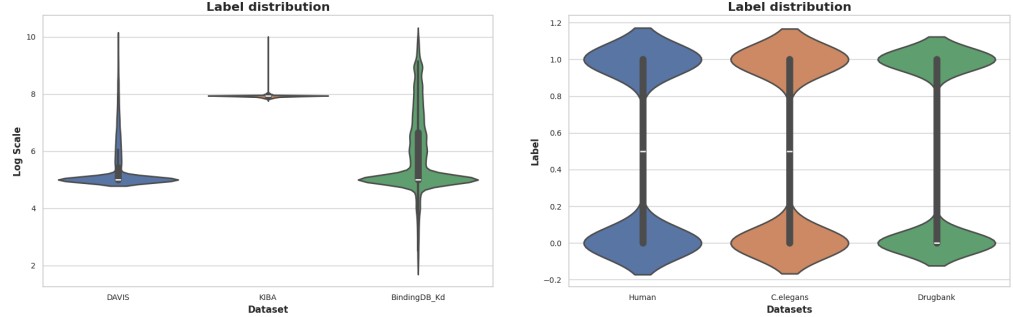

(a) Label distribution of DVAIS, KIBA and Bind-ing_Kd for regression tasks.

(b) Label distribution of Human, *C. elegans* and Drugbank for classification tasks.

Figure 7: Label distribution of different datasets for two tasks.

# D    Evaluation Metrics

We adopt distinct sets of metrics to evaluate the classification and regression tasks. In particular, considering the classification task, we utilize the common metrics including Area Under Receiver Operating Characteristic Curve (ROC-AUC), Precision-Recall Area Under Curve (PR-AUC), LogAUC, accuracy, precision, recall, and F1 score. For the continuous binding affinity regression, we benchmark the models using metrics of mean squared error (MSE), mean absolute error (MAE), coefficient of determination (R2), Pearson correlation coefficient, Concordance Index (CI), and Spearman correlation coefficient. Each of these metrics offers unique insights into different aspects of model performance, allowing us to assess predictive accuracy, correlation with observed values, and consistency in ranking predictions.

# E    Original Hyperparameter

To have a basic understanding of hyperparameters before greedy search and to find the optimized setting for each model, we summarize the hyperparameters reported in the corresponding paper or codes in Table 4.

Table 4: Configurations of basic hyperparameters adopted to implement different approaches.

| Category | Models | Batch size | Total epoch | Learning rate & Decay & Decay epoch | Weight decay | Dropout | Optimizer |
|---|---|---|---|---|---|---|---|
| GNN | GraphDTA-GCN (Nguyen et al., 2020) | 512 | 1000 | 0.0005 | - | 0.2 | Adam |
| | GraphDTA-GAT (Nguyen et al., 2020) | 512 | 1000 | 0.0005 | - | 0.2 | Adam |
| | GraphDTA-GATGCN (Nguyen et al., 2020) | 512 | 1000 | 0.0005 | - | 0.2 | Adam |
| | GraphDTA-GIN (Nguyen et al., 2020) | 512 | 1000 | 0.0005 | - | 0.2 | Adam |
| | GraphCPI-GCN (Quan et al., 2019) | 512 | 1000 | 0.0005 | - | 0.5 | Adam |
| | GraphCPI-GAT (Quan et al., 2019) | 512 | 1000 | 0.0005 | - | 0.6 | Adam |
| | GraphCPI-GATGCN (Quan et al., 2019) | 512 | 1000 | 0.0005 | - | - | Adam |
| | GraphCPI-GIN (Quan et al., 2019) | 512 | 1000 | 0.0005 | - | 0.6 | Adam |
| | MGraphDTA (Yang et al., 2022) | 512 | 3000 | 0.0005 | - | 0.1 | Adam |
| | SAGDTA (Zhang et al., 2021) | 512 | 2000 | 0.001 | - | 0.1 | Adam |
| | EmbedDTI (Jin et al., 2021) | 512 | 1500 | 0.0005 | - | 0.2 | Adam |
| | DeepGLSTM (Mukherjee et al., 2022) | 512/128 | 1000 | 0.0005 | - | 0.2 | Adam |
| | CPI (Tsubaki et al., 2018) | 1 | 100 | 0.001, 0.5, 10 | 1e-6 | 0 | Adam |
| | BACPI (Li et al., 2022) | 16 | 20 | 0.0005, 0.5, 10 | - | 0.1 | Adam |
| | DeepNC-HGC (Tran et al., 2022) | 256 | 1000 | 0.0005 | - | 0.2 | Adam |
| | DeepNC-GEN (Tran et al., 2022) | 256 | 1000 | 0.0005 | - | 0.2 | Adam |
| | DrugBAN (Bai et al., 2023) | 64 | 100 | 0.00005 | - | 0 | Adam |
| | GANDTI (Wang et al., 2021) | 1 | 30/15 | 0.001 | 1e-6 | 0.5 | Adam |
| | PGraphDTA-CNN (Bal et al., 2024) | 512 | 1500 | 0.0005 | - | 0.2 | Adam |
| | BridgeDPI (Wu et al., 2022) | 512 | 100 | 0.001 | - | 0.5 | Adam |
| | ColdDTA (Fang et al., 2023) | 128 | 700/300 | 0.0003 | - | 0 | Adam |
| | SubMDTA (Pan et al., 2023a) | 512 | 1200 | 0.0005 | - | 0.2 | Adam |
| | IMAEN (Zhang et al., 2024) | 128 | 1000 | 0.0005 | - | 0.2 | Adam |
| Transformer | CSDTI (Pan et al., 2023b) | 256 | 3000 | 0.0005 | - | 0.2 | Adam |
| | AMMVF (Wang et al., 2023) | 32 | 40 | 0.001, 0.5, 5 | 1e-4 | 0.1 | Adam |
| | TDGraphDTA (Zhu et al., 2023) | 1024 | 3000 | 0.0005 | - | 0.1 | Adam |
| | IIFDTI (Cheng et al., 2022) | 64 | 200 | 0.001 | 1e-6 | 0.2 | AdamW |
| | ICAN (Kurata & Tsukiyama, 2022) | 128 | 50 | 0.001 | - | 0.1 | Adam |
| | MolTrans (Huang et al., 2020b) | 64 | 30 | 0.00001 | - | 0.1 | Adam |
| | TransformerCPI (Chen et al., 2020) | 8 | 40 | 0.0001, 0.5, 5 | 1e-4 | 0.2 | RAdam |
| | MRBDTA (Zhang et al., 2022) | 1024/256 | 600/300 | 0.001 | - | 0.1 | Adam |
| | FOTFCPI (Yin et al., 2024) | 64 | 100 | 0.0001 | - | 0.1 | Adam |

## F   Greedy Search on Hyperparameter

Here we present the detailed greedy search experiment on hyperparameters with full metrics. The detailed results on Davis and Human datasets are in Table 5.

Table 5: The greedy hyperparameter searching results for two graph-based models and two transformer-based models on regression (DAVIS) and classification task (Human).

| Models | Hyper parameter | | | Regression | | | | | | Classification | | | | | | |
|---|---|---|---|---|---|---|---|---|---|---|---|---|---|---|---|---|
| | Batch size | Learning rate | Dropout | MSE | MAE | R2 | PCC | CI | Spearman | ROC-AUC | PR-AUC | Range-AUC | Acc. | Precision | Recall | F1 |
| GraphDTA | 512 | 0.0005 | 0.2 | 0.2838 | 0.3053 | 0.5773 | 0.7619 | 0.8508 | 0.6226 | 0.9117 | 0.8807 | 0.4712 | 0.9117 | 0.9157 | 0.9125 | 0.9141 |
| | 256 | 0.0005 | 0.2 | 0.2781 | 0.2924 | 0.5859 | 0.7677 | 0.8523 | 0.6250 | 0.9132 | 0.8805 | 0.4584 | 0.9135 | 0.9115 | 0.9218 | 0.9165 |
| | 128 | 0.0005 | 0.2 | 0.2634 | 0.2552 | 0.6077 | 0.7840 | 0.8700 | 0.6828 | 0.9134 | 0.8822 | 0.4716 | 0.9135 | 0.9157 | 0.9166 | 0.9160 |
| | 64 | 0.0005 | 0.2 | 0.2591 | 0.2506 | 0.6141 | 0.7885 | 0.8634 | 0.6862 | 0.9121 | 0.8793 | 0.4608 | 0.9123 | 0.9112 | 0.9195 | 0.9153 |
| | 512 | 0.0001 | 0.2 | 0.3209 | 0.3148 | 0.5221 | 0.7299 | 0.8369 | 0.6008 | 0.9165 | 0.8851 | 0.4775 | 0.9168 | 0.9159 | 0.9232 | 0.9195 |
| | 512 | 0.001 | 0.2 | 0.2909 | 0.3076 | 0.5668 | 0.7812 | 0.8425 | 0.6073 | 0.9133 | 0.8827 | 0.4765 | 0.9134 | 0.9172 | 0.9145 | 0.9157 |
| | 512 | 0.0005 | 0 | 0.2827 | 0.2923 | 0.5791 | 0.7632 | 0.8527 | 0.6267 | 0.9177 | 0.8863 | 0.4764 | 0.9180 | 0.9163 | 0.9252 | 0.9207 |
| | 512 | 0.0005 | 0.1 | 0.2771 | 0.2947 | 0.5873 | 0.7695 | 0.8521 | 0.6241 | 0.9170 | 0.8873 | 0.4830 | 0.9171 | 0.9204 | 0.9189 | 0.9194 |
| | 512 | 0.0005 | 0.5 | 0.2951 | 0.3165 | 0.5605 | 0.7494 | 0.8478 | 0.6176 | 0.9128 | 0.8819 | 0.4637 | 0.9129 | 0.9161 | 0.9151 | 0.9155 |
| GraphCPI | 512 | 0.0005 | 0.2 | 0.3173 | 0.3309 | 0.5275 | 0.7325 | 0.8267 | 0.5825 | 0.9049 | 0.8687 | 0.4275 | 0.9053 | 0.9005 | 0.9177 | 0.9089 |
| | 256 | 0.0005 | 0.2 | 0.2781 | 0.2924 | 0.5859 | 0.7677 | 0.8523 | 0.6250 | 0.9058 | 0.8712 | 0.4394 | 0.9061 | 0.9047 | 0.9140 | 0.9093 |
| | 128 | 0.0005 | 0.2 | 0.3064 | 0.2988 | 0.5436 | 0.7402 | 0.8467 | 0.6213 | 0.9085 | 0.8762 | 0.4602 | 0.9086 | 0.9115 | 0.9111 | 0.9112 |
| | 64 | 0.0005 | 0.2 | 0.3484 | 0.3569 | 0.4812 | 0.6944 | 0.8249 | 0.5850 | 0.9123 | 0.8774 | 0.4453 | 0.9128 | 0.9060 | 0.9235 | 0.9163 |
| | 512 | 0.0001 | 0.2 | 0.3461 | 0.3494 | 0.4846 | 0.7075 | 0.8317 | 0.5927 | 0.9063 | 0.8691 | 0.4217 | 0.9068 | 0.8985 | 0.9235 | 0.9107 |
| | 512 | 0.001 | 0.2 | 0.2870 | 0.2733 | 0.5726 | 0.7631 | 0.8598 | 0.6416 | 0.9064 | 0.8717 | 0.4438 | 0.9067 | 0.9048 | 0.9151 | 0.9099 |
| | 512 | 0.0005 | 0 | 0.3431 | 0.3527 | 0.4891 | 0.7096 | 0.8297 | 0.5890 | 0.9060 | 0.8709 | 0.4386 | 0.9064 | 0.9036 | 0.9160 | 0.9097 |
| | 512 | 0.0005 | 0.1 | 0.3291 | 0.3388 | 0.5100 | 0.7265 | 0.8294 | 0.5885 | 0.9060 | 0.8706 | 0.4385 | 0.9064 | 0.9029 | 0.9169 | 0.9098 |
| | 512 | 0.0005 | 0.5 | 0.3329 | 0.3264 | 0.5043 | 0.7304 | 0.8361 | 0.6211 | 0.9064 | 0.8702 | 0.4286 | 0.9068 | 0.9010 | 0.9203 | 0.9105 |
| MRBDTA | 128 | 0.001 | 0.1 | 0.2397 | 0.2648 | 0.6430 | 0.8025 | 0.8716 | 0.6546 | 0.9340 | 0.9119 | 0.5240 | 0.9338 | 0.9427 | 0.9281 | 0.9353 |
| | 64 | 0.001 | 0.1 | 0.2504 | 0.2679 | 0.6272 | 0.7935 | 0.8674 | 0.6483 | 0.9348 | 0.9113 | 0.5219 | 0.9348 | 0.9396 | 0.9336 | 0.9365 |
| | 32 | 0.001 | 0.1 | 0.2567 | 0.2721 | 0.6177 | 0.7878 | 0.8622 | 0.6400 | 0.9328 | 0.9127 | 0.5331 | 0.9324 | 0.9474 | 0.9200 | 0.9334 |
| | 128 | 0.0005 | 0.1 | 0.2350 | 0.2565 | 0.6499 | 0.8069 | 0.8775 | 0.6639 | 0.9382 | 0.9164 | 0.5304 | 0.9381 | 0.9442 | 0.9353 | 0.9397 |
| | 128 | 0.0001 | 0.1 | 0.2335 | 0.2696 | 0.6523 | 0.8084 | 0.8767 | 0.6630 | 0.9372 | 0.9171 | 0.5350 | 0.9370 | 0.9481 | 0.9284 | 0.9381 |
| | 128 | 0.001 | 0 | 0.2425 | 0.2663 | 0.6388 | 0.7997 | 0.8739 | 0.6587 | 0.9279 | 0.9010 | 0.4997 | 0.9280 | 0.9294 | 0.9316 | 0.9303 |
| | 128 | 0.001 | 0.2 | 0.2408 | 0.2620 | 0.6414 | 0.8027 | 0.8679 | 0.6478 | 0.9438 | 0.9218 | 0.5363 | 0.9438 | 0.9450 | 0.9460 | 0.9454 |
| | 128 | 0.001 | 0.5 | 0.2584 | 0.2666 | 0.6153 | 0.7894 | 0.8727 | 0.6573 | 0.9328 | 0.9123 | 0.5295 | 0.9325 | 0.9462 | 0.9215 | 0.9336 |
| TransformerCPI | 128 | 0.001 | 0.1 | 0.3747 | 0.4111 | 0.4420 | 0.6787 | 0.8164 | 0.5668 | 0.9294 | 0.9024 | 0.5047 | 0.9296 | 0.9301 | 0.9339 | 0.9317 |
| | 64 | 0.001 | 0.1 | 0.5916 | 0.5422 | 0.1191 | 0.3531 | 0.6644 | 0.3014 | 0.9318 | 0.9061 | 0.5100 | 0.9319 | 0.9337 | 0.9342 | 0.9339 |
| | 32 | 0.001 | 0.1 | 0.6219 | 0.5197 | 0.0740 | 0.2993 | 0.6360 | 0.2506 | 0.9269 | 0.8959 | 0.4860 | 0.9273 | 0.9198 | 0.9411 | 0.9302 |
| | 128 | 0.0005 | 0.1 | 0.2869 | 0.3389 | 0.5728 | 0.7750 | 0.8326 | 0.5910 | 0.9270 | 0.9008 | 0.5030 | 0.9270 | 0.9312 | 0.9270 | 0.9290 |
| | 128 | 0.0001 | 0.1 | 0.2877 | 0.3411 | 0.5716 | 0.7674 | 0.8348 | 0.5950 | 0.9208 | 0.8898 | 0.4664 | 0.9211 | 0.9181 | 0.9307 | 0.9239 |
| | 128 | 0.001 | 0 | 0.3874 | 0.3948 | 0.4232 | 0.6481 | 0.7985 | 0.5358 | 0.9297 | 0.9015 | 0.4999 | 0.9300 | 0.9272 | 0.9376 | 0.9324 |
| | 128 | 0.001 | 0.2 | 0.5181 | 0.5219 | 0.2284 | 0.5249 | 0.7476 | 0.4494 | 0.9236 | 0.8938 | 0.4780 | 0.9239 | 0.9217 | 0.9319 | 0.9265 |
| | 128 | 0.001 | 0.5 | 1.5318 | 1.1598 | -1.2809 | 0.2866 | 0.6441 | 0.2647 | 0.9218 | 0.8895 | 0.4746 | 0.9223 | 0.9153 | 0.9359 | 0.9253 |

## G  Comparison of different featurization

In this section, we present the summarized featurization methods in Table 6, the detailed description of all properties is shown in Table 7. Besides, an ablation study on featurization strategies is in Table 8.

Table 6: Summary of the featurization of GNN-based model. Mol. Graphs means Molecular graphs, and both means using molecular graphs and fingerprints.

| | | Atom Type | Degree | Implicit Valence | Explicit Valence | Hybridization | Aromaticity | Formal Charge | # Atom | # Hs | # Explicit Hs | # Implicit Hs | # Radical Electrons | Electron Affinity | CIP | Chirality | Ring |
|---|---|---|---|---|---|---|---|---|---|---|---|---|---|---|---|---|---|
| **Models** | **Graph** | | | | | | | | | | | | | | | | |
| GraphDTA | Mol. Graphs | ✓ | ✓ | ✓ | | | ✓ | | | ✓ | | | | | | | |
| GraphCPI | Mol. Graphs | ✓ | ✓ | ✓ | | | ✓ | | | ✓ | | | | | | | |
| MGraphDTA | Mol. Graphs | ✓ | ✓ | ✓ | ✓ | ✓ | ✓ | ✓ | ✓ | ✓ | ✓ | | ✓ | | | ✓ | |
| SAGDTA | Mol. Graphs | ✓ | ✓ | ✓ | | | ✓ | | | ✓ | | | | | | | |
| EmbedDTI | Mol. Graphs | ✓ | ✓ | ✓ | ✓ | ✓ | ✓ | ✓ | | ✓ | | | | | | | ✓ |
| DeepGLSTM | Mol. Graphs | ✓ | ✓ | ✓ | | | ✓ | | | ✓ | | | | | | | |
| CPI | Fingerprints | ✓ | | | | | ✓ | | | | | | | | | | |
| BACPI | Fingerprints | ✓ | | | | | ✓ | | | | | | | | | | |
| DeepNC | Mol. Graphs | ✓ | ✓ | ✓ | | | ✓ | | | ✓ | | | | | | | |
| DrugBAN | Mol. Graphs | ✓ | | | | ✓ | ✓ | ✓ | | ✓ | | ✓ | ✓ | | | | ✓ |
| GANDTI | Fingerprints | ✓ | | | | | ✓ | | | | | | | | | | |
| PGraphDTA-CNN | Mol. Graphs | ✓ | ✓ | ✓ | | ✓ | ✓ | ✓ | | ✓ | | ✓ | ✓ | | | | ✓ |
| BridgeDPI | Mol. Graphs | ✓ | ✓ | ✓ | | ✓ | ✓ | ✓ | | | | | | | | | |
| ColdDTA | Mol. Graphs | ✓ | ✓ | ✓ | | ✓ | ✓ | | | | | | | | ✓ | ✓ | |
| SubMDTA | Mol. Graphs | ✓ | ✓ | ✓ | | | ✓ | | | ✓ | | | | | | | |
| IMAEN | Mol. Graphs | ✓ | ✓ | ✓ | | | ✓ | | | ✓ | | | | | | | |
| CSDTI | Mol. Graphs | ✓ | ✓ | ✓ | ✓ | ✓ | ✓ | ✓ | ✓ | ✓ | ✓ | | ✓ | | | ✓ | |
| TDGraphDTA | Mol. Graphs | ✓ | ✓ | | | | ✓ | | | ✓ | | | ✓ | | ✓ | | |
| AMMVF | Both | ✓ | ✓ | | | ✓ | ✓ | | | ✓ | | | ✓ | | ✓ | | |
| TransformerCPI | Mol. Graphs | ✓ | ✓ | | | ✓ | ✓ | ✓ | | ✓ | | | ✓ | | ✓ | ✓ | |

Model Information / Atomic Properties / Hydrogen Information / Electron Properties / Stereochemistry / Structure

Table 7: Description of atomic and molecular properties for node featurization

| Name | Description |
|---|---|
| **Atomic Properties** | |
| Atom Type | Type of the atom (e.g., C, N, O, H) |
| Degree | Number of directly bonded neighbors |
| Implicit Valence | Number of implicit valence of the atom |
| Explicit Valence | Number of explicit valence of the atom |
| Hybridization | The state of hybridization (e.g., sp3, sp2) |
| Aromaticity | Whether the atom is part of an aromatic system |
| Formal Charge | The charge assigned to an atom |
| # Atom | Total number of atoms |
| **Hydrogen Information** | |
| # Hs | Total number of hydrogens |
| # Explicit Hs | Number of explicit hydrogens on the atom |
| # Implicit Hs | Number of implicit hydrogens on the atom |
| **Electron Properties** | |
| # Radical Electrons | Number of radical electrons |
| Electron Affinity | Tendency of an atom to accept electrons |
| **Stereochemistry** | |
| CIP | The CIP code (R or S) of the atom |
| Chirality | If an atom is a possible chiral center |
| **Structure** | |
| Ring | Whether the atom is part of a ring structure |

Table 8: Extra Graph embedding feature exploration. Here Basic: {Atom Type, Degree, Implicit Valence, Aromaticity, # Hs}

| Models | Intial Feature | Regression | | | | | | Classification | | | | | | |
|---|---|---|---|---|---|---|---|---|---|---|---|---|---|---|
| | | MSE | MAE | R2 | PCC | CI | Spearman | ROC-AUC | PR-AUC | Range-AUC | Acc. | Precision | Recall | F1 |
| GraphDTA | Basic | 0.2771 | 0.2947 | 0.5873 | 0.7695 | 0.8521 | 0.6241 | 0.9170 | 0.8873 | 0.4830 | 0.9171 | 0.9204 | 0.9189 | 0.9194 |
| | Basic+AP | 0.2772 | 0.2978 | 0.5873 | 0.7671 | 0.8496 | 0.6200 | 0.9153 | 0.8817 | 0.4517 | 0.9157 | 0.9099 | 0.9290 | 0.9191 |
| | Basic+HI | 0.2783 | 0.2983 | 0.5855 | 0.7663 | 0.8483 | 0.6185 | 0.9211 | 0.8903 | 0.4815 | 0.9214 | 0.9188 | 0.9296 | 0.9241 |
| | Basic+EP | 0.2775 | 0.3068 | 0.5868 | 0.7682 | 0.8499 | 0.6205 | 0.9165 | 0.8862 | 0.4795 | 0.9166 | 0.9185 | 0.9198 | 0.9191 |
| | Basic+Ste | 0.2838 | 0.3030 | 0.5773 | 0.7624 | 0.8523 | 0.6254 | 0.9200 | 0.8905 | 0.4869 | 0.9200 | 0.9216 | 0.9235 | 0.9224 |
| | Basic+Str | 0.2783 | 0.2991 | 0.5857 | 0.7668 | 0.8505 | 0.6228 | 0.9198 | 0.8865 | 0.4649 | 0.9202 | 0.9124 | 0.9351 | 0.9235 |
| | Basic+AP+HI | 0.2851 | 0.3029 | 0.5755 | 0.7610 | 0.8504 | 0.6222 | 0.9163 | 0.8822 | 0.4629 | 0.9168 | 0.9094 | 0.9313 | 0.9201 |
| | Basic+AP+HI+EP | 0.2845 | 0.2917 | 0.5763 | 0.7620 | 0.8510 | 0.6227 | 0.9140 | 0.8811 | 0.4580 | 0.9143 | 0.9115 | 0.9232 | 0.9173 |
| | Basic+AP+HI+EP+Ste | 0.2811 | 0.3099 | 0.5814 | 0.7640 | 0.8500 | 0.6212 | 0.9192 | 0.8899 | 0.4853 | 0.9193 | 0.9218 | 0.9215 | 0.9216 |
| | Basic+AP+HI+EP+Ste+Str | 0.2801 | 0.2916 | 0.5829 | 0.7659 | 0.8538 | 0.6278 | 0.9217 | 0.8905 | 0.4794 | 0.9220 | 0.9180 | 0.9319 | 0.9248 |
| GraphCPI | Basic | 0.3291 | 0.3388 | 0.5100 | 0.7265 | 0.8294 | 0.5885 | 0.9060 | 0.8706 | 0.4385 | 0.9064 | 0.9029 | 0.9169 | 0.9098 |
| | Basic+AP | 0.3331 | 0.3389 | 0.5040 | 0.7198 | 0.8223 | 0.5761 | 0.9038 | 0.8657 | 0.4103 | 0.9043 | 0.8955 | 0.9218 | 0.9084 |
| | Basic+HI | 0.3402 | 0.3457 | 0.4934 | 0.7157 | 0.8228 | 0.5769 | 0.9051 | 0.8713 | 0.4495 | 0.9052 | 0.9058 | 0.9094 | 0.9080 |
| | Basic+EP | 0.3408 | 0.3505 | 0.4926 | 0.7123 | 0.8211 | 0.5749 | 0.9053 | 0.8713 | 0.4442 | 0.9055 | 0.9060 | 0.9111 | 0.9085 |
| | Basic+Ste | 0.3398 | 0.3634 | 0.4940 | 0.7119 | 0.8274 | 0.5855 | 0.9061 | 0.8692 | 0.4261 | 0.9065 | 0.8992 | 0.9221 | 0.9104 |
| | Basic+Str | 0.3419 | 0.3562 | 0.4909 | 0.7113 | 0.8226 | 0.5766 | 0.9066 | 0.8683 | 0.4079 | 0.9073 | 0.8957 | 0.9281 | 0.9115 |
| | Basic+AP+HI | 0.3326 | 0.3471 | 0.5048 | 0.7212 | 0.8210 | 0.5734 | 0.9010 | 0.8659 | 0.4288 | 0.9012 | 0.9018 | 0.9071 | 0.9043 |
| | Basic+AP+HI+EP | 0.3404 | 0.3476 | 0.4931 | 0.7150 | 0.8212 | 0.5748 | 0.9015 | 0.8612 | 0.3821 | 0.9022 | 0.8890 | 0.9258 | 0.9070 |
| | Basic+AP+HI+EP+Ste | 0.3403 | 0.3445 | 0.4932 | 0.7111 | 0.8169 | 0.5671 | 0.9109 | 0.8763 | 0.4511 | 0.9113 | 0.9065 | 0.9229 | 0.9146 |
| | Basic+AP+HI+EP+Ste+Str | 0.3469 | 0.3550 | 0.4834 | 0.7073 | 0.8228 | 0.5775 | 0.9134 | 0.8772 | 0.4440 | 0.9140 | 0.9033 | 0.9328 | 0.9178 |

# H   Full experiment on Unified Hyperparameter

The complete result on the regression task is shown in Table 9, and the complete result on classification task is shown in Table 10. All experiments are run on the RTX 3090 with more than 1000 hours.

Table 9: Regression task benchmark on DAVIS, KIBA, and BindingDB datasets, respectively.

| Category | Models | DAVIS | | | | | | KIBA | | | | | | BindingDB | | | | | |
|---|---|---|---|---|---|---|---|---|---|---|---|---|---|---|---|---|---|---|---|
| | | MSE | MAE | R2 | PCC | CI | Spearman | MSE | MAE | R2 | PCC | CI | Spearman | MSE | MAE | R2 | PCC | CI | Spearman |
| GNN | GraphDTA-GCN | 0.2771 | 0.2947 | 0.5873 | 0.7695 | 0.8521 | 0.6241 | 0.0005 | 0.0155 | 0.3924 | 0.6291 | 0.7502 | 0.6448 | 0.5033 | 0.4314 | 0.7259 | 0.8536 | 0.8576 | 0.7795 |
| | GraphDTA-GAT | 0.2806 | 0.2928 | 0.5820 | 0.7856 | 0.8576 | 0.6331 | 0.0004 | 0.0123 | 0.6012 | 0.7720 | 0.7977 | 0.7259 | 0.5613 | 0.4460 | 0.6943 | 0.8358 | 0.8496 | 0.7662 |
| | GraphDTA-GATGCN | 0.2570 | 0.2798 | 0.6173 | 0.7867 | 0.8607 | 0.6380 | 0.0004 | 0.0126 | 0.6128 | 0.7836 | 0.8086 | 0.7357 | 0.5447 | 0.4450 | 0.7033 | 0.8415 | 0.8489 | 0.7641 |
| | GraphDTA-GIN | 0.2309 | 0.2714 | 0.6562 | 0.8105 | 0.8711 | 0.6540 | 0.0004 | 0.0131 | 0.5544 | 0.7451 | 0.7996 | 0.7400 | 0.5033 | 0.4314 | 0.7259 | 0.8537 | 0.8576 | 0.7795 |
| | GraphCPI-GCN | 0.3291 | 0.3388 | 0.5100 | 0.7265 | 0.8294 | 0.5885 | 0.0006 | 0.0156 | 0.3454 | 0.5864 | 0.7378 | 0.6154 | 0.6372 | 0.5002 | 0.6530 | 0.8133 | 0.8329 | 0.7360 |
| | GraphCPI-GAT | 0.3649 | 0.3852 | 0.4566 | 0.7609 | 0.8498 | 0.6215 | 0.0007 | 0.0179 | 0.1836 | 0.4196 | 0.6845 | 0.4896 | 0.6565 | 0.5007 | 0.6424 | 0.8163 | 0.8350 | 0.7407 |
| | GraphCPI-GATGCN | 0.3078 | 0.3193 | 0.5416 | 0.7397 | 0.8365 | 0.5995 | 0.0005 | 0.0156 | 0.4062 | 0.6421 | 0.7530 | 0.6522 | 0.6023 | 0.4858 | 0.6719 | 0.8242 | 0.8370 | 0.7442 |
| | GraphCPI-GIN | 0.2413 | 0.2848 | 0.6407 | 0.8020 | 0.8671 | 0.6478 | 0.0005 | 0.0138 | 0.4963 | 0.7093 | 0.7808 | 0.7049 | 0.5069 | 0.4247 | 0.7239 | 0.8531 | 0.8572 | 0.7784 |
| | MGraphDTA | 0.2179 | 0.2355 | 0.6755 | 0.8239 | 0.8820 | 0.6704 | 0.0003 | 0.0090 | 0.7208 | 0.8508 | 0.8649 | 0.8439 | 0.4887 | 0.3852 | 0.7338 | 0.8586 | 0.8649 | 0.7897 |
| | SAGDTA | 0.2656 | 0.2796 | 0.6045 | 0.7805 | 0.8675 | 0.6498 | 0.0039 | 0.0127 | -3.3777 | 0.5064 | 0.8096 | 0.7547 | 0.6590 | 0.4709 | 0.6410 | 0.8061 | 0.8366 | 0.7418 |
| | EmbedDTI | 0.2561 | 0.2798 | 0.6186 | 0.7874 | 0.8624 | 0.6410 | 0.0007 | 0.0175 | 0.2364 | 0.6864 | 0.6374 | 0.6897 | 0.5095 | 0.4193 | 0.7225 | 0.8516 | 0.8559 | 0.7765 |
| | DeepGLSTM | 0.2915 | 0.2941 | 0.5659 | 0.7605 | 0.8476 | 0.6176 | 0.0003 | 0.0101 | 0.6919 | 0.8325 | 0.8480 | 0.8231 | 0.5385 | 0.4246 | 0.7067 | 0.8433 | 0.8529 | 0.7713 |
| | CPI | 0.3503 | 0.3428 | 0.4784 | 0.6692 | 0.8319 | 0.5873 | 0.0003 | 0.0117 | 0.6274 | 0.7926 | 0.8190 | 0.7767 | 0.6962 | 0.5244 | 0.6208 | 0.7900 | 0.8261 | 0.7252 |
| | BACPI | 0.4036 | 0.3534 | 0.3990 | 0.6518 | 0.7982 | 0.5373 | 0.0006 | 0.0122 | 0.5320 | 0.7338 | 0.8175 | 0.7719 | 0.6468 | 0.4805 | 0.6477 | 0.8109 | 0.8297 | 0.7303 |
| | DeepNC-HGC | 0.2782 | 0.3010 | 0.5857 | 0.7669 | 0.8551 | 0.6297 | 0.0005 | 0.0141 | 0.4983 | 0.7127 | 0.7823 | 0.7052 | 0.5611 | 0.4512 | 0.6944 | 0.8367 | 0.8464 | 0.7600 |
| | DeepNC-GEN | 0.2543 | 0.2830 | 0.6213 | 0.7893 | 0.8634 | 0.6419 | 0.0004 | 0.0132 | 0.5355 | 0.7447 | 0.7981 | 0.7349 | 0.5561 | 0.4350 | 0.6971 | 0.8395 | 0.8496 | 0.7629 |
| | DrugBAN | 0.2391 | 0.2663 | 0.6440 | 0.8035 | 0.8757 | 0.6619 | 0.0004 | 0.0104 | 0.5611 | 0.7665 | 0.8388 | 0.8081 | 0.4485 | 0.3945 | 0.7557 | 0.8696 | 0.8693 | 0.7989 |
| | GANDTI | 0.3082 | 0.3305 | 0.5410 | 0.7391 | 0.8414 | 0.6079 | 0.0003 | 0.0108 | 0.6569 | 0.8146 | 0.8342 | 0.8045 | 0.6714 | 0.5221 | 0.6343 | 0.7999 | 0.8322 | 0.7342 |
| | BridgeDPI | 0.3623 | 0.3432 | 0.6477 | 0.8062 | 0.8991 | 0.6704 | 0.0004 | 0.0133 | 0.5686 | 0.7357 | 0.7849 | 0.7239 | 0.4482 | 0.3900 | 0.7559 | 0.8702 | 0.8698 | 0.7982 |
| | ColdDTA | 0.2346 | 0.2511 | 0.6507 | 0.8083 | 0.8693 | 0.6501 | 0.0004 | 0.0129 | 0.5948 | 0.7764 | 0.8018 | 0.7451 | 0.4697 | 0.3783 | 0.7442 | 0.8645 | 0.8644 | 0.7866 |
| | SubMDTA | 0.2326 | 0.2726 | 0.6537 | 0.8091 | 0.8691 | 0.6503 | 0.0003 | 0.0098 | 0.6855 | 0.8324 | 0.8485 | 0.8243 | 0.4566 | 0.3995 | 0.7513 | 0.8677 | 0.8670 | 0.7953 |
| | IMAEN | 0.2412 | 0.2764 | 0.6409 | 0.7893 | 0.8721 | 0.6557 | 0.0004 | 0.0122 | 0.5800 | 0.7632 | 0.8061 | 0.7494 | 0.4720 | 0.3935 | 0.7429 | 0.8637 | 0.8553 | 0.7938 |
| Transformer | CSDTI | 0.3029 | 0.3011 | 0.5490 | 0.7436 | 0.8395 | 0.6045 | 0.0007 | 0.0184 | 0.2448 | 0.4504 | 0.6475 | 0.3937 | 0.6408 | 0.4723 | 0.6510 | 0.8118 | 0.8369 | 0.7422 |
| | TDGraphDTA | 0.2217 | 0.2399 | 0.6698 | 0.8201 | 0.6685 | 0.8804 | 0.0008 | 0.0292 | 0.0533 | 0.2604 | 0.3429 | 0.6243 | 0.4750 | 0.3894 | 0.7413 | 0.8631 | 0.7887 | 0.8642 |
| | AMMVF | 0.3325 | 0.3433 | 0.5048 | 0.7238 | 0.8307 | 0.5896 | 0.0006 | 0.0147 | 0.4696 | 0.6957 | 0.7711 | 0.6879 | 0.6597 | 0.4879 | 0.6407 | 0.8086 | 0.8336 | 0.7398 |
| | IIFDTI | 0.2741 | 0.3006 | 0.5918 | 0.7816 | 0.8500 | 0.6202 | 0.0005 | 0.0156 | 0.2584 | 0.7894 | 0.7952 | 0.7439 | 0.5097 | 0.4474 | 0.7170 | 0.8576 | 0.8585 | 0.7792 |
| | ICAN | 0.3481 | 0.3185 | 0.4816 | 0.7168 | 0.8211 | 0.5822 | 0.0008 | 0.0194 | 0.1159 | 0.7646 | 0.8256 | 0.7877 | 0.6582 | 0.5028 | 0.6415 | 0.8145 | 0.8277 | 0.7271 |
| | MolTrans | 0.2588 | 0.2740 | 0.6146 | 0.7906 | 0.8601 | 0.5927 | 0.0003 | 0.0114 | 0.6378 | 0.8064 | 0.8453 | 0.7865 | 0.5138 | 0.4101 | 0.7201 | 0.8517 | 0.8570 | 0.7786 |
| | TransformerCPI | 0.2869 | 0.3389 | 0.5728 | 0.7750 | 0.8326 | 0.5910 | 0.0008 | 0.0233 | 0.0728 | 0.8051 | 0.8357 | 0.7809 | 0.5704 | 0.4624 | 0.6894 | 0.8394 | 0.8426 | 0.7507 |
| | MRBDTA | 0.2350 | 0.2565 | 0.6499 | 0.8069 | 0.8775 | 0.6639 | 0.0006 | 0.0171 | 0.3210 | 0.5718 | 0.7239 | 0.5878 | 0.4977 | 0.4133 | 0.7289 | 0.8557 | 0.8629 | 0.7874 |
| | FOTFCPI | 0.2803 | 0.3004 | 0.5825 | 0.7704 | 0.8546 | 0.6286 | 0.0004 | 0.0133 | 0.5342 | 0.7552 | 0.7947 | 0.7353 | 0.5743 | 0.4472 | 0.6872 | 0.8322 | 0.8444 | 0.7569 |
| | our | 0.2063 | 0.2481 | 0.6927 | 0.8330 | 0.8901 | 0.6839 | 0.0003 | 0.0094 | 0.7168 | 0.8512 | 0.8677 | 0.8432 | 0.4651 | 0.3878 | 0.7467 | 0.8657 | 0.8683 | 0.7956 |

Table 10: Classification task benchmark on Human, *C.elegans*, and Drugbank datasets, respectively. Here − means that the method can not be reproduced on this datasets.

| Categories | Models | Human | | | | | | | C.elegans | | | | | | | Drugbank | | | | | | |
|---|---|---|---|---|---|---|---|---|---|---|---|---|---|---|---|---|---|---|---|---|---|---|
| | | ROC-AUC | PR-AUC | Range-AUC | Acc. | Precision | Recall | F1 | ROC-AUC | PR-AUC | Range-AUC | Acc. | Precision | Recall | F1 | ROC-AUC | PR-AUC | Range-AUC | Acc. | Precision | Recall | F1 |
| GNN | GraphDTA-GCN | 0.9222 | 0.8922 | 0.4852 | 0.9224 | 0.9210 | 0.9293 | 0.9251 | 0.9488 | 0.9174 | 0.5342 | 0.9468 | 0.9368 | 0.9564 | 0.9465 | 0.7590 | 0.6925 | 0.1490 | 0.7589 | 0.7458 | 0.7842 | 0.7645 |
| | GraphDTA-GAT | 0.8935 | 0.8557 | 0.3986 | 0.8937 | 0.8925 | 0.9024 | 0.8974 | 0.9289 | 0.8909 | 0.4942 | 0.9287 | 0.9154 | 0.9423 | 0.9286 | 0.7684 | 0.7030 | 0.1595 | 0.7683 | 0.7587 | 0.7855 | 0.7718 |
| | GraphDTA-GATGCN | 0.9296 | 0.9024 | 0.5030 | 0.9297 | 0.9298 | 0.9342 | 0.9319 | 0.9487 | 0.9176 | 0.5318 | 0.9484 | 0.9335 | 0.9640 | 0.9485 | 0.7712 | 0.7046 | 0.1585 | 0.7711 | 0.7573 | 0.7968 | 0.7764 |
| | GraphDTA-GIN | 0.9019 | 0.8674 | 0.4393 | 0.9021 | 0.9037 | 0.9065 | 0.9051 | 0.9470 | 0.9174 | 0.5342 | 0.9468 | 0.9368 | 0.9564 | 0.9465 | 0.7871 | 0.7232 | 0.1797 | 0.7871 | 0.7797 | 0.7988 | 0.7892 |
| | GraphCPI-GCN | 0.9034 | 0.8649 | 0.4048 | 0.9040 | 0.8941 | 0.9229 | 0.9083 | 0.9322 | 0.8940 | 0.4967 | 0.9319 | 0.9155 | 0.9493 | 0.9320 | 0.7362 | 0.6688 | 0.1292 | 0.7361 | 0.7178 | 0.7764 | 0.7459 |
| | GraphCPI-GAT | 0.8935 | 0.8557 | 0.3986 | 0.8937 | 0.8925 | 0.9024 | 0.8974 | 0.9281 | 0.8913 | 0.4965 | 0.9279 | 0.9181 | 0.9373 | 0.9275 | 0.7515 | 0.6862 | 0.1470 | 0.7515 | 0.7433 | 0.7667 | 0.7548 |
| | GraphCPI-GATGCN | 0.9097 | 0.8763 | 0.4644 | 0.9099 | 0.9090 | 0.9169 | 0.9129 | 0.9372 | 0.9028 | 0.5109 | 0.9370 | 0.9245 | 0.9496 | 0.9369 | 0.7561 | 0.6909 | 0.1508 | 0.7561 | 0.7479 | 0.7711 | 0.7593 |
| | GraphCPI-GIN | 0.8870 | 0.8484 | 0.3791 | 0.8872 | 0.8881 | 0.8935 | 0.8908 | 0.9407 | 0.9072 | 0.5173 | 0.9405 | 0.9270 | 0.9543 | 0.9404 | 0.7866 | 0.7241 | 0.1845 | 0.7866 | 0.7840 | 0.7903 | 0.7870 |
| | MGraphDTA | 0.9408 | 0.9166 | 0.5244 | 0.9410 | 0.9393 | 0.9466 | 0.9429 | 0.9631 | 0.9407 | 0.5795 | 0.9630 | 0.9535 | 0.9723 | 0.9628 | 0.8146 | 0.7539 | 0.2154 | 0.8146 | 0.8090 | 0.8228 | 0.8157 |
| | SAGDTA | 0.9021 | 0.8728 | 0.4631 | 0.9018 | 0.9172 | 0.8897 | 0.9032 | 0.9380 | 0.9050 | 0.5165 | 0.9378 | 0.9281 | 0.9472 | 0.9375 | 0.7655 | 0.7018 | 0.1631 | 0.7655 | 0.7624 | 0.7700 | 0.7662 |
| | EmbedDTI | - | - | - | - | - | - | - | - | - | - | - | - | - | - | 0.7625 | 0.6963 | 0.1526 | 0.7625 | 0.7504 | 0.7852 | 0.7674 |
| | DeepGLSTM | 0.9180 | 0.8857 | 0.4758 | 0.9183 | 0.9145 | 0.9281 | 0.9212 | 0.9414 | 0.9103 | 0.5257 | 0.9412 | 0.9325 | 0.9496 | 0.9409 | 0.7640 | 0.6988 | 0.1567 | 0.7640 | 0.7551 | 0.7801 | 0.7674 |
| | CPI | 0.9110 | 0.8789 | 0.4638 | 0.9111 | 0.9126 | 0.9152 | 0.9138 | 0.9312 | 0.8940 | 0.4983 | 0.9310 | 0.9176 | 0.9446 | 0.9309 | 0.7467 | 0.6812 | 0.1434 | 0.7466 | 0.7368 | 0.7683 | 0.7515 |
| | BACPI | 0.9249 | 0.8958 | 0.4924 | 0.9251 | 0.9239 | 0.9313 | 0.9276 | 0.9556 | 0.9314 | 0.5637 | 0.9555 | 0.9493 | 0.9610 | 0.9551 | 0.7748 | 0.7100 | 0.1664 | 0.7748 | 0.7663 | 0.7895 | 0.7776 |
| | DeepNC-HGC | 0.8796 | 0.8399 | 0.3579 | 0.8798 | 0.8823 | 0.8845 | 0.8834 | 0.9418 | 0.9106 | 0.5251 | 0.9417 | 0.9325 | 0.9504 | 0.9413 | 0.7673 | 0.7013 | 0.1571 | 0.7673 | 0.7556 | 0.7889 | 0.7718 |
| | DeepNC-GEN | 0.9178 | 0.8873 | 0.4809 | 0.9180 | 0.9186 | 0.9224 | 0.9205 | 0.9501 | 0.9203 | 0.5385 | 0.9499 | 0.9366 | 0.9637 | 0.9498 | 0.7464 | 0.6785 | 0.1357 | 0.7463 | 0.7262 | 0.7928 | 0.7571 |
| | DrugBAN | 0.9302 | 0.9048 | 0.5083 | 0.9302 | 0.9342 | 0.9299 | 0.9320 | 0.9596 | 0.9346 | 0.5645 | 0.9594 | 0.9478 | 0.9710 | 0.9593 | 0.8188 | 0.7610 | 0.2325 | 0.8188 | 0.8203 | 0.8157 | 0.8179 |
| | GANDTI | 0.9333 | 0.9090 | 0.5162 | 0.9333 | 0.9374 | 0.9327 | 0.9351 | 0.9404 | 0.9082 | 0.5213 | 0.9403 | 0.9300 | 0.9504 | 0.9400 | 0.7427 | 0.6737 | 0.1298 | 0.7425 | 0.7188 | 0.7954 | 0.7550 |
| | BridgeDPI | 0.9456 | 0.9250 | 0.5435 | 0.9456 | 0.9485 | 0.9457 | 0.9471 | 0.9651 | 0.9470 | 0.6054 | 0.9651 | 0.9629 | 0.9663 | 0.9646 | 0.7825 | 0.7181 | 0.1742 | 0.7825 | 0.7739 | 0.7974 | 0.7853 |
| | ColdDTA | 0.9420 | 0.9203 | 0.5348 | 0.9421 | 0.9453 | 0.9420 | 0.9436 | 0.9623 | 0.9403 | 0.5785 | 0.9622 | 0.9543 | 0.9697 | 0.9619 | 0.8186 | 0.7578 | 0.2175 | 0.8186 | 0.8108 | 0.8300 | 0.8203 |
| | SubMDTA | 0.9326 | 0.9054 | 0.5066 | 0.9328 | 0.9304 | 0.9400 | 0.9351 | 0.9570 | 0.9313 | 0.5589 | 0.9568 | 0.9461 | 0.9674 | 0.9566 | 0.8045 | 0.7414 | 0.1968 | 0.8045 | 0.7946 | 0.8204 | 0.8072 |
| | IMAEN | 0.9058 | 0.8719 | 0.4489 | 0.9059 | 0.9067 | 0.9111 | 0.9089 | 0.9506 | 0.9255 | 0.5546 | 0.9505 | 0.9469 | 0.9530 | 0.9500 | 0.7995 | 0.7365 | 0.1931 | 0.7994 | 0.7918 | 0.8113 | 0.8014 |
| Transformer | CSDTI | 0.8630 | 0.8216 | 0.3223 | 0.8630 | 0.8707 | 0.8620 | 0.8663 | 0.8962 | 0.8446 | 0.3710 | 0.8958 | 0.8759 | 0.9183 | 0.8966 | 0.7269 | 0.6626 | 0.1304 | 0.7269 | 0.7196 | 0.7420 | 0.7306 |
| | TDGraphDTA | 0.9411 | 0.9226 | 0.5461 | 0.9409 | 0.9525 | 0.9316 | 0.9419 | 0.9573 | 0.9315 | 0.5591 | 0.9571 | 0.9460 | 0.9681 | 0.9569 | 0.8083 | 0.7448 | 0.1982 | 0.8082 | 0.7959 | 0.8281 | 0.8116 |
| | AMMVF | 0.9287 | 0.9004 | 0.4994 | 0.9290 | 0.9268 | 0.9362 | 0.9314 | 0.9636 | 0.9426 | 0.5892 | 0.9635 | 0.9567 | 0.9700 | 0.9632 | 0.7814 | 0.7142 | 0.1717 | 0.7814 | 0.7728 | 0.7973 | 0.7849 |
| | IIFDTI | 0.9392 | 0.9163 | 0.5279 | 0.9392 | 0.9420 | 0.9400 | 0.9409 | 0.9679 | 0.9500 | 0.6054 | 0.9679 | 0.9633 | 0.9718 | 0.9675 | 0.8084 | 0.7441 | 0.1950 | 0.8084 | 0.7934 | 0.8327 | 0.8125 |
| | ICAN | 0.9387 | 0.9134 | 0.5194 | 0.9389 | 0.9361 | 0.9460 | 0.9410 | 0.9464 | 0.9479 | 0.6146 | 0.9644 | 0.9662 | 0.9613 | 0.9637 | 0.7731 | 0.7092 | 0.1682 | 0.7731 | 0.7681 | 0.7812 | 0.7745 |
| | MolTrans | 0.9453 | 0.9233 | 0.5377 | 0.9455 | 0.9451 | 0.9492 | 0.9471 | 0.9639 | 0.9428 | 0.5847 | 0.9638 | 0.9565 | 0.9705 | 0.9634 | 0.7910 | 0.7283 | 0.1879 | 0.7909 | 0.7861 | 0.7988 | 0.7922 |
| | TransformerCPI | 0.9311 | 0.9045 | 0.5060 | 0.9312 | 0.9316 | 0.9351 | 0.9333 | 0.9581 | 0.9343 | 0.5696 | 0.9580 | 0.9501 | 0.9655 | 0.9577 | 0.8097 | 0.7470 | 0.2024 | 0.8097 | 0.7992 | 0.8265 | 0.8125 |
| | MRBDTA | 0.9447 | 0.9258 | 0.5521 | 0.9446 | 0.9523 | 0.9397 | 0.9458 | 0.9713 | 0.9569 | 0.6335 | 0.9713 | 0.9708 | 0.9710 | 0.9709 | 0.8102 | 0.7469 | 0.2003 | 0.8102 | 0.7978 | 0.8298 | 0.8134 |
| | FOTFCPI | 0.9444 | 0.9235 | 0.5413 | 0.9444 | 0.9477 | 0.9443 | 0.9459 | 0.9673 | 0.9486 | 0.6009 | 0.9672 | 0.9618 | 0.9721 | 0.9669 | 0.7845 | 0.7188 | 0.1713 | 0.7845 | 0.7713 | 0.8075 | 0.7889 |
| | our | 0.9435 | 0.9221 | 0.5400 | 0.9435 | 0.9462 | 0.9443 | 0.9451 | 0.9688 | 0.9518 | 0.6178 | 0.9688 | 0.9652 | 0.9718 | 0.9684 | 0.8035 | 0.7405 | 0.1961 | 0.8035 | 0.7941 | 0.8185 | 0.8061 |

# I  Full experiment on Optimized Hyperparameter

To evaluate the model's best performance, based on the hyperparameters given in its paper or codes, we finds the optimized hyperparameters for each model. On top of the mean value, we also provide the standard deviation across five-fold. The complete result on the regression task is shown in Table 11, and the complete result on the classification task is shown in Table 12.

Table 11: Regression task benchmark on DAVIS, KIBA, and BindingDB datasets, respectively.

| Category | Models | DAVIS | | | | | | KIBA | | | | | | BindingDB | | | | | |
|---|---|---|---|---|---|---|---|---|---|---|---|---|---|---|---|---|---|---|---|
| | | MSE | MAE | R2 | PCC | CI | Spearman | MSE ($\times 10^{-2}$) | MAE | R2 | PCC | CI | Spearman | MSE | MAE | R2 | PCC | CI | Spearman |
| GNN | GraphDTA-GCN | $0.315 \pm 0.019$ | $0.332 \pm 0.017$ | $0.531 \pm 0.028$ | $0.734 \pm 0.018$ | $0.840 \pm 0.006$ | $0.605 \pm 0.010$ | $0.279 \pm 0.059$ | $0.041 \pm 0.005$ | $-2.114 \pm 0.662$ | $0.112 \pm 0.026$ | $0.555 \pm 0.014$ | $0.156 \pm 0.039$ | $0.609 \pm 0.033$ | $0.505 \pm 0.028$ | $0.669 \pm 0.018$ | $0.820 \pm 0.011$ | $0.837 \pm 0.007$ | $0.744 \pm 0.011$ |
| | GraphDTA-GAT | $0.382 \pm 0.043$ | $0.380 \pm 0.025$ | $0.431 \pm 0.064$ | $0.671 \pm 0.039$ | $0.828 \pm 0.014$ | $0.588 \pm 0.024$ | $0.428 \pm 0.154$ | $0.051 \pm 0.011$ | $-3.769 \pm 1.712$ | $0.073 \pm 0.061$ | $0.534 \pm 0.026$ | $0.098 \pm 0.074$ | $1.020 \pm 0.032$ | $0.676 \pm 0.005$ | $0.445 \pm 0.018$ | $0.707 \pm 0.035$ | $0.786 \pm 0.017$ | $0.653 \pm 0.031$ |
| | GraphDTA-GATGCN | $0.306 \pm 0.011$ | $0.325 \pm 0.013$ | $0.544 \pm 0.017$ | $0.741 \pm 0.013$ | $0.847 \pm 0.003$ | $0.617 \pm 0.006$ | $0.311 \pm 0.234$ | $0.041 \pm 0.018$ | $-2.467 \pm 2.604$ | $0.197 \pm 0.254$ | $0.578 \pm 0.094$ | $0.210 \pm 0.240$ | $0.574 \pm 0.032$ | $0.478 \pm 0.027$ | $0.687 \pm 0.017$ | $0.831 \pm 0.011$ | $0.844 \pm 0.007$ | $0.756 \pm 0.012$ |
| | GraphDTA-GIN | $0.253 \pm 0.010$ | $0.295 \pm 0.014$ | $0.623 \pm 0.015$ | $0.791 \pm 0.008$ | $0.861 \pm 0.006$ | $0.638 \pm 0.009$ | $0.255 \pm 0.007$ | $0.039 \pm 0.006$ | $-1.840 \pm 0.779$ | $0.124 \pm 0.037$ | $0.553 \pm 0.019$ | $0.149 \pm 0.052$ | $0.563 \pm 0.038$ | $0.494 \pm 0.023$ | $0.693 \pm 0.021$ | $0.836 \pm 0.012$ | $0.842 \pm 0.007$ | $0.756 \pm 0.011$ |
| | GraphCPI-GCN | $0.394 \pm 0.046$ | $0.401 \pm 0.025$ | $0.414 \pm 0.068$ | $0.652 \pm 0.051$ | $0.806 \pm 0.016$ | $0.550 \pm 0.027$ | $1.185 \pm 1.720$ | $0.066 \pm 0.057$ | $-12.202 \pm 19.159$ | $0.228 \pm 0.253$ | $0.602 \pm 0.104$ | $0.272 \pm 0.270$ | $1.199 \pm 0.040$ | $0.756 \pm 0.019$ | $0.347 \pm 0.022$ | $0.606 \pm 0.026$ | $0.727 \pm 0.015$ | $0.533 \pm 0.033$ |
| | GraphCPI-GAT | $0.612 \pm 0.038$ | $0.501 \pm 0.021$ | $0.089 \pm 0.056$ | $0.419 \pm 0.135$ | $0.718 \pm 0.069$ | $0.396 \pm 0.123$ | $4.558 \pm 2.116$ | $0.168 \pm 0.048$ | $-49.762 \pm 23.563$ | $-0.021 \pm 0.023$ | $0.494 \pm 0.013$ | $0.031 \pm 0.024$ | $1.199 \pm 0.040$ | $0.756 \pm 0.019$ | $0.347 \pm 0.022$ | $0.606 \pm 0.026$ | $0.727 \pm 0.015$ | $0.533 \pm 0.033$ |
| | GraphCPI-GATGCN | $0.338 \pm 0.013$ | $0.364 \pm 0.007$ | $0.496 \pm 0.019$ | $0.708 \pm 0.012$ | $0.838 \pm 0.004$ | $0.604 \pm 0.007$ | $0.445 \pm 0.081$ | $0.051 \pm 0.005$ | $-3.957 \pm 0.907$ | $0.063 \pm 0.050$ | $0.522 \pm 0.025$ | $0.081 \pm 0.045$ | $0.629 \pm 0.012$ | $0.520 \pm 0.011$ | $0.657 \pm 0.007$ | $0.813 \pm 0.005$ | $0.834 \pm 0.004$ | $0.739 \pm 0.007$ |
| | GraphCPI-GIN | $0.274 \pm 0.009$ | $0.331 \pm 0.007$ | $0.593 \pm 0.013$ | $0.773 \pm 0.008$ | $0.851 \pm 0.008$ | $0.622 \pm 0.013$ | $1.681 \pm 0.946$ | $0.091 \pm 0.042$ | $-17.724 \pm 10.533$ | $0.142 \pm 0.220$ | $0.553 \pm 0.094$ | $0.149 \pm 0.246$ | $0.557 \pm 0.017$ | $0.475 \pm 0.016$ | $0.696 \pm 0.009$ | $0.838 \pm 0.006$ | $0.847 \pm 0.003$ | $0.760 \pm 0.006$ |
| | MGraphDTA | $0.232 \pm 0.012$ | $0.268 \pm 0.008$ | $0.655 \pm 0.018$ | $0.812 \pm 0.011$ | $0.869 \pm 0.007$ | $0.650 \pm 0.011$ | $0.032 \pm 0.012$ | $0.011 \pm 0.002$ | $0.642 \pm 0.133$ | $0.803 \pm 0.079$ | $0.832 \pm 0.040$ | $0.793 \pm 0.070$ | $0.529 \pm 0.011$ | $0.444 \pm 0.025$ | $0.712 \pm 0.006$ | $0.847 \pm 0.005$ | $0.852 \pm 0.005$ | $0.769 \pm 0.008$ |
| | SAGDTA | $0.324 \pm 0.064$ | $0.329 \pm 0.041$ | $0.518 \pm 0.096$ | $0.723 \pm 0.065$ | $0.833 \pm 0.027$ | $0.594 \pm 0.044$ | $0.065 \pm 0.008$ | $0.017 \pm 0.002$ | $0.279 \pm 0.085$ | $0.541 \pm 0.085$ | $0.713 \pm 0.032$ | $0.561 \pm 0.075$ | $0.529 \pm 0.011$ | $0.444 \pm 0.025$ | $0.712 \pm 0.006$ | $0.847 \pm 0.005$ | $0.852 \pm 0.005$ | $0.769 \pm 0.008$ |
| | EmbedDTI | $0.280 \pm 0.024$ | $0.310 \pm 0.028$ | $0.583 \pm 0.036$ | $0.764 \pm 0.023$ | $0.851 \pm 0.009$ | $0.623 \pm 0.013$ | $0.289 \pm 0.142$ | $0.041 \pm 0.012$ | $-2.217 \pm 1.579$ | $0.131 \pm 0.090$ | $0.558 \pm 0.038$ | $0.164 \pm 0.106$ | $0.542 \pm 0.019$ | $0.446 \pm 0.021$ | $0.705 \pm 0.010$ | $0.843 \pm 0.006$ | $0.850 \pm 0.004$ | $0.767 \pm 0.007$ |
| | DeepGLSTM | $0.316 \pm 0.023$ | $0.322 \pm 0.024$ | $0.529 \pm 0.035$ | $0.732 \pm 0.022$ | $0.841 \pm 0.007$ | $0.609 \pm 0.011$ | $8.539 \pm 7.479$ | $0.243 \pm 0.155$ | $-94.109 \pm 83.400$ | $0.040 \pm 0.083$ | $0.514 \pm 0.036$ | $0.071 \pm 0.078$ | $0.594 \pm 0.061$ | $0.474 \pm 0.046$ | $0.677 \pm 0.033$ | $0.825 \pm 0.021$ | $0.840 \pm 0.013$ | $0.750 \pm 0.021$ |
| | CPI | $0.402 \pm 0.082$ | $0.393 \pm 0.054$ | $0.401 \pm 0.122$ | $0.629 \pm 0.101$ | $0.811 \pm 0.033$ | $0.560 \pm 0.055$ | $0.052 \pm 0.003$ | $0.161 \pm 0.008$ | $0.416 \pm 0.036$ | $0.654 \pm 0.032$ | $0.734 \pm 0.037$ | $0.605 \pm 0.088$ | $0.762 \pm 0.165$ | $0.565 \pm 0.068$ | $0.585 \pm 0.090$ | $0.768 \pm 0.063$ | $0.815 \pm 0.028$ | $0.704 \pm 0.054$ |
| | BACPI | $0.334 \pm 0.015$ | $0.323 \pm 0.034$ | $0.502 \pm 0.023$ | $0.717 \pm 0.014$ | $0.827 \pm 0.006$ | $0.584 \pm 0.010$ | $0.031 \pm 0.004$ | $0.011 \pm 0.001$ | $0.658 \pm 0.043$ | $0.820 \pm 0.019$ | $0.831 \pm 0.020$ | $0.798 \pm 0.032$ | $0.550 \pm 0.010$ | $0.436 \pm 0.005$ | $0.700 \pm 0.006$ | $0.839 \pm 0.003$ | $0.845 \pm 0.002$ | $0.759 \pm 0.003$ |
| | DeepNC-HGC | $0.309 \pm 0.025$ | $0.331 \pm 0.022$ | $0.541 \pm 0.037$ | $0.738 \pm 0.025$ | $0.841 \pm 0.005$ | $0.608 \pm 0.008$ | $0.080 \pm 0.003$ | $0.019 \pm 0.001$ | $0.110 \pm 0.036$ | $0.342 \pm 0.041$ | $0.667 \pm 0.022$ | $0.448 \pm 0.048$ | $0.572 \pm 0.011$ | $0.486 \pm 0.010$ | $0.689 \pm 0.006$ | $0.833 \pm 0.005$ | $0.844 \pm 0.003$ | $0.757 \pm 0.005$ |
| | DeepNC-GEN | $0.270 \pm 0.012$ | $0.298 \pm 0.012$ | $0.597 \pm 0.017$ | $0.776 \pm 0.012$ | $0.852 \pm 0.009$ | $0.624 \pm 0.014$ | $0.135 \pm 0.045$ | $0.027 \pm 0.006$ | $-0.509 \pm 0.505$ | $0.266 \pm 0.073$ | $0.608 \pm 0.037$ | $0.301 \pm 0.097$ | $0.578 \pm 0.020$ | $0.474 \pm 0.012$ | $0.685 \pm 0.011$ | $0.830 \pm 0.006$ | $0.840 \pm 0.003$ | $0.749 \pm 0.005$ |
| | DrugBAN | $0.242 \pm 0.007$ | $0.272 \pm 0.007$ | $0.640 \pm 0.010$ | $0.801 \pm 0.007$ | $0.869 \pm 0.003$ | $0.651 \pm 0.005$ | $0.029 \pm 0.003$ | $0.011 \pm 0.001$ | $0.676 \pm 0.032$ | $0.826 \pm 0.020$ | $0.832 \pm 0.013$ | $0.800 \pm 0.022$ | $0.465 \pm 0.018$ | $0.420 \pm 0.016$ | $0.747 \pm 0.010$ | $0.865 \pm 0.006$ | $0.862 \pm 0.003$ | $0.787 \pm 0.006$ |
| | GANDTI | $0.318 \pm 0.018$ | $0.301 \pm 0.021$ | $0.527 \pm 0.027$ | $0.732 \pm 0.016$ | $0.844 \pm 0.006$ | $0.616 \pm 0.013$ | $0.030 \pm 0.002$ | $0.011 \pm 0.000$ | $0.662 \pm 0.026$ | $0.816 \pm 0.016$ | $0.831 \pm 0.007$ | $0.800 \pm 0.011$ | $0.621 \pm 0.012$ | $0.489 \pm 0.007$ | $0.662 \pm 0.006$ | $0.815 \pm 0.003$ | $0.836 \pm 0.002$ | $0.741 \pm 0.005$ |
| | BridgeDPI | $1.241 \pm 1.432$ | $0.705 \pm 0.600$ | $-0.848 \pm 2.133$ | $0.657 \pm 0.209$ | $0.827 \pm 0.078$ | $0.581 \pm 0.128$ | $0.325 \pm 0.109$ | $0.010 \pm 0.000$ | $0.638 \pm 0.121$ | $0.821 \pm 0.038$ | $0.857 \pm 0.001$ | $0.839 \pm 0.002$ | $0.514 \pm 0.011$ | $0.413 \pm 0.006$ | $0.720 \pm 0.006$ | $0.853 \pm 0.003$ | $0.861 \pm 0.002$ | $0.783 \pm 0.003$ |
| | ColdDTA | $0.220 \pm 0.009$ | $0.259 \pm 0.007$ | $0.672 \pm 0.014$ | $0.820 \pm 0.009$ | $0.880 \pm 0.004$ | $0.666 \pm 0.006$ | $0.110 \pm 0.029$ | $0.026 \pm 0.003$ | $-0.224 \pm 0.329$ | $0.441 \pm 0.217$ | $0.673 \pm 0.079$ | $0.459 \pm 0.191$ | $0.463 \pm 0.008$ | $0.391 \pm 0.007$ | $0.748 \pm 0.004$ | $0.866 \pm 0.002$ | $0.866 \pm 0.001$ | $0.789 \pm 0.002$ |
| | SubMDTA | $0.289 \pm 0.012$ | $0.353 \pm 0.020$ | $0.570 \pm 0.018$ | $0.766 \pm 0.012$ | $0.841 \pm 0.007$ | $0.604 \pm 0.012$ | $0.029 \pm 0.002$ | $0.011 \pm 0.001$ | $0.678 \pm 0.025$ | $0.825 \pm 0.015$ | $0.836 \pm 0.011$ | $0.805 \pm 0.018$ | $0.532 \pm 0.032$ | $0.476 \pm 0.026$ | $0.710 \pm 0.017$ | $0.845 \pm 0.010$ | $0.852 \pm 0.006$ | $0.772 \pm 0.011$ |
| | IMAEN | $0.230 \pm 0.009$ | $0.245 \pm 0.004$ | $0.657 \pm 0.014$ | $0.812 \pm 0.008$ | $0.874 \pm 0.004$ | $0.658 \pm 0.007$ | $0.046 \pm 0.018$ | $0.014 \pm 0.003$ | $0.484 \pm 0.196$ | $0.684 \pm 0.176$ | $0.781 \pm 0.056$ | $0.697 \pm 0.121$ | $0.479 \pm 0.012$ | $0.399 \pm 0.008$ | $0.739 \pm 0.006$ | $0.861 \pm 0.004$ | $0.863 \pm 0.002$ | $0.788 \pm 0.005$ |
| Transformer | CSDTI | $0.331 \pm 0.012$ | $0.339 \pm 0.020$ | $0.508 \pm 0.017$ | $0.720 \pm 0.009$ | $0.832 \pm 0.005$ | $0.593 \pm 0.008$ | $0.088 \pm 0.004$ | $0.020 \pm 0.001$ | $0.014 \pm 0.041$ | $0.273 \pm 0.084$ | $0.628 \pm 0.047$ | $0.350 \pm 0.124$ | $0.768 \pm 0.021$ | $0.572 \pm 0.012$ | $0.582 \pm 0.012$ | $0.765 \pm 0.008$ | $0.805 \pm 0.004$ | $0.689 \pm 0.006$ |
| | TDGraphDTA | $0.222 \pm 0.005$ | $0.265 \pm 0.007$ | $0.669 \pm 0.008$ | $0.820 \pm 0.004$ | $0.653 \pm 0.011$ | $0.871 \pm 0.007$ | $0.091 \pm 0.019$ | $0.022 \pm 0.003$ | $-0.009 \pm 0.209$ | $0.330 \pm 0.117$ | $0.327 \pm 0.125$ | $0.619 \pm 0.046$ | $0.497 \pm 0.016$ | $0.418 \pm 0.010$ | $0.729 \pm 0.009$ | $0.855 \pm 0.005$ | $0.777 \pm 0.005$ | $0.857 \pm 0.003$ |
| | AMMVF | $0.377 \pm 0.030$ | $0.365 \pm 0.043$ | $0.439 \pm 0.044$ | $0.669 \pm 0.036$ | $0.815 \pm 0.005$ | $0.586 \pm 0.024$ | | | | | | | | | | | | |
| | IIFDTI | $0.313 \pm 0.018$ | $0.378 \pm 0.039$ | $0.534 \pm 0.027$ | $0.754 \pm 0.008$ | $0.836 \pm 0.008$ | $0.598 \pm 0.013$ | | | | | | | $0.634 \pm 0.024$ | $0.527 \pm 0.020$ | $0.655 \pm 0.013$ | $0.820 \pm 0.009$ | $0.832 \pm 0.006$ | $0.737 \pm 0.009$ |
| | ICAN | $0.371 \pm 0.013$ | $0.359 \pm 0.007$ | $0.448 \pm 0.020$ | $0.681 \pm 0.010$ | $0.818 \pm 0.006$ | $0.582 \pm 0.009$ | $0.089 \pm 0.000$ | $0.021 \pm 0.000$ | $-2.052 \pm 0.000$ | - | $0.500 \pm 0.000$ | - | $0.747 \pm 0.031$ | $0.580 \pm 0.018$ | $0.593 \pm 0.017$ | $0.785 \pm 0.006$ | $0.813 \pm 0.004$ | $0.707 \pm 0.005$ |
| | MolTrans | $0.410 \pm 0.136$ | $0.382 \pm 0.045$ | $0.390 \pm 0.202$ | $0.670 \pm 0.107$ | $0.812 \pm 0.039$ | $0.591 \pm 0.042$ | $4.314 \pm 2.290$ | $0.169 \pm 0.055$ | $-47.055 \pm 25.515$ | $0.093 \pm 0.053$ | $0.540 \pm 0.021$ | $0.112 \pm 0.058$ | $0.695 \pm 0.183$ | $0.523 \pm 0.063$ | $0.621 \pm 0.100$ | $0.803 \pm 0.053$ | $0.822 \pm 0.009$ | $0.745 \pm 0.030$ |
| | TransformerCPI | $0.393 \pm 0.022$ | $0.445 \pm 0.043$ | $0.415 \pm 0.032$ | $0.695 \pm 0.018$ | $0.802 \pm 0.008$ | $0.542 \pm 0.015$ | $0.070 \pm 0.003$ | $0.019 \pm 0.001$ | $0.217 \pm 0.033$ | $0.779 \pm 0.006$ | $0.800 \pm 0.002$ | $0.742 \pm 0.004$ | $0.659 \pm 0.040$ | $0.548 \pm 0.024$ | $0.641 \pm 0.022$ | $0.825 \pm 0.017$ | $0.829 \pm 0.013$ | $0.727 \pm 0.022$ |
| | MRBDTA | $0.241 \pm 0.005$ | $0.265 \pm 0.006$ | $0.640 \pm 0.008$ | $0.802 \pm 0.003$ | $0.870 \pm 0.007$ | $0.651 \pm 0.011$ | $0.050 \pm 0.005$ | $0.016 \pm 0.001$ | $0.360 \pm 0.058$ | $0.600 \pm 0.050$ | $0.735 \pm 0.015$ | $0.613 \pm 0.031$ | $0.507 \pm 0.006$ | $0.411 \pm 0.006$ | $0.724 \pm 0.003$ | $0.853 \pm 0.002$ | $0.862 \pm 0.002$ | $0.786 \pm 0.003$ |
| | FOTFCPI | $0.305 \pm 0.012$ | $0.302 \pm 0.019$ | $0.546 \pm 0.018$ | $0.749 \pm 0.011$ | $0.839 \pm 0.009$ | $0.604 \pm 0.015$ | $0.229 \pm 0.180$ | $0.034 \pm 0.016$ | $-1.554 \pm 2.003$ | $0.235 \pm 0.264$ | $0.587 \pm 0.086$ | $0.414 \pm 0.292$ | $0.567 \pm 0.008$ | $0.432 \pm 0.012$ | $0.695 \pm 0.004$ | $0.832 \pm 0.004$ | $0.848 \pm 0.006$ | $0.763 \pm 0.008$ |
| | Our combos | $0.211 \pm 0.007$ | $0.251 \pm 0.008$ | $0.685 \pm 0.011$ | $0.829 \pm 0.006$ | $0.886 \pm 0.004$ | $0.676 \pm 0.007$ | $0.026 \pm 0.004$ | $0.010 \pm 0.001$ | $0.710 \pm 0.051$ | $0.845 \pm 0.031$ | $0.849 \pm 0.023$ | $0.827 \pm 0.037$ | $0.461 \pm 0.006$ | $0.389 \pm 0.007$ | $0.749 \pm 0.003$ | $0.867 \pm 0.002$ | $0.869 \pm 0.002$ | $0.796 \pm 0.003$ |

Table 12: Classification task benchmark on Human, *C.elegans*, and Drugbank datasets, respectively. Here − means that the method can not be reproduced on this datasets.

| Categories | Models | Human | | | | | | | C.elegans | | | | | | | Drugbank | | | | | | |
|---|---|---|---|---|---|---|---|---|---|---|---|---|---|---|---|---|---|---|---|---|---|---|
| | | ROC-AUC | PR-AUC | Range-AUC | Acc. | Precision | Recall | F1 | ROC-AUC | PR-AUC | Range-AUC | Acc. | Precision | Recall | F1 | ROC-AUC | PR-AUC | Range-AUC | Acc. | Precision | Recall | F1 |
| GNN | GraphDTA-GCN | $0.915 \pm 0.008$ | $0.882 \pm 0.010$ | $0.467 \pm 0.024$ | $0.915 \pm 0.008$ | $0.913 \pm 0.012$ | $0.924 \pm 0.017$ | $0.918 \pm 0.008$ | $0.940 \pm 0.008$ | $0.905 \pm 0.015$ | $0.510 \pm 0.035$ | $0.940 \pm 0.008$ | $0.924 \pm 0.018$ | $0.958 \pm 0.006$ | $0.940 \pm 0.007$ | $0.751 \pm 0.010$ | $0.685 \pm 0.010$ | $0.142 \pm 0.008$ | $0.751 \pm 0.010$ | $0.737 \pm 0.011$ | $0.780 \pm 0.010$ | $0.758 \pm 0.010$ |
| | GraphDTA-GAT | $0.889 \pm 0.008$ | $0.847 \pm 0.010$ | $0.352 \pm 0.040$ | $0.889 \pm 0.008$ | $0.880 \pm 0.012$ | $0.909 \pm 0.016$ | $0.894 \pm 0.008$ | $0.932 \pm 0.003$ | $0.898 \pm 0.006$ | $0.506 \pm 0.010$ | $0.932 \pm 0.003$ | $0.924 \pm 0.007$ | $0.939 \pm 0.008$ | $0.931 \pm 0.003$ | $0.754 \pm 0.005$ | $0.688 \pm 0.005$ | $0.146 \pm 0.005$ | $0.754 \pm 0.005$ | $0.742 \pm 0.007$ | $0.777 \pm 0.010$ | $0.759 \pm 0.005$ |
| | GraphDTA-GATGCN | $0.920 \pm 0.003$ | $0.891 \pm 0.005$ | $0.488 \pm 0.013$ | $0.920 \pm 0.003$ | $0.924 \pm 0.012$ | $0.920 \pm 0.016$ | $0.922 \pm 0.004$ | $0.944 \pm 0.003$ | $0.912 \pm 0.008$ | $0.525 \pm 0.016$ | $0.944 \pm 0.003$ | $0.931 \pm 0.012$ | $0.957 \pm 0.010$ | $0.944 \pm 0.003$ | $0.766 \pm 0.005$ | $0.698 \pm 0.005$ | $0.152 \pm 0.006$ | $0.766 \pm 0.005$ | $0.749 \pm 0.008$ | $0.797 \pm 0.014$ | $0.772 \pm 0.005$ |
| | GraphDTA-GIN | $0.896 \pm 0.001$ | $0.859 \pm 0.003$ | $0.408 \pm 0.023$ | $0.897 \pm 0.001$ | $0.895 \pm 0.006$ | $0.905 \pm 0.007$ | $0.900 \pm 0.001$ | $0.947 \pm 0.006$ | $0.916 \pm 0.010$ | $0.531 \pm 0.019$ | $0.947 \pm 0.006$ | $0.933 \pm 0.011$ | $0.962 \pm 0.007$ | $0.947 \pm 0.006$ | $0.781 \pm 0.004$ | $0.715 \pm 0.006$ | $0.169 \pm 0.009$ | $0.781 \pm 0.004$ | $0.769 \pm 0.010$ | $0.803 \pm 0.009$ | $0.785 \pm 0.003$ |
| | GraphCPI-GCN | $0.909 \pm 0.008$ | $0.874 \pm 0.010$ | $0.449 \pm 0.027$ | $0.909 \pm 0.008$ | $0.906 \pm 0.009$ | $0.918 \pm 0.012$ | $0.912 \pm 0.008$ | $0.928 \pm 0.008$ | $0.887 \pm 0.014$ | $0.473 \pm 0.051$ | $0.928 \pm 0.008$ | $0.908 \pm 0.016$ | $0.950 \pm 0.009$ | $0.928 \pm 0.007$ | $0.736 \pm 0.005$ | $0.667 \pm 0.004$ | $0.125 \pm 0.003$ | $0.736 \pm 0.005$ | $0.712 \pm 0.005$ | $0.792 \pm 0.005$ | $0.750 \pm 0.004$ |
| | GraphCPI-GAT | $0.862 \pm 0.004$ | $0.813 \pm 0.003$ | $0.266 \pm 0.013$ | $0.863 \pm 0.005$ | $0.848 \pm 0.006$ | $0.895 \pm 0.018$ | $0.871 \pm 0.006$ | $0.913 \pm 0.005$ | $0.871 \pm 0.007$ | $0.459 \pm 0.023$ | $0.913 \pm 0.005$ | $0.903 \pm 0.011$ | $0.923 \pm 0.013$ | $0.913 \pm 0.005$ | $0.713 \pm 0.004$ | $0.644 \pm 0.005$ | $0.109 \pm 0.004$ | $0.713 \pm 0.004$ | $0.683 \pm 0.007$ | $0.791 \pm 0.008$ | $0.733 \pm 0.001$ |
| | GraphCPI-GATGCN | $0.910 \pm 0.009$ | $0.880 \pm 0.014$ | $0.465 \pm 0.034$ | $0.910 \pm 0.008$ | $0.918 \pm 0.018$ | $0.906 \pm 0.017$ | $0.912 \pm 0.008$ | $0.939 \pm 0.005$ | $0.904 \pm 0.010$ | $0.511 \pm 0.016$ | $0.939 \pm 0.005$ | $0.922 \pm 0.011$ | $0.957 \pm 0.005$ | $0.939 \pm 0.005$ | $0.744 \pm 0.004$ | $0.676 \pm 0.005$ | $0.133 \pm 0.006$ | $0.744 \pm 0.004$ | $0.724 \pm 0.009$ | $0.785 \pm 0.012$ | $0.754 \pm 0.004$ |
| | GraphCPI-GIN | $0.898 \pm 0.008$ | $0.862 \pm 0.010$ | $0.413 \pm 0.037$ | $0.898 \pm 0.008$ | $0.899 \pm 0.013$ | $0.904 \pm 0.020$ | $0.901 \pm 0.009$ | $0.938 \pm 0.008$ | $0.907 \pm 0.014$ | $0.521 \pm 0.024$ | $0.938 \pm 0.008$ | $0.931 \pm 0.015$ | $0.945 \pm 0.007$ | $0.938 \pm 0.008$ | $0.783 \pm 0.009$ | $0.719 \pm 0.011$ | $0.177 \pm 0.014$ | $0.782 \pm 0.009$ | $0.777 \pm 0.013$ | $0.792 \pm 0.008$ | $0.784 \pm 0.008$ |
| | MGraphDTA | $0.939 \pm 0.010$ | $0.914 \pm 0.014$ | $0.521 \pm 0.025$ | $0.939 \pm 0.010$ | $0.937 \pm 0.013$ | $0.945 \pm 0.010$ | $0.941 \pm 0.009$ | $0.960 \pm 0.004$ | $0.936 \pm 0.007$ | $0.568 \pm 0.016$ | $0.959 \pm 0.004$ | $0.950 \pm 0.007$ | $0.969 \pm 0.006$ | $0.959 \pm 0.004$ | $0.802 \pm 0.006$ | $0.737 \pm 0.007$ | $0.186 \pm 0.011$ | $0.802 \pm 0.006$ | $0.785 \pm 0.010$ | $0.831 \pm 0.010$ | $0.807 \pm 0.005$ |
| | SAGDTA | $0.905 \pm 0.005$ | $0.875 \pm 0.009$ | $0.461 \pm 0.024$ | $0.905 \pm 0.005$ | $0.916 \pm 0.014$ | $0.898 \pm 0.016$ | $0.907 \pm 0.005$ | $0.936 \pm 0.008$ | $0.902 \pm 0.015$ | $0.509 \pm 0.031$ | $0.935 \pm 0.008$ | $0.926 \pm 0.017$ | $0.945 \pm 0.011$ | $0.935 \pm 0.008$ | $0.758 \pm 0.008$ | $0.692 \pm 0.010$ | $0.151 \pm 0.012$ | $0.758 \pm 0.008$ | $0.747 \pm 0.016$ | $0.779 \pm 0.015$ | $0.762 \pm 0.005$ |
| | EmbedDTI | | | | | | | | | | | | | | | $0.755 \pm 0.004$ | $0.686 \pm 0.006$ | $0.139 \pm 0.009$ | $0.755 \pm 0.004$ | $0.733 \pm 0.013$ | $0.802 \pm 0.019$ | $0.765 \pm 0.003$ |
| | DeepGLSTM | $0.918 \pm 0.006$ | $0.887 \pm 0.008$ | $0.478 \pm 0.018$ | $0.918 \pm 0.006$ | $0.918 \pm 0.009$ | $0.924 \pm 0.009$ | $0.921 \pm 0.005$ | $0.938 \pm 0.006$ | $0.902 \pm 0.008$ | $0.507 \pm 0.012$ | $0.938 \pm 0.006$ | $0.920 \pm 0.007$ | $0.956 \pm 0.007$ | $0.938 \pm 0.006$ | $0.746 \pm 0.006$ | $0.680 \pm 0.006$ | $0.140 \pm 0.005$ | $0.746 \pm 0.006$ | $0.734 \pm 0.007$ | $0.769 \pm 0.003$ | $0.751 \pm 0.005$ |
| | CPI | $0.911 \pm 0.007$ | $0.878 \pm 0.011$ | $0.452 \pm 0.031$ | $0.911 \pm 0.007$ | $0.910 \pm 0.015$ | $0.918 \pm 0.019$ | $0.914 \pm 0.007$ | $0.928 \pm 0.007$ | $0.893 \pm 0.010$ | $0.500 \pm 0.016$ | $0.928 \pm 0.007$ | $0.912 \pm 0.010$ | $0.932 \pm 0.011$ | $0.927 \pm 0.007$ | $0.678 \pm 0.072$ | $0.622 \pm 0.061$ | $0.106 \pm 0.033$ | $0.678 \pm 0.072$ | $0.666 \pm 0.064$ | $0.712 \pm 0.097$ | $0.687 \pm 0.074$ |
| | BACPI | $0.928 \pm 0.007$ | $0.900 \pm 0.009$ | $0.498 \pm 0.015$ | $0.928 \pm 0.007$ | $0.927 \pm 0.008$ | $0.935 \pm 0.011$ | $0.931 \pm 0.007$ | $0.952 \pm 0.003$ | $0.926 \pm 0.007$ | $0.554 \pm 0.018$ | $0.952 \pm 0.004$ | $0.946 \pm 0.009$ | $0.957 \pm 0.005$ | $0.951 \pm 0.003$ | $0.776 \pm 0.009$ | $0.710 \pm 0.010$ | $0.163 \pm 0.012$ | $0.776 \pm 0.009$ | $0.762 \pm 0.013$ | $0.803 \pm 0.016$ | $0.782 \pm 0.008$ |
| | DeepNC-HGC | $0.874 \pm 0.005$ | $0.831 \pm 0.007$ | $0.319 \pm 0.022$ | $0.874 \pm 0.005$ | $0.870 \pm 0.008$ | $0.889 \pm 0.006$ | $0.879 \pm 0.005$ | $0.934 \pm 0.004$ | $0.896 \pm 0.010$ | $0.499 \pm 0.017$ | $0.934 \pm 0.004$ | $0.915 \pm 0.013$ | $0.954 \pm 0.009$ | $0.934 \pm 0.004$ | $0.758 \pm 0.008$ | $0.691 \pm 0.010$ | $0.147 \pm 0.009$ | $0.758 \pm 0.008$ | $0.742 \pm 0.011$ | $0.788 \pm 0.011$ | $0.765 \pm 0.007$ |
| | DeepNC-GEN | $0.919 \pm 0.004$ | $0.891 \pm 0.006$ | $0.484 \pm 0.020$ | $0.919 \pm 0.004$ | $0.924 \pm 0.012$ | $0.919 \pm 0.018$ | $0.921 \pm 0.005$ | $0.941 \pm 0.006$ | $0.909 \pm 0.010$ | $0.523 \pm 0.019$ | $0.941 \pm 0.006$ | $0.932 \pm 0.012$ | $0.951 \pm 0.009$ | $0.941 \pm 0.006$ | $0.742 \pm 0.008$ | $0.673 \pm 0.009$ | $0.130 \pm 0.010$ | $0.742 \pm 0.008$ | $0.719 \pm 0.015$ | $0.792 \pm 0.018$ | $0.754 \pm 0.006$ |
| | DrugBAN | $0.935 \pm 0.004$ | $0.908 \pm 0.007$ | $0.510 \pm 0.015$ | $0.935 \pm 0.004$ | $0.931 \pm 0.011$ | $0.944 \pm 0.014$ | $0.938 \pm 0.004$ | $0.961 \pm 0.003$ | $0.942 \pm 0.007$ | $0.595 \pm 0.027$ | $0.961 \pm 0.003$ | $0.961 \pm 0.010$ | $0.959 \pm 0.007$ | $0.960 \pm 0.003$ | $0.804 \pm 0.005$ | $0.742 \pm 0.008$ | $0.200 \pm 0.014$ | $0.804 \pm 0.006$ | $0.797 \pm 0.012$ | $0.815 \pm 0.010$ | $0.806 \pm 0.004$ |
| | GANDTI | $0.933 \pm 0.006$ | $0.910 \pm 0.007$ | $0.521 \pm 0.012$ | $0.932 \pm 0.006$ | $0.942 \pm 0.006$ | $0.926 \pm 0.010$ | $0.934 \pm 0.006$ | $0.944 \pm 0.005$ | $0.913 \pm 0.010$ | $0.528 \pm 0.014$ | $0.944 \pm 0.005$ | $0.933 \pm 0.013$ | $0.953 \pm 0.010$ | $0.943 \pm 0.006$ | $0.757 \pm 0.007$ | $0.690 \pm 0.008$ | $0.138 \pm 0.008$ | $0.757 \pm 0.007$ | $0.742 \pm 0.010$ | $0.778 \pm 0.012$ | $0.759 \pm 0.005$ |
| | BridgeDPI | $0.948 \pm 0.003$ | $0.930 \pm 0.004$ | $0.557 \pm 0.016$ | $0.948 \pm 0.004$ | $0.954 \pm 0.008$ | $0.945 \pm 0.013$ | $0.949 \pm 0.003$ | $0.968 \pm 0.006$ | $0.948 \pm 0.012$ | $0.622 \pm 0.060$ | $0.969 \pm 0.001$ | $0.973 \pm 0.005$ | $0.963 \pm 0.005$ | $0.968 \pm 0.001$ | $0.785 \pm 0.007$ | $0.719 \pm 0.011$ | $0.171 \pm 0.019$ | $0.785 \pm 0.007$ | $0.769 \pm 0.020$ | $0.813 \pm 0.023$ | $0.790 \pm 0.003$ |
| | ColdDTA | $0.937 \pm 0.006$ | $0.916 \pm 0.008$ | $0.530 \pm 0.016$ | $0.937 \pm 0.006$ | $0.945 \pm 0.007$ | $0.932 \pm 0.008$ | $0.938 \pm 0.006$ | $0.959 \pm 0.004$ | $0.937 \pm 0.009$ | $0.573 \pm 0.004$ | $0.959 \pm 0.010$ | $0.968 \pm 0.007$ | $0.950 \pm 0.004$ | $0.959 \pm 0.004$ | $0.767 \pm 0.007$ | $0.698 \pm 0.009$ | $0.141 \pm 0.006$ | $0.753 \pm 0.008$ | $0.209 \pm 0.017$ | $0.815 \pm 0.006$ | $0.818 \pm 0.005$ |
| | SubMDTA | $0.916 \pm 0.014$ | $0.892 \pm 0.016$ | $0.496 \pm 0.023$ | $0.916 \pm 0.015$ | $0.934 \pm 0.012$ | $0.901 \pm 0.027$ | $0.917 \pm 0.015$ | $0.921 \pm 0.026$ | $0.880 \pm 0.048$ | $0.445 \pm 0.147$ | $0.920 \pm 0.026$ | $0.906 \pm 0.059$ | $0.934 \pm 0.022$ | $0.921 \pm 0.023$ | $0.787 \pm 0.007$ | $0.723 \pm 0.009$ | $0.177 \pm 0.012$ | $0.787 \pm 0.007$ | $0.777 \pm 0.012$ | $0.805 \pm 0.011$ | $0.790 \pm 0.006$ |
| | IMAEN | $0.888 \pm 0.009$ | $0.849 \pm 0.007$ | $0.377 \pm 0.032$ | $0.888 \pm 0.009$ | $0.887 \pm 0.009$ | $0.897 \pm 0.026$ | $0.892 \pm 0.011$ | $0.904 \pm 0.025$ | $0.864 \pm 0.028$ | $0.453 \pm 0.044$ | $0.904 \pm 0.025$ | $0.894 \pm 0.018$ | $0.902 \pm 0.040$ | $0.902 \pm 0.027$ | $0.761 \pm 0.026$ | $0.694 \pm 0.030$ | $0.150 \pm 0.029$ | $0.761 \pm 0.027$ | $0.742 \pm 0.044$ | $0.806 \pm 0.029$ | $0.772 \pm 0.013$ |
| Transformer | CSDTI | $0.848 \pm 0.009$ | $0.800 \pm 0.012$ | $0.260 \pm 0.029$ | $0.848 \pm 0.008$ | $0.844 \pm 0.015$ | $0.865 \pm 0.013$ | $0.854 \pm 0.007$ | $0.873 \pm 0.006$ | $0.816 \pm 0.005$ | $0.306 \pm 0.011$ | $0.873 \pm 0.006$ | $0.854 \pm 0.005$ | $0.894 \pm 0.017$ | $0.874 \pm 0.007$ | $0.727 \pm 0.006$ | $0.661 \pm 0.006$ | $0.128 \pm 0.005$ | $0.727 \pm 0.006$ | $0.715 \pm 0.008$ | $0.752 \pm 0.007$ | $0.733 \pm 0.004$ |
| | TDGraphDTA | $0.940 \pm 0.009$ | $0.916 \pm 0.013$ | $0.526 \pm 0.025$ | $0.941 \pm 0.008$ | $0.939 \pm 0.014$ | $0.946 \pm 0.007$ | $0.943 \pm 0.008$ | $0.959 \pm 0.004$ | $0.935 \pm 0.009$ | $0.565 \pm 0.021$ | $0.959 \pm 0.004$ | $0.948 \pm 0.007$ | $0.971 \pm 0.006$ | $0.959 \pm 0.004$ | $0.815 \pm 0.007$ | $0.755 \pm 0.009$ | $0.219 \pm 0.018$ | $0.815 \pm 0.007$ | $0.811 \pm 0.012$ | $0.820 \pm 0.012$ | $0.815 \pm 0.006$ |
| | AMMVF | $0.928 \pm 0.005$ | $0.899 \pm 0.006$ | $0.496 \pm 0.012$ | $0.929 \pm 0.005$ | $0.924 \pm 0.012$ | $0.938 \pm 0.019$ | $0.931 \pm 0.006$ | $0.962 \pm 0.003$ | $0.943 \pm 0.004$ | $0.602 \pm 0.018$ | $0.962 \pm 0.003$ | $0.961 \pm 0.003$ | $0.963 \pm 0.005$ | $0.962 \pm 0.003$ | $0.709 \pm 0.003$ | $0.645 \pm 0.022$ | $0.118 \pm 0.017$ | $0.709 \pm 0.023$ | $0.697 \pm 0.026$ | $0.738 \pm 0.025$ | $0.717 \pm 0.021$ |
| | IIFDTI | $0.918 \pm 0.016$ | $0.891 \pm 0.016$ | $0.541 \pm 0.044$ | $0.938 \pm 0.007$ | $0.946 \pm 0.007$ | $0.935 \pm 0.012$ | $0.939 \pm 0.006$ | $0.966 \pm 0.004$ | $0.949 \pm 0.007$ | $0.612 \pm 0.021$ | $0.965 \pm 0.004$ | $0.966 \pm 0.006$ | $0.966 \pm 0.004$ | $0.966 \pm 0.004$ | $0.791 \pm 0.010$ | $0.728 \pm 0.010$ | $0.180 \pm 0.009$ | $0.791 \pm 0.010$ | $0.780 \pm 0.010$ | $0.812 \pm 0.020$ | $0.794 \pm 0.012$ |
| | ICAN | $0.938 \pm 0.005$ | $0.915 \pm 0.005$ | $0.527 \pm 0.008$ | $0.937 \pm 0.005$ | $0.943 \pm 0.010$ | $0.935 \pm 0.012$ | $0.939 \pm 0.006$ | $0.956 \pm 0.003$ | $0.933 \pm 0.005$ | $0.558 \pm 0.006$ | $0.956 \pm 0.003$ | $0.961 \pm 0.008$ | $0.764 \pm 0.005$ | $0.699 \pm 0.005$ | $0.764 \pm 0.005$ | $0.699 \pm 0.005$ | $0.156 \pm 0.006$ | $0.764 \pm 0.005$ | $0.754 \pm 0.008$ | $0.781 \pm 0.008$ | $0.766 \pm 0.004$ |
| | MolTrans | $0.937 \pm 0.006$ | $0.916 \pm 0.006$ | $0.530 \pm 0.017$ | $0.937 \pm 0.006$ | $0.944 \pm 0.011$ | $0.934 \pm 0.013$ | $0.939 \pm 0.006$ | $0.958 \pm 0.003$ | $0.935 \pm 0.005$ | $0.582 \pm 0.007$ | $0.958 \pm 0.004$ | $0.956 \pm 0.008$ | $0.957 \pm 0.004$ | $0.958 \pm 0.003$ | $0.796 \pm 0.001$ | $0.735 \pm 0.001$ | $0.189 \pm 0.008$ | $0.796 \pm 0.001$ | $0.786 \pm 0.007$ | $0.809 \pm 0.010$ | $0.797 \pm 0.002$ |
| | TransformerCPI | $0.920 \pm 0.004$ | $0.887 \pm 0.007$ | $0.470 \pm 0.022$ | $0.920 \pm 0.004$ | $0.924 \pm 0.005$ | $0.916 \pm 0.004$ | $0.920 \pm 0.003$ | $0.959 \pm 0.004$ | $0.941 \pm 0.017$ | $0.576 \pm 0.030$ | $0.960 \pm 0.004$ | $0.967 \pm 0.007$ | $0.950 \pm 0.004$ | $0.959 \pm 0.003$ | $0.787 \pm 0.011$ | $0.720 \pm 0.012$ | $0.165 \pm 0.011$ | $0.787 \pm 0.011$ | $0.773 \pm 0.011$ | $0.822 \pm 0.018$ | $0.784 \pm 0.010$ |
| | MRBDTA | $0.914 \pm 0.009$ | $0.886 \pm 0.008$ | $0.479 \pm 0.017$ | $0.914 \pm 0.009$ | $0.907 \pm 0.008$ | $0.925 \pm 0.005$ | $0.915 \pm 0.008$ | $0.847 \pm 0.197$ | $0.828 \pm 0.192$ | $0.533 \pm 0.194$ | $0.882 \pm 0.100$ | $0.731 \pm 0.415$ | $0.745 \pm 0.418$ | $0.723 \pm 0.419$ | $0.801 \pm 0.009$ | $0.739 \pm 0.010$ | $0.183 \pm 0.011$ | $0.797 \pm 0.006$ | $0.785 \pm 0.011$ | $0.817 \pm 0.012$ | $0.790 \pm 0.006$ |
| | FOTFCPI | $0.941 \pm 0.004$ | $0.919 \pm 0.011$ | $0.536 \pm 0.031$ | $0.941 \pm 0.003$ | $0.944 \pm 0.019$ | $0.942 \pm 0.019$ | $0.943 \pm 0.003$ | $0.966 \pm 0.004$ | $0.948 \pm 0.009$ | $0.608 \pm 0.023$ | $0.965 \pm 0.004$ | $0.964 \pm 0.010$ | $0.966 \pm 0.002$ | $0.965 \pm 0.004$ | $0.798 \pm 0.008$ | $0.737 \pm 0.008$ | $0.184 \pm 0.010$ | $0.797 \pm 0.008$ | $0.792 \pm 0.012$ | $0.809 \pm 0.012$ | $0.800 \pm 0.007$ |
| | Our combos | $0.950 \pm 0.007$ | $0.930 \pm 0.013$ | $0.554 \pm 0.035$ | $0.950 \pm 0.007$ | $0.950 \pm 0.016$ | $0.954 \pm 0.011$ | $0.959 \pm 0.007$ | $0.966 \pm 0.001$ | $0.947 \pm 0.005$ | $0.601 \pm 0.027$ | $0.966 \pm 0.001$ | $0.962 \pm 0.010$ | $0.969 \pm 0.010$ | $0.965 \pm 0.001$ | $0.791 \pm 0.006$ | $0.723 \pm 0.008$ | $0.169 \pm 0.012$ | $0.791 \pm 0.006$ | $0.769 \pm 0.013$ | $0.831 \pm 0.010$ | $0.799 \pm 0.004$ |

## J Memory and Parameter Comparison

Table 13: Training time per epoch (s) and the max allocated memory (MB) for representative dataset on both regression (Davis) and classification (Human) tasks when BS is 32.

| Categories | Models | Regression | | | Classification | | |
|---|---|---|---|---|---|---|---|
| | | Model parameter | Memory Usage (MB) | Time(s) | Model parameter | Memory Usage (MB) | Run Time (s) |
| Graph | GraphDTA-GCN | 7.87 | 86.45 | 8.92 | 7.87 | 86.33 | 2.43 |
| | GraphDTA-GAT | 6.58 | 104.71 | 9.62 | 6.58 | 99.40 | 2.43 |
| | GraphDTA-GATGCN | 18.12 | 148.25 | 8.37 | 18.12 | 145.13 | 2.35 |
| | GraphDTA-GIN | 5.97 | 78.00 | 12.33 | 5.95 | 77.47 | 3.13 |
| | GraphCPI-GCN | 10.46 | 98.13 | 7.02 | 10.48 | 63.37 | 1.92 |
| | GraphCPI-GAT | 9.16 | 116.19 | 9.38 | 9.18 | 112.34 | 2.48 |
| | GraphCPI-GATGCN | 20.70 | 158.21 | 9.47 | 20.73 | 156.22 | 2.20 |
| | GraphCPI-GIN | 8.55 | 88.55 | 12.54 | 8.56 | 88.02 | 2.92 |
| | MGraphDTA | 11.75 | 235.97 | 69.84 | 11.43 | 217.15 | 17.59 |
| | SAGDTA | 7.45 | 88.31 | 20.87 | 7.44 | 87.54 | 4.34 |
| | EmbedDTI | 16.97 | 152.55 | 17.80 | 16.97 | - | - |
| | DeepGLSTM | 131.92 | 1287.92 | 20.69 | 131.93 | 1287.16 | 11.22 |
| | CPI | 0.37 | 14.00 | 11.29 | 0.6 | 14.82 | 2.69 |
| | BACPI | 4.05 | 1051.91 | 43.27 | 6.13 | 1058.95 | 12.38 |
| | DeepNC-HGC | 16.61 | 123.70 | 9.85 | 16.60 | 123.65 | 3.46 |
| | DeepNC-GEN | 18.84 | 174.00 | 11.35 | 18.84 | 166.55 | 3.46 |
| | DrugBAN | 4.10 | 940.22 | 30.06 | 4.10 | 940.23 | 7.84 |
| | GANDTI | 1.48 | 35.89 | 6.01 | 2.43 | 39.95 | 1.54 |
| | PGraphDTA-CNN | 9.03 | 102.85 | 13.71 | 9.03 | - | - |
| | BridgeDPI | 39.32 | 232.53 | 16.27 | 39.32 | 232.53 | 4.36 |
| | ColdDTA | 13.14 | 282.74 | 72.98 | 13.14 | 262.91 | 18.56 |
| | SubMDTA | 169.37 | 992.61 | 35.12 | 195.50 | 1095.73 | 8.49 |
| | IMAEN | 10.43 | 174.34 | 35.77 | 10.43 | 172.86 | 4.41 |
| Transformer | CSDTI | 9.67 | 281.23 | 17.66 | 9.66 | 281.02 | 4.35 |
| | TDGraphDTA | 8.62 | 247.23 | 116.38 | 8.62 | 236.02 | 28.43 |
| | AMMVF | 6.68 | 17847.62 | 216.20 | 7.49 | 17850.79 | 57.99 |
| | IIFDTI | 10.75 | 7946.92 | 141.12 | 10.75 | 11890.79 | 56.95 |
| | ICAN | 63.89 | 649.55 | 12.44 | 63.89 | 648.67 | 2.84 |
| | MolTrans | 239.73 | 10624.55 | 70.19 | 239.74 | 10624.55 | 25.06 |
| | TransformerCPI | 4.44 | 1219.58 | 28.98 | 4.45 | 1219.60 | 7.17 |
| | MRBDTA | 17.83 | 3893.76 | 66.47 | 17.84 | 3893.78 | 16.13 |
| | FOTFCPI | 189.15 | 6780.35 | 58.75 | 189.15 | 6780.35 | 14.80 |
| | Our | 19.02 | 1081.99 | 94.71 | 19.02 | 1082.68 | 13.68 |

