# OpenReview forum: "Benchmark on Drug Target Interaction Modeling from a Drug Structure Perspective"
_TMLR — Rejected by TMLR_

### Review · Reviewer_Cdht · 2025-01-29

**Summary Of Contributions:**

This paper studies the problem of drug-target interaction modeling. It focuses on two types of structure learning algorithms: GNN-based methods and Transformer-based methods. The authors choose two methods from each family of algorithms and perform a hyperparameter search to determine the best possible hyperparameter values for classification and regression tasks. A comparison is then performed between these two families of algorithms, while the impact of features from chemical and physical properties of molecules on the performance of the methods is also investigated. Then a lot of methods from the two families of algorithms are evaluated on 6 datasets in the tasks of binary interaction classification and affinity regression.

**Audience:**

Yes

**Broader Impact Concerns:**

--

**Claims And Evidence:**

Yes

**Requested Changes:**

The proposed adjustments are directly related to the weaknesses listed above.

- Optimize the hyperparameter values of each one of the compared methods on the 6 datasets.
- Fix presentation issues.
- Explain why section 4 was added to the paper, and replace the GCN encoder with GIN.
- Explain why the performance of methods from the same family in Tables 2 and 3 varies that much.
- Make clear whether the proposed combos are novel.

**Strengths And Weaknesses:**

Strengths

- Some of the findings of this paper could be of interest to practitioners in both industry and academia. For example, the results presented in Tables 2 and 3 could help practitioners choose a method for their specific application.

- The experimental evaluation in section 5 is very extensive since many different methods are compared against each other. In addition, the comparison is not limited to only a single family of methods, but methods from two different families are compared against each other.


Weaknesses

- Some of the findings that are presented in the paper are already known. For example, it is already known that the space complexity of Transformer-based methods is higher than that of GNN-based methods since the space complexity of the self-attention mechanism is quadratic in the number of tokens.

- The empirical evaluation is not carefully designed since the hyperparameter values of most methods are set equal to the optimal hyperparameter values of other models. For a fair comparison, I would recommend the authors optimize the hyperparameter values of each method that is included in the comparison.

- In my understanding there is no direct connection between the results that are presented in section 4 and those presented in section 5. It seems to me that section 4 is disconnected from the rest of the paper. Please explain why this section is added in the paper.

- In section 4, it is not clear why the authors chose GCN to serve as the drug encoder from the family of GNNs. GCN is known to be less expressive than 1-WL [1]. Furthermore, it is well-known that there exist non-isomorphic chemical compounds which cannot be distinguished even by 1-WL. I would suggest the authors to employ the GIN model instead which has the same time complexity as GCN and can be as expressive as 1-WL.

- While the paper is easy to follow, it seems like a first draft that needs to go through a couple of revision cycles. For example, the paper's title is the default title of the TMLR template. Furthermore, all Figures are hard to read because they are quite small while the font size is too small to be readable. I had to zoom in a lot to actually read Figure 1. In addition, the proposed combos are included in the comparisons of Tables 2 and 3, but these methods are presented later on in the text. These issues should be fixed in the revision.

- In Tables 2 and 3, we can see that there is quite a lot of variance in the performance of different GNN-based methods and Transformer-based methods. For example, on DAVIS, the MSEs of the GNN-based methods vary between 0.3623 and 0.2179, while those of Transformer-based methods between 0.3481 and 0.2063. What is the reason behind that?

- From the presentation of the paper, it appears that one of the contributions of the paper is the introduction of the combos presented in subsection 5.1. But are those combos actually novel? The attention mechanism of Equation 6 was first proposed in [2]. I would suggest the authors make this clear in the revision.

[1] Xu, K., Hu, W., Leskovec, J., & Jegelka, S. (2018). How powerful are graph neural networks?. ICLR'19.\
[2] Maziarka, Ł., Danel, T., Mucha, S., Rataj, K., Tabor, J., & Jastrzębski, S. (2020). Molecule attention transformer. arXiv:2002.08264.

---

> ### Author Response · Authors · 2025-03-05
> **Reply 0-4**
>
> **Comment 0:** Some of the findings that are presented in the paper are already known. For example, it is already known that the space complexity of Transformer-based methods is higher than that of GNN-based methods since the space complexity of the self-attention mechanism is quadratic in the number of tokens.
>
> **Reply 0:** We acknowledge that the higher space complexity of Transformer-based methods compared to GNN-based methods is a well-known fact due to the quadratic complexity of the self-attention mechanism. However, our study aims to provide empirical study in drug-protein interaction tasks. By comparing different methods, we seek to highlight trade-offs between accuracy and computational efficiency in DTI task. More importantly, these findings motive us to design new approache that achieve the better trade-off.
>
> **Comment 1:** The empirical evaluation is not carefully designed since the hyperparameter values of most methods are set equal to the optimal hyperparameter values of other models. For a fair comparison, I would recommend the authors optimize the hyperparameter values of each method that is included in the comparison.
>
> **Reply 1:** Thanks for the advice! To find the optimal parameters suitable for each model, we started with the hyperparameters from the original article/codebase and made some adjustments. We have started reruning all the experiments.
>
> **Comment 2:** In my understanding there is no direct connection between the results that are presented in section 4 and those presented in section 5. It seems to me that section 4 is disconnected from the rest of the paper. Please explain why this section is added in the paper.
>
> **Reply 2:** Section 4 explores various factors that may impact model performance, serving as both an introduction to Section 5 and an explanation for why certain models achieve better results. Specifically, in Section 4, we analyze different encoders for drugs and proteins, as well as different node features when modeling molecular properties, providing a microscopic perspective. The molecular properties considered are summarized from 26 methods, covering atomic properties, hydrogen information, electron properties, stereochemistry, and structural information. This section helps us understand how different encoders and features contribute to model performance.
>
> In contrast, Section 5 provides a macroscopic evaluation by comparing the 26 methods across six datasets, systematically assessing their effectiveness. The insights gained from Section 4—such as the choice of encoder structures and essential molecular features—inform the design of our proposed model combinations in Section 5. Thus, Section 4 is not an isolated component but a necessary foundation for understanding and refining the comparative analysis in Section 5.
>
> **Comment 3:** In section 4, it is not clear why the authors chose GCN to serve as the drug encoder from the family of GNNs. GCN is known to be less expressive than 1-WL [1]. Furthermore, it is well-known that there exist non-isomorphic chemical compounds which cannot be distinguished even by 1-WL. I would suggest the authors to employ the GIN model instead which has the same time complexity as GCN and can be as expressive as 1-WL.
>
> **Reply 3:** Thank you for your suggestion! We have incorporated GIN as the drug encoder with ealy stopping mechnism for a more detailed investigation. Please refer to Reply 10 in the second review for the initial experiment, and see the experiments on different features presented below.
>
> **Comment 4:** While the paper is easy to follow, it seems like a first draft that needs to go through a couple of revision cycles. For example, the paper's title is the default title of the TMLR template. Furthermore, all Figures are hard to read because they are quite small while the font size is too small to be readable. I had to zoom in a lot to actually read Figure 1. In addition, the proposed combos are included in the comparisons of Tables 2 and 3, but these methods are presented later on in the text. These issues should be fixed in the revision.
>
> **Reply 4:** Thanks for your suggestion! We have addressed these issues by updating the title, enlarging the figures for better readability, and improving the overall formatting. Regarding the organization, our methods are inspired by previous models, so we aim to include them in the benchmark comparison while maintaining a logical flow in the text.

---

> > ### Author Response · Authors · 2025-03-05
> > **Table for reply 3**
> >
> > | Models        | Initial Feature         | MSE ± std | R2 ± std | CI ± std | ROC-AUC ± std | Acc. ± std | F1 ± std |
> > |--------------|-------------------------|-----------|----------|----------|---------------|------------|----------|
> > | **GraphDTA-GIN** | Basic                  | $0.253 \pm 0.010$ | $0.623 \pm 0.015$ | $0.861 \pm 0.006$ | $0.891 \pm 0.020$ | $0.891 \pm 0.020$ | $0.894 \pm 0.019$ |
> > |              | Basic+AP                 | $0.244 \pm 0.010$ | $0.637 \pm 0.015$ | $0.866 \pm 0.007$ | $0.875 \pm 0.004$ | $0.875 \pm 0.004$ | $0.879 \pm 0.004$ |
> > |              | Basic+HI                 | $0.241 \pm 0.009$ | $0.641 \pm 0.014$ | $0.866 \pm 0.005$ | $0.884 \pm 0.010$ | $0.884 \pm 0.011$ | $0.887 \pm 0.011$ |
> > |              | Basic+EP                 | $0.260 \pm 0.020$ | $0.613 \pm 0.030$ | $0.855 \pm 0.010$ | $0.884 \pm 0.006$ | $0.884 \pm 0.006$ | $0.888 \pm 0.005$ |
> > |              | Basic+Ste                | $0.247 \pm 0.010$ | $0.633 \pm 0.015$ | $0.866 \pm 0.006$ | $0.890 \pm 0.005$ | $0.890 \pm 0.005$ | $0.894 \pm 0.005$ |
> > |              | Basic+Str                | $0.258 \pm 0.015$ | $0.615 \pm 0.023$ | $0.862 \pm 0.010$ | $0.881 \pm 0.008$ | $0.882 \pm 0.008$ | $0.887 \pm 0.007$ |
> > |              | Basic+AP+HI              | $0.249 \pm 0.012$ | $0.629 \pm 0.019$ | $0.866 \pm 0.004$ | $0.886 \pm 0.011$ | $0.886 \pm 0.011$ | $0.890 \pm 0.009$ |
> > |              | Basic+AP+HI+EP           | $0.251 \pm 0.011$ | $0.626 \pm 0.017$ | $0.861 \pm 0.005$ | $0.888 \pm 0.007$ | $0.888 \pm 0.007$ | $0.891 \pm 0.008$ |
> > |              | Basic+AP+HI+EP+Ste       | $0.245 \pm 0.009$ | $0.634 \pm 0.014$ | $0.867 \pm 0.004$ | $0.885 \pm 0.008$ | $0.885 \pm 0.008$ | $0.888 \pm 0.007$ |
> > |              | Basic+AP+HI+EP+Ste+Str   | $0.247 \pm 0.012$ | $0.632 \pm 0.018$ | $0.865 \pm 0.005$ | $0.885 \pm 0.010$ | $0.885 \pm 0.010$ | $0.889 \pm 0.009$ |
> > | **GraphCPI-GIN** | Basic                  | $0.274 \pm 0.009$ | $0.593 \pm 0.013$ | $0.851 \pm 0.008$ | $0.881 \pm 0.008$ | $0.881 \pm 0.007$ | $0.886 \pm 0.007$ |
> > |              | Basic+AP                 | $0.277 \pm 0.013$ | $0.588 \pm 0.020$ | $0.849 \pm 0.005$ | $0.883 \pm 0.015$ | $0.883 \pm 0.015$ | $0.888 \pm 0.013$ |
> > |              | Basic+HI                 | $0.290 \pm 0.042$ | $0.568 \pm 0.063$ | $0.850 \pm 0.010$ | $0.876 \pm 0.014$ | $0.876 \pm 0.014$ | $0.879 \pm 0.013$ |
> > |              | Basic+EP                 | $0.283 \pm 0.016$ | $0.578 \pm 0.023$ | $0.842 \pm 0.011$ | $0.882 \pm 0.010$ | $0.882 \pm 0.009$ | $0.885 \pm 0.009$ |
> > |              | Basic+Ste                | $0.285 \pm 0.014$ | $0.576 \pm 0.021$ | $0.848 \pm 0.005$ | $0.883 \pm 0.004$ | $0.882 \pm 0.004$ | $0.886 \pm 0.003$ |
> > |              | Basic+Str                | $0.278 \pm 0.004$ | $0.586 \pm 0.007$ | $0.848 \pm 0.004$ | $0.888 \pm 0.007$ | $0.888 \pm 0.008$ | $0.890 \pm 0.009$ |
> > |              | Basic+AP+HI              | $0.289 \pm 0.019$ | $0.568 \pm 0.029$ | $0.840 \pm 0.011$ | $0.879 \pm 0.012$ | $0.878 \pm 0.012$ | $0.881 \pm 0.014$ |
> > |              | Basic+AP+HI+EP           | $0.288 \pm 0.008$ | $0.571 \pm 0.013$ | $0.843 \pm 0.009$ | $0.876 \pm 0.005$ | $0.877 \pm 0.005$ | $0.882 \pm 0.006$ |
> > |              | Basic+AP+HI+EP+Ste       | $0.287 \pm 0.018$ | $0.573 \pm 0.026$ | $0.843 \pm 0.006$ | $0.886 \pm 0.009$ | $0.885 \pm 0.009$ | $0.888 \pm 0.009$ |
> > |              | Basic+AP+HI+EP+Ste+Str   | $0.294 \pm 0.023$ | $0.563 \pm 0.034$ | $0.840 \pm 0.014$ | $0.884 \pm 0.012$ | $0.884 \pm 0.013$ | $0.889 \pm 0.014$ |

---

> > > ### Author Response · Authors · 2025-03-05
> > > **Reply 5-6**
> > >
> > > **Comment 5:** In Tables 2 and 3, we can see that there is quite a lot of variance in the performance of different GNN-based methods and Transformer-based methods. For example, on DAVIS, the MSEs of the GNN-based methods vary between 0.3623 and 0.2179, while those of the Transformer-based methods vary between 0.3481 and 0.2063. What is the reason behind that?
> > >
> > > **Reply 5:** Thanks for the comments and findings! As discussed in the paper, even within the same category, different GNN-based methods, such as GCN and GIN, exhibit varying performance due to differences in their structure-learning capabilities. GCN operates by averaging neighboring node features, which may lead to oversmoothing and loss of fine-grained molecular details. In contrast, GIN is designed to be as powerful as the Weisfeiler-Lehman (WL) test in distinguishing graph structures, allowing it to better capture molecular fingerprints, which often results in superior performance. Similarly, other GNN methods, such as MgraphDTA, utilize multiple-scale information in graphs, which may enhance performance by capturing different-scale local structure information.
> > >
> > > For Transformer-based methods, performance differences primarily stem from how well they capture sequence dependencies and structural information. Unlike GNNs, which explicitly model molecular graphs, transformers rely on self-attention mechanisms to implicitly learn structural relationships from sequence-based representations. The choice of self-attention formulation, the number of attention heads, and the depth of the transformer layers all influence the model’s ability to extract meaningful molecular features. Additionally, differences in tokenization strategies significantly impact performance. MRBDTA treats SMILES strings as character sequences, while others like MolTrans use subword embeddings, which may better capture chemical substructures.
> > >
> > > Beyond architectural differences, featurization techniques and hyperparameter choices further impact model effectiveness. For example, node features in GNNs can be enriched with atomic-level properties (e.g., atom properties, electron, hybridization), while protein embeddings can be enhanced with evolutionary or structural information. The choice of these features, along with hyperparameters like learning rate, batch size, and dropout rate, influences generalization and convergence, leading to the observed performance variations.
> > >
> > > **Comment 6:** From the presentation of the paper, it appears that one of the contributions of the paper is the introduction of the combos presented in subsection 5.1. But are those combos actually novel? The attention mechanism of Equation 6 was first proposed in [2]. I would suggest the authors make this clear in the revision.
> > >
> > > **Reply 6:** Thanks for your suggestion! **Novelty of module combos**:
> > > i) While many different encoders and featuriazation methods has been proposed in the area of drug-target interaction predictionm, it is still unclear about their disentangled contribution due to the lack of fair benchmark platform. Without isolating the effects of diverse techniques, one might never reach convincing answers whether the drug-target interaction modeling with ceteris paribus should perform better.
> > >
> > > Additionally, our microscopic exploration reveals that more features do not always lead to better performance in GNN-based drug modeling. For example, atomic properties and hydrogen information positively contribute to predictive accuracy, while features such as electron properties surprisingly degrade DTI model performance. These findings provide valuable insights for future research, guiding more informed decisions on feature selection and improving molecular representation strategies in drug-target interaction modeling.
> > >
> > > ii) Based on the benchmark analysis on both effectiveness and efficiency, we are thus motivated to summarize the powerful trick combos and shred novel insight into the design philosophy of the interaction prediction model. As shown in the paper, these combos achieve the better accuracy-efficiency trade-off in the commonly used datsets. It is expected that this design philosophy faciliate the model development for data scientist based on their downstream applications on hand.
> > >
> > >
> > > We will clarify the novelty of the proposed combos in the revision and explicitly acknowledge prior work.

---

### Review · Reviewer_kMp2 · 2025-02-07

**Summary Of Contributions:**

The paper provides a comprehensive evaluation of drug-target interaction prediction models. The first part of the evaluation focuses on comparing various encoder architectures used to generate protein and drug embeddings. The second part assesses 31 models across six datasets. The authors also introduce their combination of encoders, which surpasses other methods on two regression tasks and achieves competitive results on the classification tasks. The proposed architecture integrates a graph transformer and a SMILES CNN for embedding drugs with a CNN protein sequence encoder. The main contributions of this work are as follows:
1) many DTI prediction models are thoroughly benchmarked across six datasets,
2) nine observations based on the experimental results are presented, which can be used as a recommendation for building better DTI models,
3) a new combination of molecule processing architectures is proposed.

**Audience:**

Yes

**Broader Impact Concerns:**

This paper benchmarks DTI prediction models and proposes a new architecture. Therefore, I believe that there are no significant ethical implications to consider besides those related to enhancing machine learning models used in drug discovery.

**Claims And Evidence:**

No

**Requested Changes:**

Please refer to the weaknesses above. All comments must be addressed for me to recommend acceptance (my comments focus on the current results and do not suggest new experiments).

Minor comments:
1. On page 3, one of the references is not cited correctly ("tsubaki2019compound" is not correctly rendered).
2. The title inside the paper should be fixed.

**Strengths And Weaknesses:**

Strengths:
1. Different types of molecule encoders are introduced in Section 2.
2. The benchmarking experiment includes more than 30 models and six datasets that represent both regression and classification tasks.
3. The conclusions derived from the experiments are clearly presented as separate observations. They can be used to improve model architectures in DTI prediction.
4. The exploration of graph featurization is an interesting experiment that leads to the conclusion that not always including all the atom features is beneficial.
5. A new combination of molecular encoders is proposed. Drugs are encoded by a graph transformer and a SMILES CNN, and proteins are encoded with a CNN. This architecture is effective in both regression and classification tasks presented in the benchmark.

Weaknesses:
1. The title does not accurately describe the content of the paper. Why is the structural perspective emphasized if there is no discussion about the protein structure? Most of the evaluated methods only use protein sequences to make predictions. Also, the title inside the paper is the template title instead of the article title.
2. I do not understand why GNNs are called explicit and transformers are called implicit in the abstract.
3. Similarly, I do not understand why some experiments are called macroscopical and others are called microscopical. Especially since the choice of encoder architecture seems more microscopical than the evaluation of the model as a whole.
4. Some statements in the introduction should be clarified. For example, it is not true that deep learning enables "more accurate and efficient predictions compared with laboratory experimental methods." The text should not suggest that deep learning methods can replace laboratory experiments as more accurate but rather accelerate the discovery of new compounds by more accurate prediction in combination with laboratory experimentation.
5. The introduction states that in DTI, protein sequences are "processed by separate convolutional neural networks," but there exist other ways to create protein embeddings, e.g. by using RNNs or transformers. This sentence should be rephrased.
6. In Section 2, graph transformers are not discussed as a way of encoding molecules. These methods should be briefly discussed, especially since one of the graph transformer architectures is used in the proposed encoder combination.
7. Section 2.3 mentions that extra molecular properties, e.g. solubility, are crucial in DTI, but they are not discussed in the remainder of this section. All the featurizers presented in this section are based on the molecular structure and not additional molecular properties.
8. In the caption of Figure 1, the meaning of the numbers in the plot is not explained. The text in the plot is too small and not readable. Ideally, the meaning of the numbers should be also indicated in the plot itself.
9. Why are 1000 epochs used for GNNs and only 300 for transformers? I expect transformers to take more epochs to train due to a higher number of trainable parameters.
10. It should be explicitly stated what datasets are used in Table 1.
11. Table 1 shows the results only on one regression and one classification dataset. Thus, the sample size is too small to conclude that "drug features extracted by the Transformer generally perform better than those by GNNs in regression tasks."
12. If results in Section 5 are averaged by five-fold cross-validation, it would be recommended to include the standard deviation of the results. This way, the readers can easily estimate if the difference between models is significant.
13. Figure 3 is not readable. The text is too small, and axis tick labels are overlapping. Please consider rotating the plot and putting model names directly on the y-axis (assuming horizontal bars after rotation).
14. I am unsure if I can agree with observation 9. All models seem to be stable, judging by Figure 4, but some models take longer to train. Also, these plots suggest that 300 epochs are not enough for some of these models to converge fully, which may impact the other results.

---

> ### Author Response · Authors · 2025-03-05
> **Reply 1-4**
>
> **Comment 0:** The title does not accurately describe the content of the paper. Why is the structural perspective emphasized if there is no discussion about the protein structure? Most of the evaluated methods only use protein sequences to make predictions. Also, the title inside the paper is the template title instead of the article title.
>
> **Reply 0:** The "structure" in the title refers that we focus on the GNN and Transformer based methods, which explicitly or implicitly leverge the structure of drug molecules into consideration. While GNNs explicitly learn structural information by passing information along chemical bonds, transformer-based models implicitly learn atom interactions with attention. Considering the protein information, most of the drug-target interaction prediction models leverage the primary sequences to estimate the interaction score, and thus we follow this common setting.
>
>
> As stated in start of Section 2, we focus on the formulations of recently-emerging structure modeling approaches for drug molecules. We change the title name to **Benchmark on Drug Target Interaction Modeling from a Drug Structure Perspective** and appreciate your comment. We have also corrected the title format, many thanks!
>
> **Comment 1:** I do not understand why GNNs are called explicit and transformers are called implicit in the abstract.
>
> **Reply 1:** From our perspective, GNNs explicitly model molecular structures by representing atoms as nodes and chemical bonds as edges, preserving the inherent graph nature of molecules. The message-passing mechanism in GNNs propagates and aggregates information across the molecular graph, explicitly capturing atomic connectivity and local structural relationships. In contrast, transformer-based methods treat molecular representations as sequential tokenized inputs (e.g., SMILES strings or learned embeddings) and rely on the self-attention mechanism to implicitly capture structural patterns and long-range dependencies without explicitly modeling atomic connectivity. This distinction leads to GNNs being considered explicit in their structural modeling, and transformers extract structure-related information in an implicit manner through learned attention weights.
>
> **Comment 2:** Similarly, I do not understand why some experiments are called macroscopical and others are called microscopical. Especially since the choice of encoder architecture seems more microscopical than the evaluation of the model as a whole.
>
> **Reply 2:** For the macroscopical experiment, we consider encoder as a coarse-grained control variable and conduct experiments for each category of encoder by running a set of representative models. Since we only consider a limited number of representative models, it is kind of more macroscopical than the later section, where we have a detailed and complete experiments on all the models we selected. Thanks for the advice and feel free to provide any suggestions that can help us make the subtitles less confusing.
>
> **Comment 3:** Some statements in the introduction should be clarified. For example, it is not true that deep learning enables "more accurate and efficient predictions compared with laboratory experimental methods." The text should not suggest that deep learning methods can replace laboratory experiments as more accurate but rather accelerate the discovery of new compounds by more accurate prediction in combination with laboratory experimentation.
>
> **Reply 3:** Corrected. Thank you!
>
> Deep learning-based frameworks have recently revolutionized this field, enabling more accurate predictions and accelerating the discovery of new compounds by guiding laboratory experiments more efficiently.
>
> **Comment 4:** The introduction states that in DTI, protein sequences are "processed by separate convolutional neural networks," but there exist other ways to create protein embeddings, e.g. by using RNNs or transformers. This sentence should be rephrased.
>
> **Reply 4:** Corrected. Thank you!
>
> Within deep learning frameworks \citep{ozturk2018deepdta,ozturk2019widedta}, drugs are commonly represented using the Simplified Molecular Input Line Entry System (SMILES) \citep{Smiles}, and proteins are represented as sequences of amino acids. These representations are typically processed using various neural network architectures, such as convolutional neural networks (CNNs)~\citep{krizhevsky2017imagenet, he2016deep}, recurrent neural networks (RNNs), transformers and so on, before being integrated and processed by a multi-layer perceptron (MLP) for DTI prediction.

---

> ### Author Response · Authors · 2025-03-05
> **Reply 5-9**
>
> **Comment 5:** In Section 2, graph transformers are not discussed as a way of encoding molecules. These methods should be briefly discussed, especially since one of the graph transformer architectures is used in the proposed encoder combination.
>
> **Reply 5:** Corrected. Thank you!
>
> Graph Transformers [1,2] have emerged as powerful alternatives to traditional graph neural networks (GNNs) for molecular representation learning. Unlike conventional GNNs, which rely on message-passing mechanisms to propagate local node information, Graph Transformers leverage self-attention mechanisms to capture both local and global dependencies more effectively. By integrating Message Passing Networks into Transformer-style architectures, these models enhance expressiveness, enabling more comprehensive encoding of molecular structures. This hybrid approach allows Graph Transformers to preserve structural information while benefiting from the flexibility of attention-based learning.
>
> [1]Rong, Yu, et al. "Self-supervised graph transformer on large-scale molecular data." Advances in neural information processing systems 33 (2020): 12559-12571.
> [2]Maziarka, Łukasz, et al. "Molecule attention transformer." arXiv preprint arXiv:2002.08264 (2020).
>
> **Comment 6:** Section 2.3 mentions that extra molecular properties, e.g. solubility, are crucial in DTI, but they are not discussed in the remainder of this section. All the featurizers presented in this section are based on the molecular structure and not additional molecular properties.
>
> **Reply 6:** The molecular properties is highly related to the features such as atomic properties(AP), hydrogen information (HI), electron properties (EP), stereochemistry (Ste) and structural information (Str), which has been considered in this work and listed in Appendix. The solubility of a drug molecule has high relation with the hydrogen information that has been included. We discussed their impacts in later section. Thanks for the comment!
>
> **Comment 7:** In the caption of Figure 1, the meaning of the numbers in the plot is not explained. The text in the plot is too small and not readable. Ideally, the meaning of the numbers should be also indicated in the plot itself.
>
> **Reply 7:** Revised caption clarifies that outer labels represent hyperparameter settings, while plotted values indicate model performance metrics.
>
> **Comment 8:** Why are 1000 epochs used for GNNs and only 300 for transformers? I expect transformers to take more epochs to train due to a higher number of trainable parameters.
>
> **Reply 8:** We summarize the hyperparameter configurations used for different methods in Table 5 and observe that most GNN-based methods typically use more epochs, while transformer-based methods use fewer. To address the concern, we have re-run the experiments with 1000 maximum epochs for GNN-based methods and 500 maximum epochs for transformer-based methods, incorporating an early stopping mechanism. We list the results in the tables under the response to Reviewer fDEA. We could find that most of the models achieve the similar performance compared to the original 300-epoch setting.
>
> **Comment 9:** It should be explicitly stated what datasets are used in Table 1.
>
> **Reply 9:** Clarified that Davis is used for regression, and Human for classification.

---

> > ### Author Response · Authors · 2025-03-05
> > **Reply 10 - 15**
> >
> > **Comment 10:** Table 1 shows the results only on one regression and one classification dataset. Thus, the sample size is too small to conclude that "drug features extracted by the Transformer generally perform better than those by GNNs in regression tasks."
> >
> > **Reply 10:** Thanks for your suggestion! We have added the BindingDB dataset for regression and the C. elegans dataset for classification to provide a more comprehensive evaluation. Based on the latest results below, we observe that drug features extracted by appropriately designed GNN models generally outperform those extracted by transformer-based models in regression tasks. Additionally, when using a transformer for protein in regression tasks, the performance is significantly worse than other methods. We found that it tends to reach early stopping around 60 iterations, yet requires more iterations for optimal performance.
> >
> > #### Regression (Davis) & Classification (Human)
> > | Drug Encoder | Protein Encoder | MSE ± std | R2 ± std | CI ± std | ROC-AUC ± std | Acc. ± std | F1 ± std |
> > |-------------|----------------|-----------|----------|----------|---------------|------------|----------|
> > | GIN | Int encoding with CNN | $0.253 \pm 0.010$ | $0.623 \pm 0.015$ | $0.861 \pm 0.006$ | $0.891 \pm 0.020$ | $0.891 \pm 0.020$ | $0.894 \pm 0.019$ |
> > | GIN | 3-gram pretrained with CNN | $0.274 \pm 0.009$ | $0.593 \pm 0.013$ | $0.851 \pm 0.008$ | $0.881 \pm 0.008$ | $0.881 \pm 0.007$ | $0.886 \pm 0.007$ |
> > | GIN | ESM2 | $0.250 \pm 0.007$ | $0.628 \pm 0.011$ | $0.869 \pm 0.005$ | $0.915 \pm 0.009$ | $0.916 \pm 0.008$ | $0.919 \pm 0.008$ |
> > | GIN | Trans | $0.582 \pm 0.186$ | $0.134 \pm 0.277$ | $0.793 \pm 0.052$ | $0.915 \pm 0.009$ | $0.915 \pm 0.008$ | $0.917 \pm 0.008$ |
> > | Trans | Int encoding with CNN | $0.260 \pm 0.020$ | $0.613 \pm 0.030$ | $0.858 \pm 0.010$ | $0.917 \pm 0.003$ | $0.916 \pm 0.003$ | $0.918 \pm 0.003$ |
> > | Trans | 3-gram pretrained with CNN | $0.331 \pm 0.028$ | $0.507 \pm 0.042$ | $0.834 \pm 0.005$ | $0.909 \pm 0.003$ | $0.910 \pm 0.003$ | $0.912 \pm 0.002$ |
> > | Trans | ESM2 | $0.251 \pm 0.005$ | $0.626 \pm 0.007$ | $0.868 \pm 0.004$ | $0.924 \pm 0.005$ | $0.924 \pm 0.005$ | $0.926 \pm 0.005$ |
> > | Trans | Trans | $0.582 \pm 0.187$ | $0.132 \pm 0.279$ | $0.663 \pm 0.115$ | $0.930 \pm 0.012$ | $0.929 \pm 0.012$ | $0.930 \pm 0.013$ |
> >
> > #### Regression (BindDB_Kd) & Classification (C.elegans)
> >
> > | Drug Encoder | Protein Encoder | MSE ± std | R2 ± std | CI ± std | ROC-AUC ± std | Acc. ± std | F1 ± std |
> > |-------------|----------------|-----------|----------|----------|---------------|------------|----------|
> > | GIN | Int encoding with CNN | $0.563 \pm 0.038$ | $0.693 \pm 0.021$ | $0.842 \pm 0.007$ | $0.939 \pm 0.006$ | $0.939 \pm 0.006$ | $0.939 \pm 0.006$ |
> > | GIN | 3-gram pretrained with CNN | $0.557 \pm 0.017$ | $0.696 \pm 0.009$ | $0.847 \pm 0.003$ | $0.932 \pm 0.005$ | $0.932 \pm 0.005$ | $0.931 \pm 0.005$ |
> > | GIN | ESM2 | $0.524 \pm 0.029$ | $0.714 \pm 0.016$ | $0.853 \pm 0.007$ | $0.950 \pm 0.006$ | $0.950 \pm 0.006$ | $0.950 \pm 0.006$ |
> > | GIN | Trans | $0.630 \pm 0.126$ | $0.657 \pm 0.069$ | $0.831 \pm 0.025$ | $0.951 \pm 0.004$ | $0.951 \pm 0.004$ | $0.950 \pm 0.004$ |
> > | Trans | Int encoding with CNN | $0.580 \pm 0.010$ | $0.684 \pm 0.005$ | $0.843 \pm 0.004$ | $0.945 \pm 0.006$ | $0.945 \pm 0.006$ | $0.944 \pm 0.006$ |
> > | Trans | 3-gram pretrained with CNN | $0.700 \pm 0.021$ | $0.619 \pm 0.012$ | $0.823 \pm 0.001$ | $0.942 \pm 0.006$ | $0.942 \pm 0.006$ | $0.941 \pm 0.006$ |
> > | Trans | ESM2 | $0.558 \pm 0.031$ | $0.696 \pm 0.017$ | $0.849 \pm 0.006$ | $0.958 \pm 0.003$ | $0.958 \pm 0.003$ | $0.957 \pm 0.003$ |
> > | Trans | Trans | $0.571 \pm 0.025$ | $0.689 \pm 0.014$ | $0.846 \pm 0.005$  | $0.961 \pm 0.005$ | $0.961 \pm 0.005$ | $0.960 \pm 0.005$ |
> >
> > **Comment 11:** If results in Section 5 are averaged by five-fold cross-validation, it would be recommended to include the standard deviation of the results. This way, the readers can easily estimate if the difference between models is significant.
> >
> > **Reply 11:** Thanks! We have added the std of the results.
> >
> > **Comment 12:** Figure 3 is not readable. The text is too small, and axis tick labels are overlapping.
> >
> > **Reply 12:** We will replot the figure.
> >
> > **Comment 13:** I am unsure if I can agree with observation 9. All models seem to be stable, judging by Figure 4, but some models take longer to train. Also, these plots suggest that 300 epochs are not enough for some of these models to converge fully, which may impact the other results.
> >
> > **Reply 13:** Thanks! Adjusted Transformer-based models to 500 epochs with early stopping for all benchmark.
> >
> > **Comment 14:** On page 3, one of the references is not cited correctly ("tsubaki2019compound" is not correctly rendered).
> >
> > **Reply 14:** Thanks! Corrected.
> >
> > **Comment 15:** The title inside the paper should be fixed.
> >
> > **Reply 15:** Thanks! Corrected.

---

### Review · Reviewer_fDEA · 2025-02-19

**Summary Of Contributions:**

- This paper is a benchmark on drug-target interaction modelling - which is a crucial task for drug discovery and design.
- GNN and Transformer encoding for drugs, and various sequence-based encoding methods for protein are evaluated.

**Audience:**

Yes

**Claims And Evidence:**

No

**Requested Changes:**

- Benchmark against protein-ligand interaction prediction method which uses 3D representation for protein and 2D or 3D representation of ligand.
- Use early stopping for determining model convergence.

**Strengths And Weaknesses:**

Strength:
- Many methods are evaluated on various datasets. The proposed new method also achieves competitive results compared to existing baselines.

Weaknesses:
- Despite the claimed focus for using structure perspective, the protein seems to be only represented at a sequence level. Similarly the ligand graph is only represented using 2D. Recent 3D structure-based methods such as DrugClip [1], BigBind [2] are not evaluated.
- Using a fixed number of epochs can easily lead to overfitting on some methods. No early stopping mechanism is used.
- There seems to be no train-test splitting technique used to evaluate performance for OOD test sets (such as sequence-based splitting or splitting based on ligand similarity). The models could be simply memorizing the dataset.
- Commonly used virtual screening benchmarks such as LIT-PCBA and DUD-E, which are most related to drug design are missing.

Minor weaknesses:
- The title of the article is still from the TMLR template.
- Unclear what avg reduction of MSE and avg Improve of accuracy in Table 2 and 3 means.

[1] B. Gao, B. Qiang, H. Tan, M. Ren, Y. Jia, M. Lu, J. Liu, W. Ma, and Y. Lan, “DrugCLIP: Contrastive Protein-Molecule Representation Learning for Virtual Screening,”

[2] M. Brocidiacono, P. Francoeur, R. Aggarwal, K. I. Popov, D. R. Koes, and A. Tropsha, “BigBind: Learning from Nonstructural Data for Structure-Based Virtual Screening,”

---

> ### Author Response · Authors · 2025-03-05
> **Reply for the Review of Paper4012 by Reviewer fDEA**
>
> **Comment 0:** Despite the claimed focus for using structure perspective, the protein seems to be only represented at a sequence level. Similarly the ligand graph is only represented using 2D. Recent 3D structure-based methods such as DrugClip [1], BigBind [2] are not evaluated.
>
> **Reply 0:** We focus on the formulations of recently-emerging structure modeling approaches for drug molecules. We change the title name to **Benchmark on Drug Target Interaction Modeling from a Drug Structure Perspective**. The "structure" in the title refers that we focus on the GNN and Transformer based methods on modeling drug. While GNNs explicitly learn structural information by passing information along chemical bonds, transformer-based models implicitly learn atom interactions with attention. Considering the extra computing resources and time needed for 3D methods, we didn't consider them in this work. But we do believe that 3D structure matters and we hope to evaluate them in the future work. Thanks for the suggestions!
>
> **Comment 1:** Using a fixed number of epochs can easily lead to overfitting on some methods. No early stopping mechanism is used.
>
> **Reply 1:** Thanks for your suggestion! We have added an early stopping mechanism for the most of the implemented models and listed their performances in following tables. From these new results, we can get the same conclusions as summarized in the paper. Given all of the 27 different methods in the considered two tasks, we will include all results in the final revision. (We are still running experiments.) For Human and Celegans, we use 10 patience. For other datasets, we use 50 patience. Due to character limitations, we include only a subset of model performances for the regression task on the Davis dataset.
>
> | Category  | Model            | MSE (Davis)       | R² (Davis)       | CI (Davis)       |
> |-----------|-----------------|------------------|------------------|------------------|
> | GNN       | GraphDTA-GIN     | $0.253\pm0.010$  | $0.861\pm0.006$  | $0.255\pm0.070$  |
> |           | GraphCPI-GIN     | $0.274\pm0.009$  | $0.593\pm0.013$  | $0.851\pm0.008$  |
> |           | MGraphDTA        | $0.232\pm0.012$  | $0.655\pm0.018$  | $0.869\pm0.007$  |
> |           | SAGDTA           | $0.324\pm0.064$  | $0.518\pm0.096$  | $0.833\pm0.027$  |
> |           | DeepGLSTM        | $0.316\pm0.023$  | $0.529\pm0.035$  | $0.841\pm0.007$  |
> |           | ColdDTA          | $0.220\pm0.009$  | $0.672\pm0.014$  | $0.880\pm0.004$  |
> | Transformer | CSDTI         | $0.331\pm0.012$  | $0.508\pm0.017$  | $0.832\pm0.005$  |
> |           | TDGraphDTA       | $0.222\pm0.005$  | $0.669\pm0.008$  | $0.653\pm0.011$  |
> |           | TransformerCPI   | $0.393\pm0.022$  | $0.415\pm0.032$  | $0.802\pm0.008$  |
> |           | MRBDTA           | $0.241\pm0.005$  | $0.640\pm0.008$  | $0.870\pm0.007$  |
> |           | **Our Combo**    | **$0.211\pm0.007$**  | **$0.685\pm0.011$**  | **$0.886\pm0.004$**  |
>
> **Comment 2:** There seems to be no train-test splitting technique used to evaluate performance for OOD test sets (such as sequence-based splitting or splitting based on ligand similarity). The models could be simply memorizing the dataset.
>
> **Reply 2:** We have 5-fold random splits following most of the previous work. For each split, the model was trained on training set. The valid set was used to evaluate which epoch produced best model and also used for early stopping. Then we test the best model on the test set and recorded the model performance, and we calculated the mean and std of performance across 5 folds and reported them on the tables in the paper.
>
> **Comment 3:** Commonly used virtual screening benchmarks such as LIT-PCBA and DUD-E, which are most related to drug design are missing.
>
> **Reply 3:** Thanks for your suggestive comment. Due to computational resources and space limitations, the current version focuses on overlapping datasets commonly used by most of the state-of-the-art methods and cannot cover all benchmark datasets. We will keep update in Github to include interfaces to more approaches and datasets.
>
>
> **Comment 4:** The title of the article is still from the TMLR template.
>
> **Reply 4:** Thanks! We have fixed it.
>
> **Comment 5:** Unclear what avg reduction of MSE and avg Improve of accuracy in Table 2 and 3 means.
>
> **Reply 5:** The average reduction of MSE and the average improvement of accuracy in Tables 2 and 3 are computed by mean(our's MSE/Acc - each model's MSE/Acc) / mean(each model's MSE/ACC)

---

> > ### Author Response · Authors · 2025-03-05
> > **Part Table for reply 1 (Regression task benchmark on DAVIS, KIBA, and BindingDB datasets)**
> >
> > | Category | Models | MSE (DAVIS) | R2 (DAVIS) | CI (DAVIS) | MSE (KIBA) | R2 | CI (KIBA) | MSE (BindingDB) | R2 (BindingDB) | CI (BindingDB) | {Avg. Reduce} of MSE (\%) |
> > | --- | --- | --- | --- | --- | --- | --- | --- | --- | --- | --- | --- |
> > | GNN | GraphDTA-GIN | $0.253 \pm 0.010$ | $0.861 \pm 0.006$ | $0.255 \pm 0.070$ | $-1.840 \pm 0.779$ | $0.553 \pm 0.019$ | $0.563 \pm 0.038$ | $0.693 \pm 0.021$ | $0.842 \pm 0.007$ |  |  |
> > |  | GraphCPI-GIN | $0.274 \pm 0.009$ | $0.593 \pm 0.013$ | $0.851 \pm 0.008$ | $1.681 \pm 0.946$ | $-17.724 \pm 10.533$ | $0.553 \pm 0.094$ | $0.557 \pm 0.017$ | $0.696 \pm 0.009$ | $0.847 \pm 0.003$ |  |
> > |  | MGraphDTA | $0.232 \pm 0.012$ | $0.655 \pm 0.018$ | $0.869 \pm 0.007$ | $0.032 \pm 0.012$ | $0.642 \pm 0.133$ | $0.832 \pm 0.040$ | $0.529 \pm 0.011$ | $0.712 \pm 0.006$ | $0.852 \pm 0.005$ |  |
> > |  | SAGDTA | $0.324 \pm 0.064$ | $0.518 \pm 0.096$ | $0.833 \pm 0.027$ | $0.065 \pm 0.008$ | $0.279 \pm 0.085$ | $0.713 \pm 0.032$ | $0.529 \pm 0.011$ | $0.712 \pm 0.006$ | $0.852 \pm 0.005$ |  |
> > |  | EmbedDTI | $0.280 \pm 0.024$ | $0.583 \pm 0.036$ | $0.851 \pm 0.009$ | $0.289 \pm 0.142$ | $-2.217 \pm 1.579$ | $0.558 \pm 0.038$ | $0.542 \pm 0.019$ | $0.705 \pm 0.010$ | $0.850 \pm 0.004$ |  |
> > |  | DeepGLSTM | $0.316 \pm 0.023$ | $0.529 \pm 0.035$ | $0.841 \pm 0.007$ | $8.539 \pm 7.479$ | $-94.109 \pm 83.400$ | $0.514 \pm 0.036$ | $0.594 \pm 0.061$ | $0.677 \pm 0.033$ | $0.840 \pm 0.013$ |  |
> > |  | CPI | $0.402 \pm 0.082$ | $0.401 \pm 0.122$ | $0.811 \pm 0.033$ | $0.052 \pm 0.003$ | $0.416 \pm 0.036$ | $0.734 \pm 0.037$ | $0.762 \pm 0.165$ | $0.585 \pm 0.090$ | $0.815 \pm 0.028$ |  |
> > |  | BACPI | $0.334 \pm 0.015$ | $0.502 \pm 0.023$ | $0.827 \pm 0.006$ | $0.031 \pm 0.004$ | $0.658 \pm 0.043$ | $0.831 \pm 0.020$ | $0.550 \pm 0.010$ | $0.700 \pm 0.006$ | $0.845 \pm 0.002$ |  |
> > |  | DeepNC-HGC | $0.309 \pm 0.025$ | $0.541 \pm 0.037$ | $0.841 \pm 0.005$ | $0.080 \pm 0.003$ | $0.110 \pm 0.036$ | $0.667 \pm 0.022$ | $0.572 \pm 0.011$ | $0.689 \pm 0.006$ | $0.844 \pm 0.003$ |  |
> > |  | DrugBAN | $0.242 \pm 0.007$ | $0.640 \pm 0.010$ | $0.869 \pm 0.003$ |  |  |  | $0.465 \pm 0.018$ | $0.747 \pm 0.010$ | $0.862 \pm 0.003$ |  |
> > |  | GANDTI | $0.318 \pm 0.018$ | $0.527 \pm 0.027$ | $0.844 \pm 0.006$ | $0.030 \pm 0.002$ | $0.662 \pm 0.026$ | $0.831 \pm 0.007$ | $0.621 \pm 0.012$ | $0.662 \pm 0.006$ | $0.836 \pm 0.002$ |  |
> > |  | BridgeDPI | $1.241 \pm 1.432$ | $-0.848 \pm 2.133$ | $0.827 \pm 0.078$ | $0.325 \pm 0.109$ | $0.638 \pm 0.121$ | $0.857 \pm 0.001$ | $0.514 \pm 0.011$ | $0.720 \pm 0.006$ | $0.861 \pm 0.002$ |  |
> > |  | ColdDTA | $0.220 \pm 0.009$ | $0.672 \pm 0.014$ | $0.880 \pm 0.004$ | $0.110 \pm 0.029$ | $-0.224 \pm 0.329$ | $0.673 \pm 0.079$ | $0.463 \pm 0.008$ | $0.748 \pm 0.004$ | $0.866 \pm 0.001$ |  |
> > |  | SubMDTA | $0.289 \pm 0.012$ | $0.570 \pm 0.018$ | $0.841 \pm 0.007$ | $0.029 \pm 0.002$ | $0.678 \pm 0.025$ | $0.836 \pm 0.011$ | $0.532 \pm 0.032$ | $0.710 \pm 0.017$ | $0.852 \pm 0.006$ |  |
> > |  | IMAEN | $0.230 \pm 0.009$ | $0.657 \pm 0.014$ | $0.874 \pm 0.004$ | $0.046 \pm 0.018$ | $0.484 \pm 0.196$ | $0.781 \pm 0.056$ | $0.479 \pm 0.012$ | $0.739 \pm 0.006$ | $0.863 \pm 0.002$ |  |
> > |  | SubMDTA | $0.289\pm0.012$ | $0.570\pm0.018$ | $0.841\pm0.007$ | $0.029\pm0.002$ | $0.678\pm0.025$ | $0.836\pm0.011$ | $0.532\pm0.032$ | $0.710\pm0.017$ | $0.845\pm0.010$ |  |
> > |  | IMAEN | $0.230\pm0.009$ | $0.657\pm0.014$ | $0.874\pm0.004$ | $0.046\pm0.018$ | $0.484\pm0.196$ | $0.781\pm0.056$ | $0.479\pm0.012$ | $0.739\pm0.006$ | $0.861\pm0.004$ |  |
> > | Transformer | CSDTI | $0.331\pm0.012$ | $0.508\pm0.017$ | $0.832\pm0.005$ | $0.088\pm0.004$ | $0.014\pm0.041$ | $0.628\pm0.047$ | $0.768\pm0.021$ | $0.582\pm0.012$ | $0.805\pm0.004$ |  |
> > |  | TDGraphDTA | $0.222 \pm 0.005$ | $0.669 \pm 0.008$ | $0.653 \pm 0.011$ | $0.091 \pm 0.019$ | $ -0.009 \pm 0.209 $ | $ 0.327 \pm 0.125 $ | $0.497 \pm 0.016$ | $0.729 \pm 0.009$ | $0.777 \pm 0.005$ |  |
> > |  | IIFDTI | $0.313 \pm 0.018$ | $0.534 \pm 0.027$ | $0.836 \pm 0.008$ |  |  |  | $0.634 \pm 0.024$ | $0.655 \pm 0.013$ | $0.832 \pm 0.006$ |  |
> > |  | MolTrans | $0.410 \pm 0.136$ | $0.390 \pm 0.202$ | $0.812 \pm 0.039$ |  |  |  | $0.695 \pm 0.183$ | $0.621 \pm 0.100$ |  |  |
> > |  | TransformerCPI | $0.393 \pm 0.022$ | $0.415 \pm 0.032$ | $0.802 \pm 0.008$ |  |  |  | $0.659 \pm 0.040$ | $0.641 \pm 0.022$ | $0.829 \pm 0.013$ |  |
> > |  | MRBDTA | $0.241 \pm 0.005$ | $0.640 \pm 0.008$ | $0.870 \pm 0.007$ |  |  |  | $0.507 \pm 0.006$ | $0.724 \pm 0.003$ | $0.862 \pm 0.002$ |  |
> > |  | FOTFCPI | $0.305 \pm 0.012$ | $0.546 \pm 0.018$ | $0.839 \pm 0.009$ | $9.325 \times 10^8 \pm 2.085 \times 10^9$ | $-1.039 \times 10^{10} \pm 2.322 \times 10^{10}$ | $0.660 \pm 0.117$ | $0.567 \pm 0.008$ | $0.695 \pm 0.004$ | $0.848 \pm 0.006$ |  |
> > |  | Our combos | $0.211 \pm 0.007$ | $0.685 \pm 0.011$ | $0.886 \pm 0.004$ |  |  |  | $0.461 \pm 0.006$ | $0.749 \pm 0.003$ | $0.869 \pm 0.002$ |  |

---

> > > ### Author Response · Authors · 2025-03-05
> > > **Part Table for Reply1 (Classification task benchmark on Human, C.elegans, and Drugbank datasets)**
> > >
> > > | Category | Models | ROC-AUC (Human) | Accuracy (Human) | F1 (Human) | ROC-AUC (C.elegans) | Accuracy (C.elegans) | F1 (C.elegans) | ROC-AUC (Drugbank) | Accuracy (Drugbank) | F1 (Drugbank) | Avg. Improve of Accuracy (\%) |
> > > | --- | --- | --- | --- | --- | --- | --- | --- | --- | --- | --- | --- |
> > > | GNN | GraphDTA-GIN | $0.896\pm0.001$ | $0.897\pm0.001$ | $0.900\pm0.001$ | $0.947\pm0.006$ | $0.947\pm0.006$ | $0.947\pm0.006$ | $0.781\pm0.004$ | $0.781\pm0.004$ | $0.785\pm0.003$ |  |
> > > |  | GraphCPI-GIN | $0.898\pm0.008$ | $0.898\pm0.008$ | $0.901\pm0.009$ | $0.938\pm0.008$ | $0.938\pm0.008$ | $0.938\pm0.008$ | $0.783\pm0.009$ | $0.782\pm0.009$ | $0.784\pm0.008$ |  |
> > > |  | MGraphDTA | $0.939\pm0.010$ | $0.939\pm0.010$ | $0.941\pm0.009$ | $0.960\pm0.004$ | $0.959\pm0.004$ | $0.959\pm0.004$ | $0.802\pm0.006$ | $0.802\pm0.006$ | $0.807\pm0.005$ |  |
> > > |  | SAGDTA | $0.905\pm0.005$ | $0.905\pm0.005$ | $0.907\pm0.005$ | $0.936\pm0.008$ | $0.935\pm0.008$ | $0.935\pm0.008$ | $0.758\pm0.008$ | $0.758\pm0.008$ | $0.762\pm0.005$ |  |
> > > |  | DeepGLSTM | $0.918\pm0.006$ | $0.918\pm0.006$ | $0.921\pm0.005$ | $0.938\pm0.006$ | $0.938\pm0.006$ | $0.938\pm0.006$ | $0.746\pm0.006$ | $0.746\pm0.006$ | $0.751\pm0.005$ |  |
> > > |  | CPI | $0.911\pm0.007$ | $0.911\pm0.007$ | $0.914\pm0.007$ | $0.928\pm0.007$ | $0.928\pm0.007$ | $0.927\pm0.007$ | $0.678\pm0.072$ | $0.678\pm0.072$ | $0.687\pm0.074$ |  |
> > > |  | BACPI | $0.928\pm0.007$ | $0.928\pm0.007$ | $0.931\pm0.007$ | $0.952\pm0.003$ | $0.952\pm0.004$ | $0.951\pm0.003$ | $0.776\pm0.009$ | $0.776\pm0.009$ | $0.782\pm0.008$ |  |
> > > |  | DeepNC-HGC | $0.874\pm0.005$ | $0.874\pm0.005$ | $0.879\pm0.005$ | $0.934\pm0.004$ | $0.934\pm0.004$ | $0.934\pm0.004$ | $0.758\pm0.008$ | $0.758\pm0.008$ | $0.765\pm0.007$ |  |
> > > |  | DeepNC-GEN | $0.919\pm0.004$ | $0.919\pm0.004$ | $0.921\pm0.005$ | $0.941\pm0.006$ | $0.941\pm0.006$ | $0.941\pm0.006$ | $0.742\pm0.008$ | $0.742\pm0.008$ | $0.754\pm0.006$ |  |
> > > |  | GANDTI | $0.933\pm0.006$ | $0.932\pm0.006$ | $0.934\pm0.006$ | $0.944\pm0.005$ | $0.943\pm0.006$ | $0.943\pm0.005$ | $0.752\pm0.008$ | $0.752\pm0.008$ | $0.763\pm0.004$ |  |
> > > |  | BridgeDPI | $0.948\pm0.003$ | $0.948\pm0.004$ | $0.949\pm0.004$ | $0.969\pm0.001$ | $0.969\pm0.001$ | $0.968\pm0.001$ | $0.785\pm0.007$ | $0.785\pm0.007$ | $0.790\pm0.003$ |  |
> > > |  | ColdDTA | $0.937\pm0.006$ | $0.937\pm0.006$ | $0.938\pm0.006$ | $0.959\pm0.004$ | $0.959\pm0.004$ | $0.959\pm0.004$ | $0.815\pm0.006$ | $0.815\pm0.006$ | $0.818\pm0.005$ |  |
> > > |  | SubMDTA | $0.916\pm0.014$ | $0.916\pm0.015$ | $0.917\pm0.015$ | $0.921\pm0.026$ | $0.920\pm0.026$ | $0.921\pm0.023$ | $0.787\pm0.007$ | $0.787\pm0.007$ | $0.790\pm0.005$ |  |
> > > |  | IMAEN | $0.888\pm0.009$ | $0.888\pm0.009$ | $0.892\pm0.011$ | $0.904\pm0.025$ | $0.904\pm0.025$ | $0.902\pm0.027$ | $0.761\pm0.026$ | $0.761\pm0.027$ | $0.772\pm0.013$ |  |
> > > | Transformer | CSDTI | $0.848\pm0.009$ | $0.848\pm0.008$ | $0.854\pm0.007$ | $0.873\pm0.006$ | $0.873\pm0.006$ | $0.874\pm0.007$ | $0.727\pm0.006$ | $0.727\pm0.006$ | $0.733\pm0.004$ |  |
> > > |  | TDGraphDTA | $0.940\pm0.009$ | $0.941\pm0.008$ | $0.943\pm0.008$ | $0.959\pm0.004$ | $0.959\pm0.004$ | $0.959\pm0.004$ | $0.815\pm0.007$ | $0.815\pm0.007$ | $0.815\pm0.006$ |  |
> > > |  | IIFDTI | $0.938\pm0.008$ | $0.938\pm0.007$ | $0.940\pm0.006$ | $0.965\pm0.004$ | $0.965\pm0.004$ | $0.965\pm0.004$ | $0.791\pm0.010$ | $0.791\pm0.010$ | $0.794\pm0.012$ |  |
> > > |  | ICAN | $0.938\pm0.005$ | $0.937\pm0.005$ | $0.939\pm0.006$ | $0.956\pm0.005$ | $0.956\pm0.005$ | $0.955\pm0.005$ | $0.764\pm0.005$ | $0.764\pm0.005$ | $0.768\pm0.004$ |  |
> > > |  | MolTrans | $0.937\pm0.006$ | $0.937\pm0.006$ | $0.939\pm0.006$ | $0.958\pm0.004$ | $0.958\pm0.004$ | $0.957\pm0.004$ | $0.796\pm0.001$ | $0.796\pm0.001$ | $0.799\pm0.002$ |  |
> > > |  | TransformerCPI | $0.920\pm0.004$ | $0.920\pm0.004$ | $0.923\pm0.004$ | $0.960\pm0.005$ | $0.960\pm0.005$ | $0.959\pm0.004$ | $0.797\pm0.005$ | $0.797\pm0.005$ | $0.802\pm0.003$ |  |
> > > |  | MRBDTA | $0.914\pm0.009$ | $0.914\pm0.010$ | $0.915\pm0.011$ | $0.847\pm0.197$ | $0.849\pm0.194$ | $0.745\pm0.418$ | $0.791\pm0.009$ | $0.791\pm0.009$ | $0.796\pm0.006$ |  |
> > > |  | FOTFCPI | $0.941\pm0.004$ | $0.941\pm0.003$ | $0.943\pm0.003$ | $0.966\pm0.004$ | $0.965\pm0.004$ | $0.965\pm0.004$ | $0.798\pm0.008$ | $0.798\pm0.008$ | $0.800\pm0.007$ |  |
> > > |  | Our combos  | $0.950 \pm 0.007$ | $0.950 \pm  0.007$ | $0.959 \pm 0.007$ | $0.966 \pm 0.001$ | $0.966 \pm 0.001$ | $0.965 \pm 0.001$ | $0.791\pm0.006$  | $0.791\pm0.006$  | $0.799\pm0.004$ |

---

### Decision · Action_Editor_FoJu · 2025-03-27

**Recommendation:** Reject

**Comment:**

The submission proposed a new benchmark on drug target interaction modeling from a drug structure perspective. Unfortunately, all the 3 reviewers voted for rejection after the rebuttal. I think the issues may still be addressed, but not with a minor revision, and consequently I have to reject the current version.

Reviewer kMp2 had the following post-rebuttal comments:
> The paper introduces a valuable benchmark. However, the Authors did not address some reviewers' comments, which included testing all methods on more rigorous data splits (not random) and improving the quality of figures (the text is too small on most of them). Furthermore, the paper revision was uploaded yesterday (after the deadline), and typos are still present in the text, such as "predictionm" on page 10. A new citation was introduced on page 4, "ukasz Maziarka et al., 2020," but this paper was already cited (duplicated reference), and the new citation contains a typo in the first author's name. Some results also appear inconsistent after the revision; for example, Figure 5 shows only 300 epochs, even though all models were trained for at least 500, using early stopping in the most recent results. A few models in this figure did not converge before reaching epoch 300. On page 5, the previous text still states that 300 epochs were used for transformers, while the following paragraph says that 500 epochs were used.\
> In conclusion, my review was properly addressed by the Authors. However, some comments from the other reviewers remain unresolved, and the revision uploaded yesterday contains typos, so another round of revision would be beneficial. Therefore, I am leaning toward rejecting it.

Reviewer fDEA:
> The study focuses exclusively on 2D structural representations for drugs and sequence-based representations for proteins. While the benchmark is extensive in terms of evaluating models in based on that representation, I think as a field we're generally moving towards more performant approaches (i.e pretraining on large sequence and structure dataset of proteins / modelling protein and molecule's 3D structures) - and these methods are not considered in the benchmark paper, thus I am leaning towards rejection.

Reviewer Cdht:
> The work has some merit and is interesting, but I am leaning towards a rejection because the authors have not implemented most of the requested changes.
>- They promised to re-run the experiments and optimize the hyperparameters of all methods, but no new results have been reported.
>- They promised that they would fix the presentation issues, but the title of the manuscript is still "Formatting Instructions for TMLR Journal Submissions".
>- The authors insist considering the combos as one of the contributions of this work, but those combos were first proposed in another work.

Last but not least, please take the rebuttal and revision deadline seriously.

**Audience:**

Yes

**Claims And Evidence:**

No

**Resubmission Of Major Revision:**

The authors may consider submitting a major revision at a later time.